# Comparison of the oxygen isotope signatures in speleothem records and iHadCM3 model simulations for the last millennium

Janica C. Bühler[1], Carla Roesch[1], Moritz Kirschner[1], Louise Sime[2], Max D. Holloway[3], and Kira Rehfeld[1]

[1]Institute of Environmental Physics, Ruprecht-Karls-Universität Heidelberg, INF 229, 69120 Heidelberg, Germany
[2]British Arctic Survey, High Cross, Madingley Road, Cambridge, CB3 0ET, United Kingdom
[3]Scottish Association for Marine Science, Scottish Marine Institute, Oban, Argyll, PA37 1QA, United Kingdom

**Correspondence:** Janica Bühler (jbuehler@iup.uni-heidelberg.de), Kira Rehfeld (krehfeld@iup.uni-heidelberg.de)

**Abstract.**

Improving the understanding of changes in the mean and variability of climate variables as well as their interrelation is crucial for reliable climate change projections. Comparisons between general circulation models and paleoclimate archives using indirect proxies for temperature or precipitation have been used to test and validate the capability of climate models to represent climate changes. The oxygen isotopic ratio $\delta^{18}O$, a proxy for many different climate variables, is routinely measured in speleothem samples at decadal or higher resolution and single specimens can cover full Glacial-Interglacial cycles. The calcium carbonate cave deposits are precisely dateable and provide well preserved (semi-) continuous, albeit multivariate climate signals in the lower and mid-latitudes, where the measured $\delta^{18}O$ in the mineral does not directly represent temperature or precipitation. Therefore, speleothems represent suitable archives to assess climate model abilities of simulating climate variability beyond the timescales covered by meteorological observations ($10^1 - 10^2$ yr).

Here, we present three transient isotope-enabled simulations from the Hadley Center Climate Model version 3 (iHadCM3) covering the last millennium (850-1850CE) and compare them to a large global dataset of speleothem $\delta^{18}O$ records from the Speleothem Isotopes Synthesis and AnaLysis (SISAL) database version 2 (Comas-Bru et al., 2020). We systematically evaluate offsets in mean and variance of simulated $\delta^{18}O$ and test for the main climate drivers recorded in $\delta^{18}O$ for individual records or regions.

The time-mean spatial offsets between the simulated $\delta^{18}O$ and the speleothem data are fairly small. However, using robust filters and spectral analysis, we show that the observed archive-based variability of $\delta^{18}O$ is lower than simulated by iHadCM3 on decadal, and higher on centennial timescales. Most of this difference can likely be attributed to the records' lower temporal resolution and averaging or smoothing processes affecting the $\delta^{18}O$ signal e.g. through soil water residence times. Using cross-correlation analyses at site-level and modeled gridbox level, we find evidence for highly variable but generally low signal-to-noise ratios in the proxy data. This points at a high influence of cave-internal processes and regional climate particularities and could suggest low regional representativity of individual sites. Long-range strong positive correlations dominate the speleothem correlation network but are much weaker in the simulation. One reason for this could lie in a lack of longterm internal climate variability in these model simulations, which could be tested by repeating similar comparisons with other isotope-enabled climate models and paleoclimate databases.

# 1 Introduction

The impacts of a changing climate have been observed over the last century (IPCC, 2013) and indicate a strong sensitivity of human societies and natural systems to changes in climate. While the mean state of the climate is well observed, direction and magnitude of potential changes to its variability are still largely unclear (Franzke et al., 2020). However, changes in variability influence the occurrence of extreme temperature and precipitation events (Katz and Brown, 1992) and have major impacts on society, economy (Hänsel et al., 2020), and ecosystems (Vasseur et al., 2014).

Past climate changes provide a testbed to evaluate climate models and to better understand projected changes in the future (Schmidt et al., 2012; Braconnot et al., 2012). Instrumental records only cover a short period of time, since systematic observations of climate variables only began in 1750 CE (black line in Fig. 1a, Morice et al., 2012). For model evaluation on longer than centennial time scales, we have to rely on evidence from paleoclimate archives, such as trees, ice cores, foraminifera from marine sediment cores, or speleothems. The abundance of the heavy oxygen isotope $^{18}O$, further denoted as $\delta^{18}O$, is a proxy for many climate variables and can be measured on these, and quite a few other paleoclimate archives with high precision (Schmidt et al., 2014). The climatic interpretation of $\delta^{18}O$ changes, however, are not always straightforward (Fairchild and Baker, 2012). Speleothem archives, which we rely on here, allow sampling of a wide range of climates in the low- to mid-latitudes and provide (semi-)continuous precisely dated time series of oxygen isotope ratios.

Few other transient model-data comparison studies focused on $\delta^{18}O$ (e.g., Wackerbarth et al., 2012; Dee et al., 2015; Colose et al., 2016; Stevenson et al., 2019; Parker et al., 2020). For example, Sjolte et al. (2018) compared the variability of the simulated ECHAM5/MPI-OM $\delta^{18}O$ to Greenland ice cores over the last millennium assimilating the ice core data to produce gridded reconstructions. They were able to differentiate between solar and volcanic forcing effects from their reconstructions. On orbital timescales (150,000 yr), Caley et al. (2014) compared a transient isotope-enabled simulation with the model of intermediate complexity iLOVECLIM to speleothem records from South East Asia. They found model-data similarity for the broad temporal trends, but differences at shorter timescales, highlighting the role of seasonality.

For our model-data comparison, we focus on the time period of the last millennium (850-1850CE Taylor et al. (2012)) for which a fairly high number of well-preserved datasets are available. This time period is characterized by stable, close-to-present-day, boundary conditions (fairly constant greenhouse gas concentrations and sea level) and climate variability due to natural, solar and volcanic forcings (Schurer et al., 2014; PAGES2k-Consortium, 2019; Taylor et al., 2012; Neukom et al., 2019). It is also one of the key paleoclimate periods included in the joint experiments of the Paleoclimate Model Intercomparison Project Phase 3 and 4 (PMIP3/PMIP4, Jungclaus et al., 2010; Kageyama et al., 2018) and the overarching Coupled Model Intercomparison Project Phase 5 and 6 (CMIP5/CMIP6, Taylor et al., 2012; Eyring et al., 2016).

Modeled climate variability can be a consequence of either internal interactions and processes (internal variability) or of radiative forcings such as depicted in Fig. 1c-e (forced external variability), e.g. greenhouse gases, volcanic eruptions, or total solar irradiance.

Previous studies have suggested that simulated temperature variability is systematically too low on decadal and longer time scales, especially on the regional scale (Laepple and Huybers, 2014a). This has been attributed to models being too diffusive,

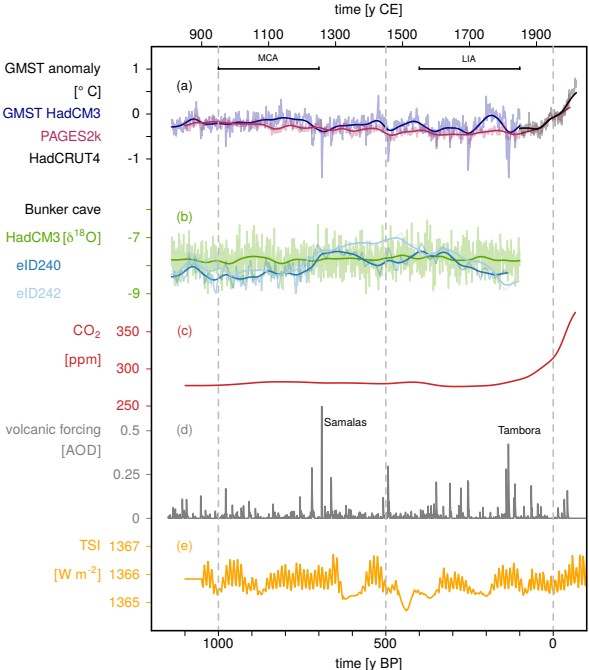

**Figure 1.** Climate and main climatic drivers over the last millennium. (a) Global annual mean surface temperature (GMST) as modeled (iHadCM3, blue), reconstructed (PAGES2k, red) (PAGES2k-Consortium, 2019) and observed (HadCRUT4, black) (Morice et al., 2012). (b) Annual mean $\delta^{18}$O in precipitation at Bunker Cave, Germany as modeled (iHadCM3, green) and measured calcite $\delta^{18}$O (drip water equivalent) from SISAL entity 240 (dark blue) and 242 (light blue) at the Bunker cave site (Comas-Bru et al., 2019; Fohlmeister et al., 2012). Comparison plots for all entities are given in supplementary figures SF1-2. (c) Atmospheric $CO_2$ concentration, (d) volcanic forcing in units of aerosol optical depth (AOD) (Crowley and Unterman, 2013) and (e) total solar irradiance (TSI) as used in the model simulations (Steinhilber et al., 2009; Wang et al., 2005).

denoting energy being dissipated too quickly across temporal scales (Laepple and Huybers, 2014b), or to missing processes and feedbacks (Rehfeld et al., 2016). Variability induced by external radiative forcing only accounts for a small fraction of the regional climate variance (Goosse et al., 2005; Laepple and Huybers, 2014b). Discrepancies increase towards longer timescales (Laepple and Huybers, 2014a), and are substantial already at the multidecadal to centennial timescales that we target here.

The incorporation of an isotopic water cycle into isotope-enabled General Circulation Models (iGCM) provides additional means for understanding the hydrology of the climate system (Werner et al., 2016; Sturm et al., 2010; Tindall et al., 2009). The ratio of $H_2{}^{18}O$ to $H_2{}^{16}O$ in precipitation is an indicator of evaporation temperature, precipitation amount, and altitude, as well as distance to source water (Dansgaard, 1964). It is given in the $\delta$-notation as

$$\delta^{18}\mathrm{O} = \left(\frac{\frac{^{18}O}{^{16}O}\text{ sample}}{\frac{^{18}O}{^{16}O}\text{ standard}} - 1\right) \cdot 1000\text{‰},$$

where standard indicates the Vienna Standard Mean Ocean Water standard V-SMOW (Kendall and Caldwell, 1998).

On monthly to decadal time scales, the Global Network of Isotopes in Precipitation (GNIP) database (IAEA/WMO, 2020) provides measurements of $\delta^{18}O$ in collected precipitation water, which have been used in model-data comparisons for the present climate (Tindall et al., 2009; Werner et al., 2011; Comas-Bru et al., 2020). On decadal and longer timescales, paleoclimate archives such as speleothems are crucial. $\delta^{18}O$ variations in stalagmites, to first order, represent changes in $\delta^{18}O$ in the

meteoric precipitation above the cave.

Speleothem cave deposits form in karst regions (Fairchild and Baker, 2012) under climatic conditions spanning from extremely cold (Lauritzen and Lundberg, 1999) and very arid (Neff et al., 2001) to extremely hot and humid conditions (Partin et al., 2007). As a terrestrial climate archive, they are able to store information on continental climate changes. They form as a calcite or aragonite matrix from calcium dissolved in acidic drip water and hence archive the oxygen isotope from precipitation

water in accumulated growth layers (Fairchild and Treble, 2009). $\delta^{18}O$ can be regarded as a proxy e.g. for surface temperature variations in higher latitudes, or precipitation amount in the tropics (Dansgaard, 1964). The proxy signal is, however, overlayed with distinct observable signatures of source water evaporation, transportation over longer distances (Bradley, 1999; Dansgaard, 1964), and large scale-climate patterns of circulation such as e.g. the North Atlantic Oscillation (NAO) (e.g. Vinther et al., 2010) or the El-Niño Southern Oscillation (ENSO) (Tindall et al., 2009). All these $\delta^{18}O$ signatures in precipitation may

be visible in speleothem records, including additionally fractionation processes involved in the calcite formation, which is primarily temperature-dependent (Urey, 1948; McCrea, 1950). The climatic interpretation of speleothem $\delta^{18}O$ variations in calcite or aragonite (hereafter $\delta^{18}O_{\text{speleo}}$) can be hampered by non-linear growth processes (Dreybrodt and Scholz, 2011), and multiple cave-specific parameters such as vegetation cover (Haude, 1954; Wackerbarth et al., 2010), karst (Jean-Baptiste et al., 2019), and inner cave processes (Fairchild et al., 2006), which influence $\delta^{18}O_{\text{speleo}}$. Especially in the comparison be-

tween $\delta^{18}O_{\text{speleo}}$ of different speleothems, dating uncertainties complicate the assessment of climatic drivers, as they increase the uncertainty in pairwise comparisons and similarity estimates (Breitenbach et al., 2012; Rehfeld and Kurths, 2014). For speleothems, in particular, positive correlations to ice core $\delta^{18}O$, which is considered a proxy for temperature, have been reported (McDermott et al., 2001) but also negative correlations to local annual mean temperatures at the cave site (e.g. Lauritzen and Lundberg (1999)). This highlights the complexity of the system and the potential regionality of the signal. In studies on

drip water, $\delta^{18}O$ and annual mean temperature, regions with different dominant climate controls could be distinguished (Baker et al., 2019).

Here, we present three new last millennium isotope-enabled simulations from the iGCM version 3 of the Hadley Model (iHadCM3) and test how similar the $\delta^{18}O$ variations in iHadCM3 and speleothem records are (Sec. 4.1). A characterization of the datasets and relevant forcing can be found in Fig. 1. The robustness of the findings and methods are evaluated over

the last millennium, for which a large number of high-resolution proxy datasets from the SISAL v.2. database (Comas-Bru et al., 2020) are available. Our key questions are: i) how similar are the modeled $\delta^{18}O$ signatures to the speleothem records especially regarding variability, ii) can we distinguish main drivers for these signatures, and iii) how representative are the speleothem records for their region. To address these questions, we explore similarities on both spatial and temporal scales, to distinguish patterns of the mean state (Sec. 4.1), the variability (Sec. 4.2 and Sec. 4.3), and the spatial representativity

of speleothem climate records (Sec. 4.4 and Sec. 4.5). We examine the simulation's capability to simulate and the records' capability to capture variability on different time scales to improve our understanding of processes and uncertainties of both.

## 2  Data

### 2.1  Model description and simulation overview

In this study, we use the coupled atmosphere-ocean isotope-enabled GCM iHadCM3, which has been widely used to simulate present and future climate (Sime et al., 2008; Tindall et al., 2009; IPCC, 2013), as well as for past climates such as the late Holocene and Last Glacial Maximum (Holloway et al., 2016), the last interglacial (Sime et al., 2009, 2013; Holloway et al., 2016, 2018) and the Eocene (Tindall et al., 2010).

The model consists of several components: the atmosphere model HadAM3 (Pope et al., 2000), the ocean model HadOM3 (Gordon et al., 2000), a sea ice model (Valdes et al., 2017) and a dynamic land surface and vegetation model (Cox, 2001). The atmospheric component is run at a horizontal resolution of $2.5° \times 3.75°$, 19 vertical levels and time steps of $30\,\mathrm{min}$. The oceanic output has a horizontal resolution of $1.25° \times 1.25°$, 20 vertical levels and time steps of 1h. For the isotope-enabled version, water isotopes $HD^{16}O$ and $H_2^{18}O$ were added as two separate water species in the atmospheric model, and as tracers in the ocean model. Fixed isotope fractions are added to a fixed volume gridbox of the ocean and experience changes due to evaporation, precipitation, and runoff through a virtual isotope flux, altering the $\delta^{18}O$ ratio in the top level of the ocean accordingly (Tindall et al., 2009). The land surface and vegetation evolve dynamically and are based on TRIFFID (Cox, 2001) with timesteps of $5\,\mathrm{yr}$.

Compared to instrumental observations, the model represents sea surface temperature (SST), sea ice, and ocean heat content well (Gordon et al., 2000). The freshwater hydrological cycle in the model shows only a slight overestimation in the local evaporation (Pardaens et al., 2003). The model simulates the major isotopic fractionation effects as in Dansgaard (1964) (e.g. the latitude effect, the amount effect, and the continental effect) appropriately compared to GNIP data (Zhang et al., 2012). Additionally, a broad agreement in isotopic output with GNIP data in the general spatial distribution can be observed and the above mentioned general oxygen isotopic ratio features are represented well (Tindall et al., 2009). As such, iHadCM3 captures large scale features of climate and oxygen isotope ratios while remaining computationally efficient for the simulation of timescales such as the last millennium. The three ensemble members, which are identified with the *LM* prefix, were initialized from different years of the same spinup simulation. The basic characteristics and boundary conditions of the last millennium simulations used in this analysis are listed in Tab. 1.

### 2.2  The speleothem isotope dataset

The oxygen isotope ratio measured in speleothems is subject to many processes, starting from the source water which is influenced by the atmospheric circulation and climate. Therefore, the amount of precipitation, its composition, the annual

**Table 1.** Basic characterization of the *LM1, LM2* and *LM3* last millennium simulations.

| | |
|---|---|
| Years | 850-1850 CE |
| | 1100 - 100BP |
| Orography | fixed to 0BP |
| Orbital Parameter | fixed to 0BP |
| GHG | well mixed $CO_2$, $CH_4$, $NO_2$ and |
| | other trace gases; Schurer et al. (2014), |
| | Schmidt et al. (2012) |
| Vegetation | dynamic; based on Cox (2001) |
| Total Solar Irradiance | Steinhilber et al. (2009) |
| | Wang et al. (2005) |
| | Schurer et al. (2014) |
| Volcanic Forcing | Crowley and Unterman (2013) |

mean temperature, and the variability of these events are in part imprinted in the archive. A comprehensive summary of the processes involving speleothem growth can be found in Fairchild and Baker (2012).

Vegetation above the cave can alter the amount of infiltrating water and its isotopic signature, where the meteoric $\delta^{18}O$ is subject to additional fractionation processes and seasonal effects (Haude, 1954; Thornthwaite and Mather, 1957; Wackerbarth
et al., 2010). Filter processes and transportation through the soil and upper karst influence the signal and may lead to varying transit times between several minutes and multiple years (Jean-Baptiste et al., 2019) at different drip sites within the same cave. Infiltrating surface water is charged with soil gas $CO_2$, where the partial $CO_2$ pressure is larger than in the atmosphere, facilitating the carbonic acid-driven $CaCO_3$ dissolution of the host rock. The generally lower partial $pCO_2$ pressure conditions in the cave environment compared to that of the soil and epikarst makes the drip water degas and precipitate calcite in a
fractionation process, which consequently forms a speleothem (Tremaine et al., 2011).

Varying environmental conditions within the cave can also be imprinted in the isotopic signal and may pronounce or attenuate the climate signal (Fairchild and Baker, 2012). During the calcification process, interactions with the cave environment or water inclusions within the mineral are still possible and, therefore, may further change the $\delta^{18}O_{speleo}$ archived in the speleothem.

The oxygen isotope composition of dripwater is influenced by all above-mentioned factors. Due to the multivariate processes
impacting speleothem growth, the interpretation of the $\delta^{18}O_{speleo}$ signal is not straightforward, although systematic evaluation has identified patterns of similar climate influence based on modern observations (Baker et al., 2019). Proxy System Models

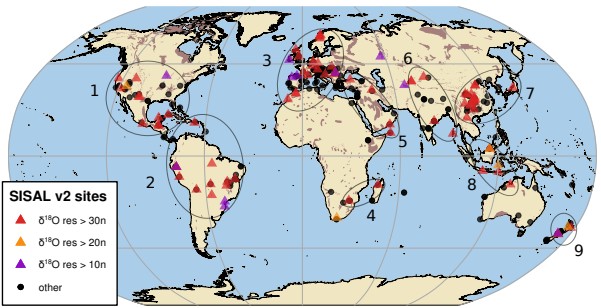

**Figure 2.** Site locations of the SISAL database on a global karst map (brown shadings Williams and Ford, 2006). The sites with entities that fulfill the prerequisites for our analysis are marked in colored triangles. These entities cover at least a period of $600\,\mathrm{yr}$ within the last millennium and have a minimum of 30 (red), 20 (orange), or 10 (purple) $\delta^{18}\mathrm{O}$ measurements and two dating points in this period. All other sites in the SISAL database v.2. are marked with a black dot. The nine clusters used in the network analysis contain sites in North America (c1, 12 entities), South America (c2, 12 entities), Europe with North Africa (c3, 21 entities), Southern Africa (c4, 2 entities - too few for systematic analysis), Middle East (c5, 6 entities), India and Central Asia (c6, 8 entities), East Asia (c7, 18 entities), South East Asia (c8, 3 entities), and New Zealand (c9, 3 entities).

(PSMs), where the input signal modification is modeled based on known processes in the karst may also help with the interpretation (Evans et al., 2013; Dee et al., 2015). PSMs of varying complexity have been proposed from the simple exponential decay filter, mimicking karst mixing (Dee et al., 2015) with the delay time as the single tunable parameter, to full-blown karst
system models with numerous parameters describing soil water and gas equilibration or carbonate bedrock dissolution (Owen et al., 2018).

The Speleothem Isotopes Synthesis and Analysis is an international working group, collecting speleothem datasets in a quality-controlled and cross-references database with rich metadata for samples and dating procedures (Atsawawanunt et al., 2018; Comas-Bru et al., 2020). The second version of the database SISAL v.2. includes measurements of stable $^{13}\mathrm{C}$ and
$^{18}\mathrm{O}$ isotopes on speleothems of 691 individual entities from 294 globally distributed sites (Comas-Bru et al., 2020). In order to provide a comprehensive and reliable analysis, we only use data from entities which are not superseded ($entity\_state = current$) and that cover at least a $600\,\mathrm{y}$ period within the analysis period (850-1850CE). Furthermore, records considered must have at least two radiometric dates, or one radiometric date (in the analysis period) and be marked as actively forming at the time of collection, or be lamina counted. We only check for dates that are marked as *used*, indicating that they are known
to have been used in the original chronology in the database. Samples without sample or depth information, are omitted. In the analysis, we filter the database adapted to the requirements of the different analyses, as depicted in Fig. 2. For the last millennium, we remain with 110 records from 91 different sites with at least ten isotopic measurements, that we used for the assessment of the mean $\delta^{18}\mathrm{O}$ offset, and 85 records from 71 sites with at least 30 isotope samples for the correlation and network analyses.

For each U/Th-dated speleothem, SISAL v.2. provides the original age model (if available), and new possible age-models based on up to seven methods. Methods include linear interpolation, linear regression, Bchron (Haslett and Parnell, 2008) as adapted by Roesch and Rehfeld (2019), Bacon (Blaauw and Christeny, 2011), Oxcal (Ramsey, 2009), copRa (Comas-Bru et al., 2020, modified R version after Breitenbach et al. (2012)), and StalAge (Scholz and Hoffmann, 2011). Details on the automated age modeling procedure are given in Roesch and Rehfeld (2019) and Comas-Bru et al. (2020). For each entity and ensemble method, one median best-fit estimate with confidence intervals, and between 129 and 7737 age-models based on perturbations of the radiometric ages are available (Comas-Bru et al., 2020). These ensembles are available for 69 of the 85 entities that we used in the network-correlation analysis, resulting in a total of 464383 ensemble age-models in our analysis. In all other analyses, we use the corresponding original age model as provided by the original authors.

## 3 Methods

### 3.1 Speleothem analysis and drip water conversion

To increase the robustness of the results, we maximize the number of records by adaptive filtering of the database (Fig. 2). For calculations involving the time-averaged $\delta^{18}$O -values, we only use speleothem data with at least 10 $\delta^{18}$O$_{\text{speleo}}$ measurements and two dating points within a 600 yr period in 850-1850CE. For variance analyses, we demand at least 20 and for spectral, correlation, and network analyses 30 $\delta^{18}$O$_{\text{speleo}}$ measurements. For the investigation of spatial correlation patterns by network analysis, the set of speleothems is divided into nine regional clusters (Fig. 2), as explained in detail in Sec. 3.3. We primarily use the chronologies provided by the original authors, but test for the sensitivity to age-modeling choice by considering the age model ensembles (details below in Sec. 3.3).

Within the last millennium, we remain with 15 aragonite and 89 calcite speleothems with 10 or more $\delta^{18}$O samples. Following Comas-Bru et al. (2019), we exclude six speleothems of mixed mineralogy, as the extent, to which the applied conversion is appropriate, is unclear. The $\delta^{18}$O$_{\text{speleo}}$ signal of calcite and aragonite speleothems is converted to its drip-water equivalent ($\delta^{18}$O$_{\text{dw.eq}}$ ) relative to the V-SMOW standard as in Comas-Bru et al. (2019). For calcite, we use the empirically-based fractionation formula of Tremaine et al. (2011)

$$\delta^{18}\text{O}_{\text{dw.eq}} = \delta^{18}\text{O}_{\text{calcite}} - \left( \left( \frac{16.1 \cdot 1000}{T} \right) - 24.6 \right), \tag{1}$$

where $T$ is in K and $\delta^{18}$O in units of ‰. For aragonite, we use the fractionation factor from Grossman and Ku (1986)

$$\delta^{18}\text{O}_{\text{dw.eq}} = \delta^{18}\text{O}_{\text{arag.}} - \left( \left( \frac{18.34 \cdot 1000}{T} \right) - 31.954 \right). \tag{2}$$

Here, temperature values $T$ represent the local cave temperature in units of K. These are often not available. The annual mean temperature on the surface above the cave can, however, serve as a surrogate for local cave air temperatures (Fairchild

and Baker, 2012). Both for aragonite and calcite drip water conversion, we use the simulated annual mean temperatures at the cave location, down-sampled to the temporal resolution of the record. Note that, as a consequence, the conversion changes the time-averaged mean and the variance in our analysis. Finally, the V-PDB to V-SMOW conversion from Coplen et al. (1983) is used.

$$\delta^{18}O_{SMOW} = 1.03092 \cdot \delta^{18}O_{PDB} + 30.92. \tag{3}$$

Whenever we directly compare simulation output values with the speleothem records, e.g. when comparing means, variances, or spectra, we use $\delta^{18}O_{dw.eq}$, accounting for the different mineralogies. The conversion would, however, add an extra source of uncertainty in correlation analyses, as it implicitly builds on transient simulation data. Therefore, we denote the raw values of $\delta^{18}O$ measured directly in the calcite or aragonite matrix by $\delta^{18}O_{speleo}$ and focus on those in the network and correlation analyses.

## 3.2 Statistical tests and time series processing

Speleothems form naturally, and therefore provide irregular time series with reconstructed and uncertain observation time series (Rehfeld and Kurths, 2014). We account for this in our assessment as outlined below. Temperature, precipitation, and isotopic data are extracted from the simulation at cave locations by bi-linear interpolation. Annual mean values for temperature, precipitation, and isotopic composition of precipitation are formed by averaging over all months from April onwards to March of the following year. This is also the time span for which precipitation weighted $\delta^{18}O$ ($\delta^{18}O_{pw}$) values are calculated, all for each simulation individually. This allows examining the dynamic response in the signal. All analyses are conducted using both simulated $\delta^{18}O$ and $\delta^{18}O_{pw}$. Differences in mean are given in $\Delta\delta^{18}O = \delta^{18}O - \delta^{18}O_{dw.eq}$ (model-data difference) and variance ratios in the record's variance divided by the variance of the simulation at the cave location ($Var_{Rec}/Var_{Sim}$). If not explicitly stated otherwise, we always provide 90% confidence intervals by bootstrapping (Efron and Tibshirani, 1986) with 1000 repetitions. To reduce potential bias due to the irregular spatial distribution of cave sites, we use *area-weighting* in spatial mean estimates, where stated. This is done by calculating gridbox-means of all speleothems within a $3.75° \times 2.5°$ gridbox similar to the simulation, which is then area-weighted across latitudes, following Marcott et al. (2013).

While the simulation data is available at monthly basis, the proxy time series are irregular and at annual or lower resolution. Therefore, the simulation data at cave location is *down-sampled* to the record's reconstructed time axis by block averaging. The *power spectral density (PSD)* of a time series over a finite interval of time describes the distribution of power in frequency components of the time series. The integration over all spectral components yields the variance of the time series (Chatfield, 2003). For spectral analyses, the proxy records are interpolated to their mean resolution in a double interpolation and filtering procedure (following Laepple and Huybers, 2014a, b; Rehfeld et al., 2018; Dolman et al., 2020). Spectra of sufficient resolution can then be averaged to a mean spectrum over a certain frequency range (Kunz et al., 2020).

We test the impact of karst storage of drip water (Gelhar and Wilson, 1974; Dee et al., 2015) by applying an additional simplified aquifer recharge model style filter (hereafter *karst filter*). The impulse response of the Green's function depends solely on the transit time $\tau$, as $g(t) = 1/\tau \cdot e^{-t/\tau}$, with $t > 0$. The Green's function is convolved with the simulated input $\delta^{18}O$ or $\delta^{18}O_{pw}$ signal to obtain the simulated karst-filtered signal in the cave. Following Dee et al. (2015) we use a normalization such that $\int g(t)dt = 1$, integrated over the length of the respective time series. For the down-sampled case, we first apply the filter to the annual resolution simulated $\delta^{18}O$, and down-sample to record resolution afterward.

The *correlation* of irregular time series is estimated by Person-correlation adapted for irregular time series (Rehfeld et al., 2011; Rehfeld and Kurths, 2014). The *signal-to-noise ratio* (SNR) is estimated from the estimated cross-correlation $\hat{r}_{ij}$ between two time series $i$ and $j$ by calculating SNR $= \hat{r}_{ij}/(1 - \hat{r}_{ij})$, as described by Fisher et al. (1985). If more than two estimates are available, e.g. at the gridbox level, the median between all possible combinations of cross-correlations between the time series is used. For correlation estimated, we choose a significance level of $\alpha = 0.1$. In balancing the strictness and the expected level of false positives against that of data demands and the available number of samples $N$, the level is appropriate for both paleoclimate archive and model data time series. The p-values for irregular series are estimated based on a t-distribution, with the degrees of freedom estimated from the temporal coverages $R_{x,y}$ and the persistence time $\tau_{x,y}$ as $N_{eff} = \min(\max(R_x/\tau_x, R_y/\tau_y, \text{na.rm=TRUE}), \max(N_x, N_y))$. This is implemented in the R package `nest` (https://github.com/krehfeld/nest, Rehfeld et al., 2011; Rehfeld and Kurths, 2014). In the case of the speleothem records, the estimated effective degrees of freedom range from $N_{eff} = 20$ to $N_{eff} = 470$, and they are generally similar to the length of the records. For the regular time series, p-values are calculated via Pearson's product moment correlation (via the function `cor.test`). We account for age-model sensitivity by calculating cross-correlation estimates for all possible combinations of available age-model ensembles (Comas-Bru et al., 2020). The provided age-models are not a priori ranked by likelihood and are all consistent with the radiometric chronological constraints. The age-model pair that results in the strongest significant absolute correlation estimate ($p < 0.1$) between two records is selected for the *best selection tuning*.

### 3.3 Spatial correlation via network analysis

Networks are practical representations for complex systems with interacting components and can be used to analyze dynamics in the climate system (Tsonis et al., 2006; Tupikina et al., 2014; Rehfeld et al., 2013). Here we use a network with $n$ nodes, where $n$ is the number of SISAL v.2. entities that fulfill the sampling criteria. The speleothem entities are joined in pairs by edges or links, where the $n \cdot (n-1)$ links are formed if the cross-correlations $\hat{r}_{i,j}$ between two speleothem entities $i$ and $j$, are significantly different from zero with a p-value of $p_{i,j}$.

We split the network into eight sub-networks by hierarchial distance-based clustering of the node locations. The cluster that includes all East Asian caves is manually split into two clusters, one for East Asia (all caves above 20°N) and a cluster of South East Asia (all caves below 20°N). With this, we end up with nine clusters as depicted in Fig. 2. Links in the plots (Fig. 8) are visualized if they are stronger than a certain threshold $|r| > r_{5\%}$, where $r_{5\%}$ is minimum correlation strength of the 5% absolute strongest correlations ('fixed link density').

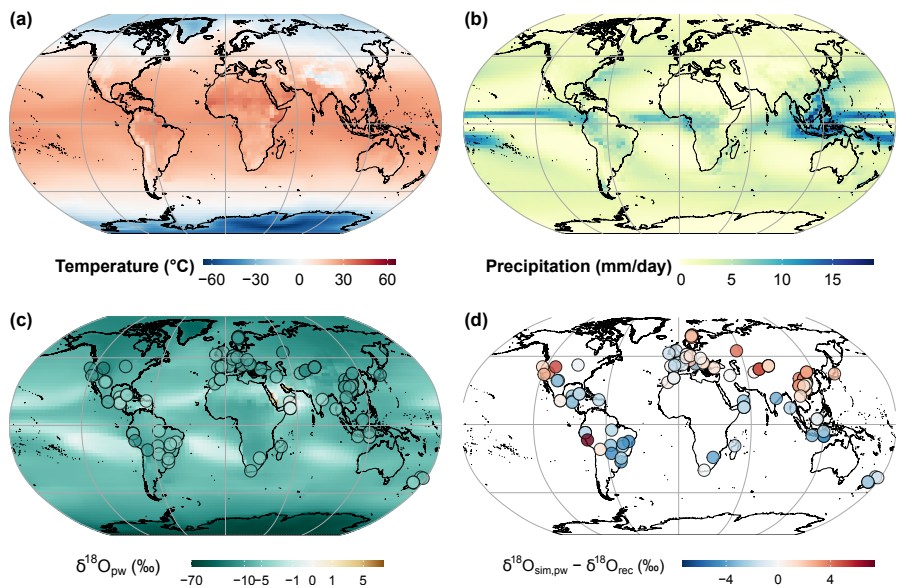

**Figure 3.** Characterization of the mean state of the simulation (LM1): Shown are (a) mean annual surface temperature, (b) precipitation, and (c) $\delta^{18}O_{pw}$, including $\delta^{18}O_{\text{dw.eq}}$ at cave sites in drip water equivalents. Note the logarithmic color scale. Point-wise differences between the mean simulated $\delta^{18}O$ and proxy-based $\delta^{18}O_{\text{dw.eq}}$ (d) show anomalies. Spatially aggregated differences at the global and cluster level for simulations LM1-LM3 are given in supplementary table ST1.

## 4 Results

### 4.1 Assessing model-data differences in time-averaged $\delta^{18}O$

We first compare the mean SISAL v.2.-record $\delta^{18}O_{\text{dw.eq}}$ and iHadCM3 $\delta^{18}O_{pw}$ to assess potential model biases, using the $104$ records with more than 10 $\delta^{18}O_{\text{speleo}}$ measurements within the last millennium. Annual mean temperature and precipitation fields (Fig. 3a,b), and the mean modeled $\delta^{18}O_{pw}$ together with the mean $\delta^{18}O_{\text{dw.eq}}$ in the SISAL records (Fig. 3c) is shown in Fig. 3. Shown and described are the fields and results for LM1, results for the other ensemble members are generally very

similar and given in supplementary table ST1. The major oxygen isotope ratio depletion features as described by Dansgaard (1964), can be distinguished. Modeled values show progressive depletion towards higher latitudes, the interior of continents, and towards regions with high precipitation amounts.

The offsets between modeled and measured $\delta^{18}O_{pw}$ ($\Delta\delta^{18}O = \delta^{18}O_{pw} - \delta^{18}O_{\text{dw.eq}}$) show a heterogeneous pattern (Fig. 3d). Generally, modeled values appear to be more depleted overall than the mean values of speleothem $\delta^{18}O_{\text{dw.eq}}$, except in the NH

extratropics. There are some localized clusters and individual sites with large positive and negative differences. One example is site 38 (eID = 113, Diva cave in Brasil) which is visible as a dark blue dot in Fig. 3d. The $\delta^{18}O_{\text{dw.eq}}$ record shows only slightly depleted $\delta^{18}O$ in calcite ($\delta^{18}O_{\text{dw.eq}} = -2.89\permil$), while the simulation shows much more depletion ($\delta^{18}O_{pw} = -7.68\permil$). This

results in a model-data difference of $\Delta\delta^{18}O = -4.79‰$. The surrounding sites in Brasil are also less depleted as in the simulation. Site 277 (eID = 598, Huagapo cave in Peru) visible as a dark red dot in Fig. 3d, shows a strong depletion in calcite ($\delta^{18}O_{dw.eq} = -13.7‰$) while the simulation is not as strongly depleted ($\delta^{18}O_{pw} = -6.47‰$). This results in a large positive offset of $\Delta\delta^{18}O = 7.33‰$. The cave is located at an altitude of 3850m above sea level, whereas model altitude at the gridbox is close to sea-level. This should explain part of the offset.

At the regional scale, the largest cluster offset can be seen over China and East Asia (c7) $\Delta\delta^{18}O = +2.2‰$ $(-0.18, 4.65,$ 90% confidence interval). However, the most consistent negative difference is visible over neighboring Indonesia (c8) $\Delta\delta^{18}O = -2.95‰$ $(-5.89, -0.02)$. The smallest differences are found in Europe with $\Delta\delta^{18}O = +0.51‰$ $(-1.95, 2.96)$. Overall, the simulated $\delta^{18}O$ is smaller than the $\delta^{18}O_{dw.eq}$ measured in speleothems ($\Delta\delta^{18}O = -0.07‰$ $(-4.31, 4.17)$). The gridbox-level area weighted global mean difference is $-0.02‰$ $(-0.22, 1.00)$ for LM1.

We further explore the impact of site conditions on the model-data offset (Fig. 4). We find a decreasing $\delta^{18}O_{dw.eq}$ towards northern higher latitudes (Fig. 4a), and most notably, a dependency of $\delta^{18}O_{dw.eq}$ on the local mean annual temperature (Fig. 4b). We see more positive offsets in the northern hemisphere and mostly negative offsets in the southern hemispheres (Fig. 4c) as was also distinguishable in the map in Fig. 3d.

The offsets also show a strong influence of temperature (Fig. 4d) and elevation (Fig. 4i), which are both controlling factors during the isotopic fractionation process. The elevation difference between the simulation and the record spans from a 1332m higher elevation in the simulation (eID = 538 in Shenqi cave in China) and 3065m higher elevation in the records (eID = 598 in Huagapo cave in Peru, visible outlier in Fig. 4c,d. Here, the offsets increase with increasing absolute difference (Fig. 4j). The offset shows a weak correlation with precipitation (Fig. 4f), both in the annual mean and for the boreal winter/summer season (see DJF and JJA precipitation in Fig. 4g,h). No relation can be seen with mineralogy, parent rock (Fig. 4e) or cover thickness (Fig. 4k).

## 4.2 Assessing model-data differences in the local variance of $\delta^{18}O$

To analyze how similar the variability of the isotopic signal is in the iHadCM3 climate model and the speleothems, we compare the total variance of the simulation to that of the 92 speleothem records with more than 20 $\delta^{18}O_{speleo}$ measurements over the last millennium. The global distribution of variance ratios between $\delta^{18}O_{dw.eq}$ and down-sampled $\delta^{18}O$ (Fig. 5a) shows overall higher variability in the speleothem records than in the simulation, with local exceptions. This is also corroborated by the density plots of the ratio for both $\delta^{18}O$ and $\delta^{18}O_{pw}$ in Fig. 5b,c. Generally, the observed proxy variance is roughly two times higher than that of the down-sampled simulation $\delta^{18}O_{pw}$ at the cave location (median of the histogram at $1.8$ $(1.4, 2.6)$ in Fig. 5b,c). This is consistent with the predominance of red-shaded variance ratio visualizations in the spatial view indicating $Var_{Rec}/Var_{Sim} > 1$ (Fig. 5a). However, there is a clear impact of averaging on the total variance, as down-sampling results in a variance ratio above unity. Overall, this shows a discrepancy between the variance observed in $\delta^{18}O_{dw.eq}$ and the simulated variance at the cave location over the total time period.

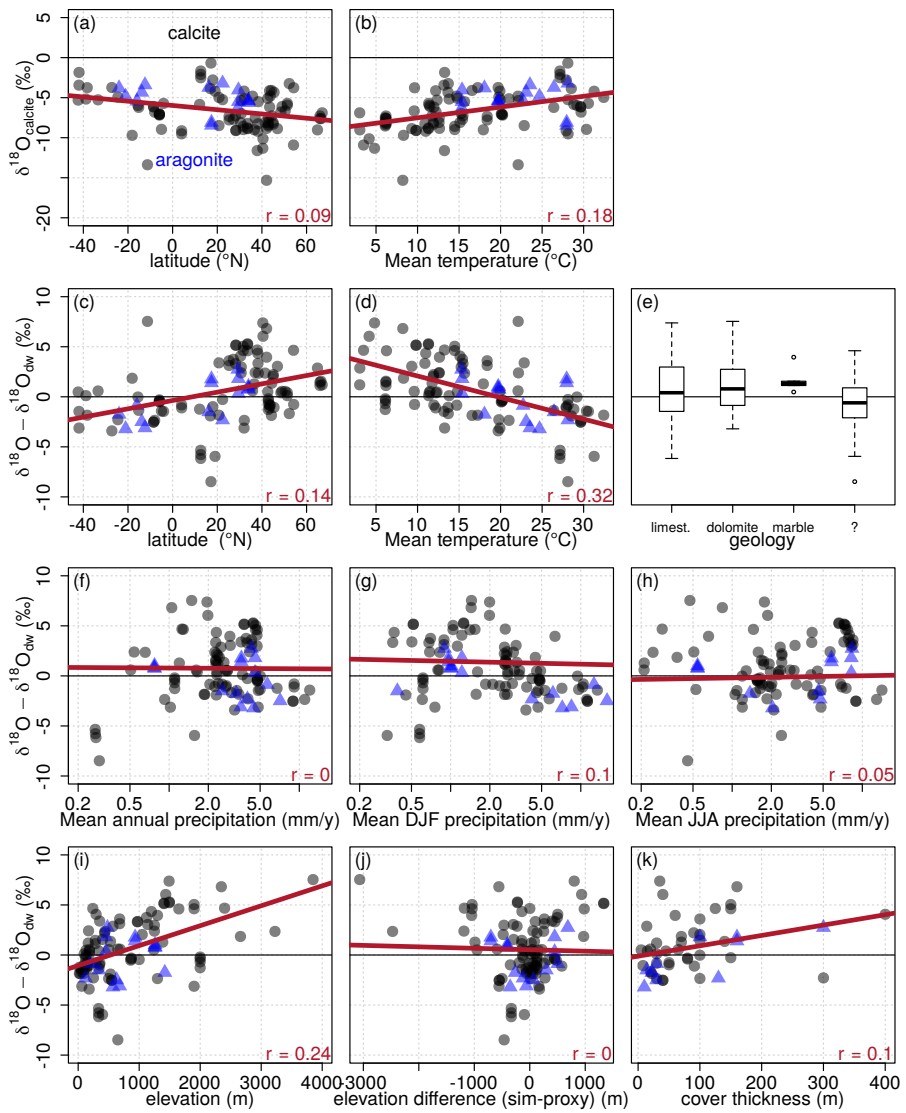

**Figure 4.** Systematic comparison of climate variables from LM1 and cave parameters on $\delta^{18}O_{\mathrm{dw.eq}}$ and the offset $\Delta\delta^{18}O$ to the simulation. Shown are the absolute values of $\delta^{18}O_{\mathrm{dw.eq}}$ against: (a) site latitude, and (b) simulated local annual mean temperature, and the model-data difference against (c) latitude, (d) simulated mean annual temperature, (e) geology surrounding the cave ('?' means unknown geology), (f) mean annual (g) DJF and (h) JJA precipitation amount as well as (i) cave elevation, (j) the elevation difference between the model grid and actual cave, and (k) the overall cover thickness above cave. Symbols denote calcite (black circles) or aragonite (blue triangles) specimens. An unweighted linear regression (red line) is added for illustration, but without consideration of significance.

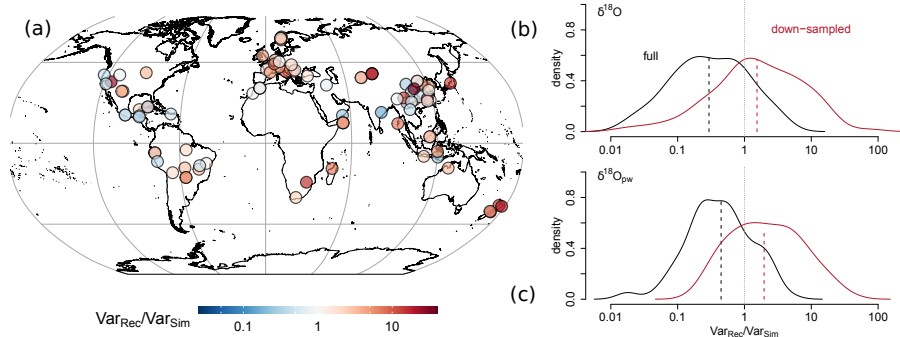

**Figure 5.** (a) Spatial visualization of the site-based dimensionless variance ratio $V_{Rec}/V_{Sim}$, where the simulated $\delta^{18}$O is down-sampled to record resolution, based on LM1. Aggregated density plots of the variance ratio of $\delta^{18}$O (b) and precipitation-weighted $\delta^{18}$O$_{pw}$ (c) for the raw records ('full', black lines) and where the simulation has been down-sampled to the record resolution ('down-sampled', red lines) illustrate the variance loss due to temporal averaging in the archive (uses LM1-3).

The highest variance ratio for down-sampled $\delta^{18}$O$_{pw}$ is found in Jiuxian cave in China (eID 330, with a variance ratio of 49.5), the lowest variance ratio in Dandak cave in India (eID 130, with a variance ratio of 0.2), while neighboring caves show very different variance ratios. As the modeled patterns are fairly smooth, this indicates a large heterogeneity of the speleothem data from the cave environment. We find no strong or significant relationship between variance or variance ratios to any tested climate or cave parameter (SF4 where we show a similar figure to Fig. 4 but for variance ratios).

### 4.3 Assessing $\delta^{18}$O variability at interannual to centennial timescales

We extend the analysis of total variance (Fig. 5) to the time scale dependent variance (Fig. 6) to better explore variability on interannual, decadal, and centennial time scales as compared to the total variance over the last millennium. We set stronger criteria on the speleothem records and only analyze the 85 with more than 30 measurements over the last millennium. The spectra in Fig. 6d give an insight into the variability over different time scales and the representativity of records for reconstruction resolution.

On the left side (Fig. 6a-c), the time series of $\delta^{18}$O$_{dw.eq}$ of eID 240 (Bunker cave, Germany) is depicted (Fig. 6a), together with the simulated $\delta^{18}$O$_{pw}$ at the cave site at different temporal resolutions (Fig. 6b,c), including karst-filtered $\delta^{18}$O$_{pw}$ . Comparing Fig. 6a-c visually, different levels of variance can already be distinguished e.g. between the filtered and unfiltered simulated data. The iHadCM3 $\delta^{18}$O$_{pw}$ spectrum of the yearly resolved signal has similar variance over all frequencies and shows a fairly constant PSD (Fig. 6d). Variance at decadal timescales (i.e. the PSD for higher frequencies) is just as high as the variance on centennial time scales (i.e the PSD for lower frequencies).

After down-sampling to the irregular resolution of the record, the simulated spectrum loses power in the higher frequency range. Comparing for example the time series in Fig. 6c to the spectra in Fig. 6d, the down-sampled spectrum indicates lower

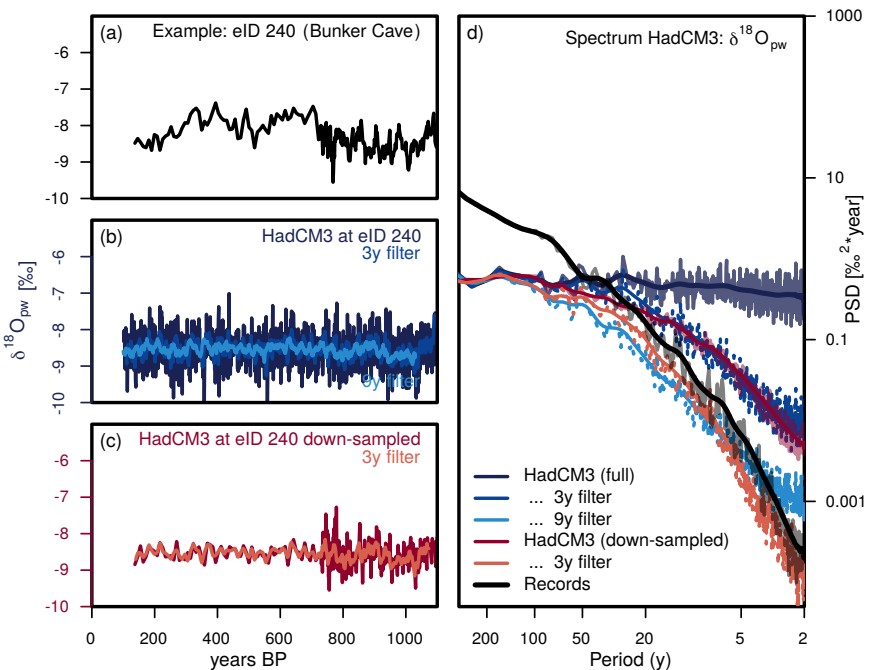

**Figure 6.** Variability on different time scales through comparison of measured $\delta^{18}O_{dw.eq}$ and simulated $\delta^{18}O_{pw}$ time series as well as of their spectra. (a-c) Example time series of eID 240 in Bunker cave (Germany) (Fohlmeister et al., 2012). (a) The measured $\delta^{18}O_{dw.eq}$ in the speleothem, (b) the iHadCM3 simulated $\delta^{18}O_{pw}$ at the cave location with two filters (3 and 9 years) and (c) the simulated $\delta^{18}O_{pw}$ but down-sampled to the same temporal resolution as in (a) with 3 year filter. (d) Power spectral density (PSD) of mean spectra of simulated $\delta^{18}O_{pw}$ at the cave site in yearly resolution (blue), down-sampled to the caves resolution (red) and mean spectrum of the $\delta^{18}O_{dw.eq}$ of the records (black), including the karst-filteres as shown in (a-c). The spectra are area-weighted and averaged over the three simulations (LM1, LM2 and LM3). The colors for the example eID in (a-c) correspond to the colors of the mean spectra over all entities in (d).

variability than the annual resolution spectrum on decadal timescales. On centennial timescales, both spectra display similar variability. Contrasting Fig. 6b to Fig. 6c, this loss in decadal time scale variability is also visible on the time series level.

The proxies' spectra have even fewer frequency components in the high frequency range, due to the lower temporal resolution. They do, however, show a higher PSD at lower frequencies. The records are, therefore, less variable on decadal time scales, and more variable than both the down-sampled and the full resolution simulated $\delta^{18}O_{pw}$ on centennial time scales.

An additional impact of karst processes and storage on the $\delta^{18}O_{pw}$ variability could be expected. To test the impact of this, we apply simple karst filters (see Sec. 3) with increasing filter length and test whether they reduce the spectral mismatch. Filters of different lengths resulted in increasing spectral slopes with increasing transit times. A 3 yr filter for the down-sampled $\delta^{18}O_{pw}$ achieves equivalent variance trends as the record spectrum with less power on decadal timescales. It eventually flattens again for longer timescales, without exceeding the PSD of the unfiltered signal, such that it is less variable than the proxies

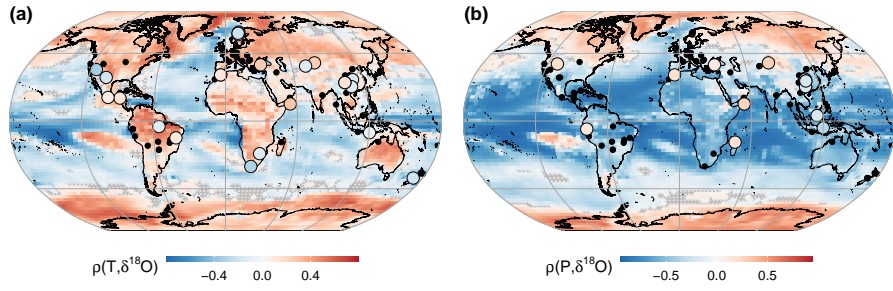

**Figure 7.** Correlation fields of simulated $\delta^{18}O_{pw}$ and the related climate variables surface temperature (a) and precipitation (b) for simulation LM1 ($|c| > 0.2, p < 0.1$). Colored symbols give the correlation between simulated climate variables and the $\delta^{18}O_{dw.eq}$ of the speleothem records. Empty tiles mask non-significant correlations. Black dots show cave locations with non-significant correlations.

340 on longer time scales. A full set of individual spectra (full simulation, down-sampled, record spectrum, and all filters) for all entities used in this analysis can be found in SF5 and SF6.

## 4.4 Climatic drivers of $\delta^{18}O$ variability

To distinguish main important climatic drivers for specific areas for $\delta^{18}O$ both in the simulation and in speleothems, we correlate simulated $\delta^{18}O_{pw}$ with the simulated temperature (Fig. 7a) and the precipitation signal (Fig. 7b) on a gridbox level after

345 temporal down-sampling. Grey (empty) tiles indicate non-significant correlation estimates. The correlation between $\delta^{18}O_{speleo}$ and the climate variable is also shown.

 We see strong correlations of simulated $\delta^{18}O_{pw}$ to simulated temperature at high latitudes as well as over some landmasses (background in Fig. 7). The speleothem signals show positive as well as strong negative correlations. The absolute highest correlation is found for eID 124 in Leviathan cave in the USA ($c = -0.4\ (-0.7, 0.1)$). In the simulation, this correlation is

350 locally positive, which indicates that the simulated temperature is a positive $\delta^{18}O_{pw}$ driver in the general area in the model. The correlation of the simulated climate and the record's $\delta^{18}O_{speleo}$ is, however, negative.

 The correlation between the simulated precipitation and $\delta^{18}O_{pw}$ is especially strong in the tropics. We find the highest absolute correlation for eID 523 in Gempa Bumi cave in Indonesia ($c = -0.5\ (-0.7, -0.1)$). Here, the background also shows a negative correlation.

355 Comparing the two proposed climatic drivers of $\delta^{18}O_{pw}$ variability, we observe that the correlations to temperature are higher in the higher latitudes, while correlations to the precipitation appear more important in the tropics. A fairly clear zonal structure of correlations between the climate and oxygen isotope ratio fields is visible in the model. However, only few of the records show a significant correlation ($p < 0.1$). We find 18, 15 and 22 significant correlations from 85 entities for temperature for the three LM ensemble runs respectively. 44 of these are from entities that show significance in 2 of the 3 LM runs. For

360 precipitation, we find 14, 7, and 10 significant correlations where 54 entities are significant in at least 2 of the 3 LM runs. No clear climatic driver can, therefore, be extracted alone from record correlation results. Fewer records show significant

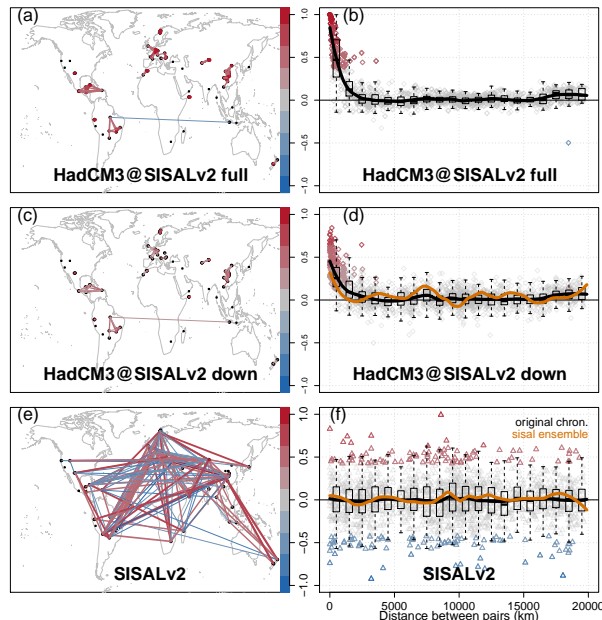

**Figure 8.** Network spanned by the $5\%$ strongest absolute correlations of simulated iHadCM3 LM1 $\delta^{18}O_{pw}$ at the SISAL cave sites ( a) full i.e. annual resolution, c) down-sampled). All model-based between-site-correlations are shown in the distance-binned boxplot (b,d). Network visualizations (e) and distance-binned boxplot (f) of the cross-correlations between SISAL site $\delta^{18}O_{speleo}$ for the original age models. The color values indicate the $5\%$ strongest correlations in network and boxplot. The LOESS smoother (span = 0.2) in the boxplots indicate the correlation for the original chronology (black) as well as the absolute highest correlation through selection of age-models (orange).

correlation to both climate variables. The direct correlation of the time series of the simulated and proxy-based $\delta^{18}O$ results in only 19, 17, and 19 significant correlations from 85, i.e. at around $20\%$ of the sites. Here, 45 entities show significant correlations in 2 or 3 of the LM runs.

## 4.5 Similarity measures and network analysis

Computing all statistical similarity between the $\delta^{18}O_{speleo}$ signals within a cave ('site-level-correlation') or across nearby caves (regional or gridbox-level correlation) yields a measure of representativity useful for model comparison and uncertainty assessment. The networks in Fig. 8 are based on the simulated signal (annual resolution and down-sampled) $\delta^{18}O_{pw}$ (Fig. 8a-d) and for $\delta^{18}O_{speleo}$ (Fig. 8e,f) for 85 entities.

Network links are based on the $5\%$ highest absolute correlations. The highest correlations are found at close proximity for the models (Fig. 8a-d), whereas links across a wide range of distance can be seen in the proxy data (Fig. 8e,f). High local correlations for the model data can be expected, as the simulated $\delta^{18}O_{pw}$ within one cave will be the same, and only differs on a temporal scale after the adjustment to the entity's temporal resolution (down-sampling). The mean absolute correlation for the $5\%$ strongest significant links in Fig. 8c) is $c = 0.42$ $(0.41, 0.43)$. Comparing the down-sampled distance-to-correlation

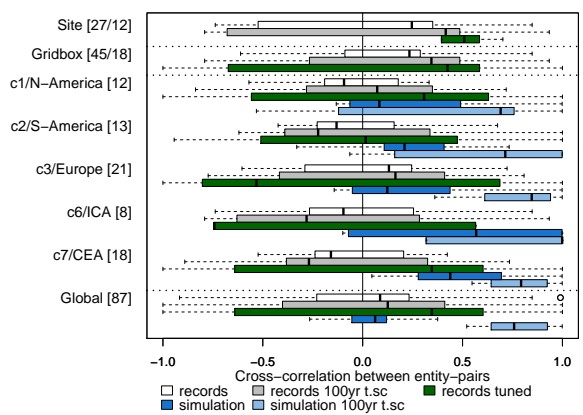

**Figure 9.** Cross-correlation on site, gridboxes, clusters and global scale for speleothem records and the locally interpolated model output for $\delta^{18}O_{pw}$. 12 (18) sites (gridboxes) contain more than one speleothem entity with a total of 27 (45). At each aggregation level the correlation estimates between all entities is shown for $\delta^{18}O_{speleo}$ (white bars), and the down-sampled model output $\delta^{18}O_{pw}$ of LM1-3 at cave locations (blue bars). Different temporal scales (original resolution and 100 yr-timescale (t.sc)) are compared as well as the age-model ensemble that gives the highest absolute correlation (dark green bars). Clusters are indicated with the number of speleothem entities in brackets, where c4, c5, c8, and c9 are not included because they contain too few entities. c6/ICA is the India and Central Asia cluster, c7/CEA is the China and Eastern Asia cluster.

plot (Fig. 8d) to that on annual resolution analysis(Fig. 8b), an additional scattering of correlation estimates at longer than 2000 km distance is visible.

The network of $\delta^{18}O_{speleo}$ does not display large-scale spatial patterns and no observable relationship between correlation and distance. The mean absolute correlation for the $5\%$ strongest significant links shown in Fig. 8e is $c = 0.52$ $(0.52, 0.53)$. Computing the networks based on the ensemble age models, and selecting the age models that maximize the absolute correlation between sites, amplifies both positive and negative correlation estimates but does not change the correlation-to-distance relationship. The sensitivity test performed on the simulated down-sampled $\delta^{18}O_{pw}$ still shows strong yet weaker correlation estimates at short distances. Comparing the results for simulated $\delta^{18}O_{pw}$ (Fig. 8a-d), and $\delta^{18}O_{speleo}$ (Fig. 8e,f), we obtain low correlation estimates at the local scale.

We can also investigate relationships using regional networks. For this, we look at correlation on different spatial levels and separate the network analysis from Fig. 8 by sites, gridboxes, and clusters. Cluster c4, c8, and c9 contain less than four entities and are excluded. We check for representativity on different time scales of record resolution (white, and dark blue) and a 100 yr Gaussian smoothing filter (grey and light blue) and on different spatial scales using boxplots (Fig. 9).

At the site level, we find 27 entities within 12 sites that contain at least 2 entities. The median correlation of these 28 pairs is $c_{raw} = 0.25$ $(-0.17, 0.33)$. On a 100 yr timescale, this increases to $c_{100} = 0.42$ $(-0.46, 0.49))$. The SNR gives a measure of the relative importance of non-climatic overprints on the proxy signal. We obtain a local SNR estimate of $0.5$ $(0.4, 1.1)$. On the

gridbox level (45 entities in 18 gridboxes), we find a median correlation of $c_{raw} = 0.23$ $(0.2, 0.25)$ (100 yr timescale: $c_{100} = 0.34$ $(0.27, 0.47)$). As on spatial resolutions below gridbox level, correlations between simulated $\delta^{18}O_{pw}$ are not meaningful, which is why the analysis in Fig. 9 shows only the correlations between record $\delta^{18}O_{speleo}$ and not those of the simulated $\delta^{18}O_{pw}$ on a site and gridbox level.

For regional clusters, the correlation between proxies shows positive and negative median values. In the simulation, the median values are always positive. For clusters containing more than ten records detailed correlation maps including correlation matrices are depicted for Europe (SF7), China and Eastern Asia (SF8), South America (SF9), and North America (SF10) where very red maps and matrices can be found, indicating mostly positive correlations for the down-sampled simulation when compared to more blue ones in the records, indicating that also negative correlations are present. On the global scale, the median correlation between all records is slightly positive ($c_{raw} = 0.1$ $(-0.09, 0.11)$, 100 yr timescale: $c_{100} = 0.13$ $(-0.14, 0.17)$), whereas for the simulation this median is positive ($c_{raw} = 0.06$ $(0.06, 0.06)$) and strongly enhanced at centennial timescale to $c_{100} = 0.76$ $(0.73, 0.81)$.

By the selection of the age-model that maximizes the absolute correlation, we obtain a significant positive correlation at site level and a stronger significantly positive correlation at gridbox level. A detailed table with median correlations and SNR using the original chronologies as well as using the described age-model selection on different spatial levels is shown in ST2. Calculating correlations for different age-model ensembles was only done for the 69 entities, where both age-model ensembles were available (U/Th-dated entities) in Comas-Bru et al. (2020), and our strongest criteria were matched.

## 5   Discussion

### 5.1   $\delta^{18}O$ model-data comparison in mean and variance

In our study, we found the last millennium mean iHadCM3-simulated $\delta^{18}O$ to agree well with the mean state of the measured $\delta^{18}O_{dw.eq}$ (Fig. 3). The average unweighted offset of $\Delta\delta^{18}O = 0.1‰$ $(-4.6, 4.4)$ was small compared to the total $\delta^{18}O_{pw}$ and the area weighted standard deviation of $\sigma^2 = 0.78‰^2$ $(0.77, 0.8)$ of the global simulated mean $\delta^{18}O_{pw}$. Measured $\delta^{18}O_{dw.eq}$ followed general isotopic signature patterns as described by Dansgaard (1964). The offsets are more positive in the extratropics of the Northern Hemisphere, which is also shown by their temperature dependency (Fig. 4).

Baker et al. (2019) distinguished between temperature zones of climatic controls on $\delta^{18}O$ in offset analyses on drip water. They find a stronger influence of seasonality of precipitation in warmer climates, highlighting the importance of a karst-recharge model. Here, we also observed a strong temperature dependency reflected in the offset and $\delta^{18}O_{dw.eq}$ over the last millennium, showing the influence of fractionation and other cave internal processes on the $\delta^{18}O$ in drip water (Fig. 4) but also additional fractionation processes of weighting through evaporation before the precipitated water enters the epikarst. The higher offsets on the Northern Hemisphere possibly indicate a stronger influence of the continental effect. Still, from the records alone and with no karst-recharge or evaporation information, we were not able to distinguish specific climatic control regions. This requires a more thorough analysis including monitoring data as well as more simulated variables.

We found no evidence that the variance ratio between record variance and simulated variance is related to the offset between simulation and records (SF4 is similar to Fig. 4, with variance vatios instead of $\Delta\delta^{18}$O). Specifically, there is no correlation between site-level offset and site-level variance ratio (results not shown, $r = 0.1$ $(0.0, 0.3)$, $p = 0.2$). In general, the total variance of the simulated $\delta^{18}O_{pw}$ and of the speleothem isotopic signatures over the last millennium are consistent. Differences in variance can, to some extent, be attributed to the sample resolution of the records, whereas down-sampling of simulated $\delta^{18}O_{pw}$ decreases the variability on decadal time scales. The resolution to which the simulation is temporally aggregated impacts whether the variance in the simulation appears to be larger or smaller than in the records. The variance over the last millennium in the records is, overall $1.8$ $(1.4, 2.6)$ times as high as the simulated down-sampled variance in Fig. 5.

Furthermore, the simulated $\delta^{18}$O time series at the cave sites show less variability on centennial timescales than the time series of the records. This is true even when comparing the same temporal resolutions (time scale dependent variance depicted in SF11 for $\delta^{18}O_{dw.eq}$, yearly-resolution $\delta^{18}$O and down-sampled $\delta^{18}$O). This is in agreement with the findings of Laepple and Huybers (2014b), who compared simulated and reconstructed temperature variability across different timescales and found that the model-data discrepancies increased with time scale, particularly on a regional level.

If we assume that paleoclimate archives record climate variability correctly and that the proxy-climate relationships are not timescale-dependent or transient, discrepancies at the centennial timescale could in part be explained by the models' underestimation of variability, in particular on centennial time scales. However, we find little regional consistency and high heterogeneity in the variance estimates from the speleothem records. These findings point to the influence of karst and cave internal processes on meteoric $\delta^{18}$O or the impact of seasonally filtered data captured by speleothems e.g. through strong evaporation in warm months, which is in agreement with McDermott et al. (2001). Age uncertainties, that are not covered by the age-model ensembles, could also be responsible for the low similarity between isotopic signals of neighboring speleothem entities.

## 5.2 Influence of the karst-filter

By delaying the simulated down-sampled signal through a simplified karst-filter with a transit time of $3$ yr, we obtained matching equivalent power spectra for the simulation and the records. Studies observing cave reaction time in karst systems find increases in drip rate after an increase in precipitation e.g. after days (Riechelmann et al., 2011). More complex tritium measurements show actual transit times of e.g. years for the Bunker cave in Germany (Kluge et al., 2010) to decades in the Villars cave in France (Jean-Baptiste et al., 2019), depending on the karst hydrology. The karst filter effectively reduces the temporal resolution of the record beyond the nominal median of $5.6$ yr (Fig. 6). Such low-pass filtering to model drip water transit times has been used (Wackerbarth et al., 2010; Dee et al., 2015; Lohmann et al., 2013) to produce similar time lags of $2 - 10$ yr, indicating that the best fit mean time lag for our karst filter of $3$ yr (down-sampled) is a realistic estimate for transit times.

We find low interannual to decadal variability in the $\delta^{18}O_{speleo}$ signals recorded by speleothems (Fig. 6). In part, this is likely due to the average resolution of the records, which lies close to these timescales. Furthermore, mixing processes in the

soil and karst could play an important role, where soil $\delta^{18}$O is found to have much lower variability than precipitation $\delta^{18}$O (Tang and Feng, 2001).

On decadal timescales (shorter than 50 yr), the karst filter reduced the resolution-adjusted variance by 34% (20, 43), on longer timescales (longer than 50 yr) by 4.0% (3.3, 4.4) of the non-filtered down-sampled variance. The total filtered and down-sampled variance over the last millennium decreased by 14% (9, 27) of the unfiltered down-sampled variance. Still, this is equivalent to only 29% (23, 38) of the record variance, as the filter only decreases variance on annual to decadal time scales. On centennial time scales the filter has little to no effect, so the record's variance on these time scales is not strongly affected.

## 5.3 Representativity of $\delta^{18}$O at different spatial levels

A clear picture of the relationship between the climatic drivers for the simulation was distinguishable. However, no systematic pattern and few significant correlations were found for the speleothem records (Fig. 7). Accounting for seasonal sensitivity could enhance the number of simulation-to-record correlations of SF12, which shows the selected strongest seasonal correlation. However, this does neither enhance the overall correlation (histogram of correlation distribution using annual down-sampled time series and seasonal down-sampled time series in SF13), nor the SNR (results not shown). Still, the strong influence of seasonality suggests a dependency of $\delta^{18}$O$_{\mathrm{speleo}}$ on certain seasons rather than the annual mean. Supporting this, SF14 shows a correlation map with the strongest seasonal correlation of $\delta^{18}$O$_{\mathrm{speleo}}$ to the simulated climate variables temperature, precipitation, and $\delta^{18}$O in precipitation. Further drip water monitoring studies combined with a comparison to model data output and observation data would help to characterize the seasonality of individual caves and would, therefore, lead to deeper understanding of which climatic signal is captured by speleothems and enhance comparability between different caves.

We found low spatial representativity of individual speleothems for sites, gridboxes, and regions when compared to the simulation (Fig. 8 and Fig. 9). We obtained stronger correlations between entities by selection of the best-matching age-model ensemble for entities where these ensembles were available. This age-model ensemble 'tuning' increased the median of correlations on site and gridbox-level by roughly a factor of 2, while also increasing the SNR by a factor of 3 and 2.5, respectively. However, no improvement could be observed on the cluster and global level. A detailed table of correlations and SNRs using the original chronologies as well as using the age-model ensemble selection that gives the highest absolute correlation is showed in the supplement table ST2. Testing other 'tuning' options, such as the consideration of only the 50% of the records at closest proximity within a cluster, or the 50% with the smallest mean offset showed no improvement (boxplot similar to Fig. 9 for the other selection criteria in SF15). We also found no correlation between the total variance and the number of significant links in the network ($c = -0.02(-0.23, 0.19), p = 0.8$). Testing for age-model sensitivity and analyzing the resulting 'tuning' for the down-sampled simulated $\delta^{18}$O in Fig. 8, however, yielded that the method is useful but better selection criteria are needed.

Examining climatic modes such as ENSO, NAO, and others (as shown in SF18 for LM1), which modulate hydroclimate variability across spatio-temporal scales, may provide additional help in the interpretation of the climatic drivers (e.g., following the recent example of Midhun et al., 2021). They found that changes in modeled climatic mode strengths lead to small changes in $\delta^{18}$O$_{\mathrm{speleo}}$. The methods applied, especially regarding teleconnections could provide deeper insight into climatic controls

on speleothem isotopic signals. In particular Midhun et al. (2021) point out the potential to use of speleothem networks in the reconstruction of climatic modes.

A strong between-site variability has been attributed to controls of regional atmospheric circulation according to Lachniet (2009). We also find a strong heterogeneity in the recorded variance of $\delta^{18}O$ at the gridbox and cluster levels. In part, this can be due to heterogeneous temporal resolution, but could also be influenced by non-climatic environmental overprints on the $\delta^{18}O$ signal up to the centennial scale. This could be investigated by comparing the $\delta^{18}O$ and the $\delta^{13}C$ signal recorded within the cave to vegetation, climate, and landscape evolution archives in the region. However, representativity tests across Western Europe noted coherent $\delta^{18}O_{\text{speleo}}$ trends on glacial-interglacial time scales, where trends are less clearly expressed during the Holocene (Lechleitner et al., 2018). Therefore, this study could be extended to longer time scales, when longer transient isotope-enabled simulations become available.

## 5.4 Limitations

Simulated isotope variability is primarily dictated by the model's climatology and the complexity of its dynamics and hydrological cycle. We use a three-member initial-condition ensemble from a single iGCM in this study. Therefore, all results relate to these iHadCM3 last millennium ensemble runs and the chosen radiative forcings. While solar forcing has little influence on simulated $\delta^{18}O$ , the impact of volcanic forcing is much clearer yet still weak (SF16). In this respect, a more thorough comparison with more simulations is needed in order to estimate the capability of models to simulate variability and to find common biases. However, the establishment of isotope-enabled GCMs requires substantial work for the addition of isotopic tracers and their evaluation, and the computational costs increase. This still inhibits the simulation of large transient ensembles with iGCMs over centennial to millennial and orbital time scales. Nevertheless, the three-member ensemble we provide could also be used to test offline data assimilation methods, as suggested by Dalaiden et al. (2020) or Sjolte et al. (2020). With their precise U/Th-dating, speleothems are a well-suited archive for this method and age uncertainties can be accounted for similar to this study by the available age-model ensembles in Comas-Bru et al. (2020). This might also help to better identify the climate factors that govern the speleothem archiving of $\delta^{18}O$ and its variability.

An uncertainty factor in our study comes from the temperature-dependence of the calcite- and aragonite-to-drip-water conversion. We calculated the adjustment factors using the simulated annual mean temperature at the cave location, sampled to the speleothem's temporal resolution. We take this simulated temperature as a surrogate for the longterm-changes of the inside-cave air temperature. Knowing the actual temperature history of the caves better could strongly reduce the uncertainty, as a bias of $\Delta 1^\circ C$ in the simulated temperature would account for a change in $\delta^{18}O_{\text{dw.eq}}$ of approximately $\Delta 0.2‰$. Following Eq. (1)-(3), a bias of $\Delta 1‰$ in the $\delta^{18}O_{\text{dw.eq}}$ however, accounts for a temperature change of $4.5^\circ C$ for the lowest simulated annual mean cave temperature ($3.1^\circ C$ in Norway), and a change of $5.5^\circ C$ for the highest simulated annual mean cave temperature ($32.5^\circ C$ in the tropics).

This model-data comparison focuses on the comparison between simulated $\delta^{18}O$ and precipitation-weighted $\delta^{18}O_{\text{pw}}$ to the drip-water-converted $\delta^{18}O_{\text{speleo}}$. Especially in the more arid regions, evaporation processes play an important role and $\delta^{18}O_{\text{speleo}}$ might be in better agreement with simulated infiltration-weighted $\delta^{18}O$ or a karst recharge model. Further studies

explicitly addressing evaporative effects might help in the interpretation of the results, for example in the region of South America.

Furthermore, our study focussed solely on $\delta^{18}O$ as one particular proxy for climate and environmental changes and not other geochemical proxies that can be measured on speleothem samples (Kaufmann, 2003; Schwarcz et al., 1976) or a combination of proxies, which have the potential of a more thorough interpretation of a climate signal. A multi-proxy approach, such as in Fohlmeister et al. (2017) or Baker et al. (2017) who also include $\delta^{13}C$ along with $\delta^{18}O$, could offer deeper insights. Many proxies for climate processes, such as $\delta^{13}C$ have not (yet) been implemented in comprehensive GCMs, as it requires a detailed and complex representation of the biology, physics, and ecology and dedicated model development. Therefore, in order to consider the vast majority of models in the evaluation, time series have to be calibrated to climatic and environmental parameters that are explicitly modeled. This would introduce additional uncertainty, that could counteract the added value of considering multiple proxies in the first place.

We have considered a regional to global view on speleothem $\delta^{18}O$ signal. Therefore, influences and processes known for individual cave systems could not be considered. For example, Kluge et al. (2013) account for kinetic fractionation changes over time via clumped isotope measurements, and Jean-Baptiste et al. (2019) were able to extract transit times of dripwater in Villar cave. Considering these and other local factors might give deeper insight into individual speleothem records, but it is difficult to scale quantitatively and systematically. Nevertheless, including monitoring datasets from different caves globally might give deeper insight into the filter and fractionation processes involved, and PSM studies informed by the monitoring and local expertise throughout the database could help in further comparison studies.

## 6 Conclusions

We presented an ensemble of iHadCM3 last millennium simulations and compared the oxygen isotope ratios, temperature, and precipitation variability to oxygen isotope ratio observations from a large speleothem dataset (a subset of SISAL v.2.). Overall, time-mean patterns of oxygen isotope ratio were fairly similar in both. Considering total variance as well as the variability on different time scales, we observed that the effects of resolution adjustment and a convolution karst filter were sufficient to bring simulated and observed $\delta^{18}O$ spectra into good agreement. Still, total variability in the speleothem records is much higher than in the simulation. This supports previous studies that found that climate models currently do not capture appropriate variability on centennial time scales.

However, we find that the climatological and environmental interpretation of $\delta^{18}O_{\mathrm{speleo}}$ is not straightforward. We found low signal-to-noise ratios for the isotopic signatures in the speleothem records, which imply a low spatial representativity of individual entities. Furthermore, while regional climatic signals were distinguished in the simulation, the main climatic drivers for $\delta^{18}O_{\mathrm{speleo}}$ at the regional scale were difficult to isolate. It is difficult to establish the size of the spatial footprint of representativity, the seasonality, and the relevant climatological and environmental parameters for reconstructions. Here, expert knowledge on local cave processes, environmental history, and, in particular, the availability of monitoring data are crucial to aid the interpretation of the climate signal. Inner cave and karst processes, which influence the seasonality of the input signal

above the cave and inside the cave, may need to be taken under consideration. However, monitoring data for evaluation and potential calibration of reconstructions are currently only available for a few sites (e.g. Tremaine et al. (2011)). Furthermore, some parameters, such as transit times, are difficult to measure (Jean-Baptiste et al., 2019).

Proxy system models that account for the cave internal fractionation processes may give a deeper insight into how climate variability is captured in speleothem archives. To gain a deeper understanding of the underlying concepts that influence the capability of speleothems to capture and resolve climate variability and the capability of models to simulate them, further model-data comparison studies are required.

*Code and data availability.* Code to reproduce figures and analyses in this paper are provided at https://github.com/paleovar/iHadCM3LastMill (Bühler and Rehfeld, 2020, (accessed February 19, 2021). Model data is freely available on Pangaea https://doi.org/10.1594/PANGAEA.924795 (Rehfeld et al., 2021) and Zenodo https://doi.org/10.5281/zenodo.4551065 (Rehfeld and Bühler, 2021). The SISAL v.2. database (Comas-Bru et al., 2020) can be downloaded at https://researchdata.reading.ac.uk/256/ (Comas-Bru et al., 2020, (accessed September 10, 2020). We use R for the data analysis (R Core Team, 2020). The main packages are tidyverse (Wickham et al., 2019), ncdf4 (Pierce, 2019), ggplot2 (Wickham, 2016), and raster (Hijmans, 2019). We use the nest R-package (https://github.com/krehfeld/nest (Rehfeld et al., 2011; Rehfeld and Kurths, 2014)) and the PaleoSpec package (github.com/EarthSystemDiagnostics/PaleoSpec (Kunz et al., 2020)).

*Author contributions.* KR and JB conceived this study. KR conducted the model simulations, with the advice of MH and LS. JB, MK and KR prepared the model data, and JB, CR and KR prepared the speleothem data. JB and KR wrote the paper. CR, MK and MH contributed to revisions. All authors approved of the final version of the manuscript.

*Competing interests.* The authors declare that they have no conflict of interest.

*Acknowledgements.* We thank one anonymous referee, Jens Fohlmeister and the editor for insightful comments and helpful suggestions. We thank the SISAL (Speleothem Isotopes Synthesis and Analysis) working group of the Past Global Changes (PAGES) program. We thank PAGES for their support of their activity. We acknowledge Laia Comas-Bru, Denis Scholz, Nils Weitzel, Jean-Philippe Baudouin, Martin Werner, and Eric Wolff for advice and helpful discussions and Beatrice Ellerhoff, Elisa Ziegler and Moritz Adam for helpful comments on text and figures. This work was carried out using the ARCHER UK National Supercomputing Service (http://www.archer.ac.uk). Financial support for this study was given by the Deutsche Forschungsgemeinschaft (DFG, projects 316076679 and 395588486) as well as the German Federal Ministry of Education and Research (BMBF) through the PalMod project.

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
