# Peer review of "Comparison of the oxygen isotope signatures in speleothem records and iHadCM3 model simulations for the last millennium"

_Climate of the Past, 2020_

## Referee Comment (RC1) · Anonymous Referee #1 · 22 Oct 2020

Bühler et al. explore the temporal and spatial variability of speleothem d18O for the past1000 time frame (850-1850) using a global compilation of speleothem data and a 1000-year run with an isotope enabled climate model. The authors briefly investigate the relation of d18O to temperature and precipitation in the model, and compare the modelled temperature and precipitation to speleothem d18O. Next the authors explore the spatial relation between a number of variables such as latitude, annual mean temperature, precipitation, and the mean speleothem d18O. They go on to compare the spectrum of temporal variability in the model to the speleothem data. Finally they investigate the teleconnection patterns in speletothem data and in the model. The main conclusions are that i) high frequency variability is dampened in the speleothem data

due to the hydrological residence time before reaching the cave ii) modelled centennial variability is underestimated iii) teleconnections are hard to find in the speleothem data, while more easily identified in the model iv) low signal-to-noise ratio for the speleothem data due to local processes makes it difficult to interpret.

This study contains a lot of interesting and useful work to better understand speleothem and modelled d18O. However, I find some aspects missing that would motivate some of the things studied in the paper, and more analysis is needed to round off the study. Overall the paper is well-written with minor typographical issues. I hope that the authors will see my comments as a positive contribution, as I think the study has a lot of potential, but requires some more work. In summary, I recommend that the manuscript requires major revisions, but will then most certainly be a valued contribution to the topic.

Major comments. 1) Why even look at teleconnections in the d18O data (Figure 8)? There is no mention of ITCZ variability, monsoon, NAO, or other mechanisms driving large-scale d18O variability. I can understand if the authors want to keep the analysis general, but through the whole paper there is no mention of any of the main climate patterns that could explain the teleconnections in Figure 8. See references listed in comment for L78-L80. This should at the very least be mentioned in the introduction, and included in the discussion. There is a lack in information of how HadCM3 performs when it comes to large-scale patterns, and what the imprint is on d18O. For example, add extra correlation maps in Figure 7 for the most important patterns, which quickly could be done. Correlating the monsoon index (e.g. Vuille et al., 2005) to d18O in precipitation should show a very clear pattern across the region around the Indian Ocean. This is not obvious when looking at grid point correlation of climate fields, because the main driving factor is not local precipitation amount, but down wind recycling of vapour in large-scale organized convection. 2) The authors mention external forcing several times as a driver of variability, but never explains or does any analysis to show how this this is related to climate or d18O. This is of course a big topic (e.g. Swingedouw et al., 2017) and might be beyond the scope of the paper. Please either perform analysis of the impact of forcings or be more careful when making statements about what variability is forced and what is not forced. 3) The authors have three simulations but appear to make very little use of the additional information to be gained from this. While three simulations is not a huge ensemble it still yields much more information on forced versus internal variability than a single simulation. When you perform correlation analysis between speleothem data and simulated d18O, this should be done using the ensemble mean. How similar are the ensemble runs in variability? How is the ensemble setup? There is very little information on this. 4) The study uses a shot gun kind of approach to age-model uncertainties. As I understand the different age-models of individual speleothems are sampled independently when testing the range of possible age-models. But are all age-models really equally likely, for example for neighbouring speleothems that we expect to be correlated? Related to this. When comparing the down-sampled modelled d18O to speleothem data in Figure 8, shouldn't the age-model uncertainties also be included for the model data to make the results truly comparable? For completeness there should be two more tests plotted in Figure 8: i) model data which is not down sampled (include SF7 a) and b) in Figure 8, I suppose?) ii) model data including age-model uncertainties. I think this issue with the comparison of model and speleothem data and differences in teleconnections depending on data treatment should be more emphasized.

Detailed comments.

L5-L12 Briefly say that d18O is a climate proxy before discussing all the implications of sampling etc.

L15 "We evaluate systematically..." change to "We systematically evaluate ..."?

L16 "... and test for the main climate drivers for individual records or regions." change to "... and test for the main climate drivers recorded in d18O for individual records or regions."?

L17-L19 Maybe it is be better (worse) to explain in full sentences (fancy truncated syntax)?

L28 "... natural and human systems ... " maybe change to "... human societies and the environment ... "?

L36 The delta-notation comes in here before defining it or telling what the proxy is good for. Either move the definition up in the manuscript and description of the d18O proxy or call it "the relative abundance of 18-O" until you get to the definition, and then explain briefly that this is a climate proxy. You can't discuss the challenges of the interpretation before telling the basics.

L41 You need to include the simulations with GISS ModelE2-R (Colose et al., 2016) and iCESM1 (Stevenson et al., 2019).

L42 Sjolte et al. (2018) compared the variability of the modelled ECHAM5/MPI-OM d18O to Greenland ice core d18O and used the model to assimilate the ice core data to produce gridded reconstructions. Never compare the proxy data to the model – it's the other way around!

L44 Again: Never compare the proxy data to the model! It's not the observations that are being evaluated.

L56-L61 These are a very important points and is written in almost bullet point-style. Please add more details to make it more comprehensible to non-experts. For example, Laepple and Huybers (2014a) are talking about decadal and longer time scales. Laepple and Huybers (2014b) say that the models are too diffusive which is not the same as saying "too high diffusivity", depending on context. Here, they mean that the energy dissipates too quickly across the spectra of temporal variability, which is not clear in your text. My advice is to spend a bit more space on this part of the introduction and don't mix topics, such as too diffusive models and missing processes and feedbacks in the same sentence, unless linking these things directly together.

L64 Add white space after "climate system".

L78-L80 There is quite some evidence that d18O is not primarily a proxy of neither local temperature nor precipitation, but strongly related to circulation modes, large-scale climate patterns and downwind fractionation. For example, in the North Atlantic region the North Atlantic Oscillation (NAO) is important for d18O variability (Vinther et al., 2010; Sjolte et al., 2011; Deininger et al., 2016), while downwind fractionation connected with the Indian summer monsoon impacts the cave d18O in the region around the Northern Indian Ocean, China and South-East Asia (Vuille et al., 2005; Fleitmann et al., 2007; Pausata er al., 2011; Kurita et al., 2013; Lekshmy et al., 2014; Liu et al., 2014; Sjolte et al., 2014; Zhang and Jin, 2015). I think these factors should be highlighted in the introduction.

L91-L99 Here you mainly list the contents of the paper. Can you make the science questions that you are pursuing more clear? Maybe you are testing the climate controls on the variability in simulated d18O using an isotope enabled climate model and compare this to speleothem d18O in a global dataset? Formulate more like hypothesis testing rather than say what kind of analysis you are doing.

L108 Add white space ". . . 30min . . . "

L115 What is meant by "ice sheet" here? I suppose the model doesn't have an ice sheet model?

L120 ". . . features like latitude effect, amount effect, or the continental effect . . . " this is partly repetition from L118. Why not lump these things together?

L123 So, what are the differences between the three model runs? Initial state of the ocean?

Figure 2, caption. Add white space "600yr".

L153 "600y" add white space, and I believe Clim Past uses "yr" shorthand for year.

L161-169 As I understand you allow any type of age model to be used out of the many options, and you pick the best fit independently for each site/speleothem? What are the criteria for accepting an age model, be sides that it is the best fit? Are there cases where the "best" age model is outside of the uncertainty range of the U/Th dating?

L176 How do you decide on the nine clusters? Is this what you describe L236-L239. Please clarify.

L180 "... 10 or more $\delta$18O sampled." should it be "... 10 or more $\delta$18O samples."? Otherwise please rephrase.

L181 "We exclude six speleothems of mixed mineralogy." Why?

L228 If you chose the highest correlation out of a large ensemble of possible solutions, how do you account for this when determining the significance of the correlation?

L256-L257 If you calculate the regional lapse rate of 18O in the model you can estimate the contribution of the model orography to the d18O biases.

L265 Did you try doing multivarite regression? To know the influence on one parameter you need to isolate it from the other parameters.

L273 Add white space "both in the annual mean andfor ...".

L277-L278 "To analyze ..." please rewrite this sentence more concisely and remember, again, that you are comparing the model to the data.

L311 Add white space "3yr". I see space missing many places before "yr". Please check in general.

Figure 7, caption. Here, "insignificant" should be "non-significant". I assume you use the term "significant" in a statistical sense?

Section 4.4 Did you look at the relation of d18O to climate modes? See comment above to L78-L80. For example, the monsoon index (Vuille et al., 2005) might have a

stronger imprint on d18O in the tropical Indian Ocean than local precipitation amount.

L320 "and the climate variable is shown." Change to "and the climate variable is also shown."?

L325 What about the correlation of LM2 and LM3 to the proxy data? Using the model ensemble could give a clue if the variability is related to forcing.

L332 p < 0.1 is not a strong significance criterion. How many samples are there? And in the first place can we expect much correlation between a single model run and observed climate? Changing the initial conditions of the model run would likely affect these correlations, since this is just one realisation, no?

Figure 8: I found the choice of colours confusing in Figure 8d. The smoothed lines are red and blue in the same shade as the markers for the correlation, which made me think at first that the smoothed lines were for the data marked of similar colours, which doesn't make sense. It's quite a busy plot. Consider making it easier to read by choosing different colours or making an extra subplot.

L396-L398 "In general, ... " I don't follow this sentence. Seems like a leap in topic. How can you say anything about forced variability without analysing it? Also, I believe Jungclaus et al. (2010) are discussing the hemispheric mean temperature, while the speleothem d18O data is temperature, precipitation, evaporation and circulation dependent.

L410 "However, we find little regional consistency ... " couldn't this be due to time scale uncertainties? You find no structure in correlation for the speleothem data in Figure 8, but there could be a correlation/regional climate signal, just as well as there could be no correlation.

L428 "longer than50yr" Spaces!

L428 "by 4% (3, 4)" Upper confidence bounds same as median? Or is this due to the number of significant digits?

L434 "However, no systematic pattern and few significant correlations were found for the speleothem records (Fig. 7)." Again, I'm really not surprised that there is no correlation between a free running simulation and the proxy data. There might be forced common variability between model run and proxies (volcanic, solar), but then you need to check the model and proxy response to forcings.

L464 "We use a three member initial-condition ensemble from a single iGCM in this study." Please describe the model ensemble initiation in the methods section.

L470 " . . . as suggested by Dalaiden et al. (2020)." There are lots of examples of offline data assimilation. Maybe provide a few more? E.g., see references in introduction of Sjolte et al. (2020). Ice core data is synchronized using volcanic markers. Any particular age-model related uncertainties to take into account that might complicate the assimilation of speleothem data?

L483 ". . . such as $\delta13C$ cannot (yet) be implemented in GCMs . . . " It's not that far away (Scholze et al., 2008; Camino-Serrano et al., 2019).

L503 ". . . low signal-to-noise ratios . . . " For the speleothem data?

References

Camino–Serrano, M., Tifafi, M., Balesdent, J., Hatté, C., Peñuelas, J., Cornu, S., & Guenet, B. (2019). Including stable carbon isotopes to evaluate the dynamics of soil carbon in the land–surface model ORCHIDEE. Journal of Advances in Modeling Earth Systems, 11, 3650–3669. https://doi.org/10.1029/2018MS001392

Colose, C. M., LeGrande, A. N., and Vuille, M.: The influence of volcanic eruptions on the climate of tropical South America during the last millennium in an isotope-enabled general circulation model, Clim. Past, 12, 961–979, https://doi.org/10.5194/cp-12-961-2016, 2016.

Deininger, M., Werner, M., and McDermott, F.: North Atlantic Oscillation controls on oxygen and hydrogen isotope gradients in winter precipitation across Europe; implica-

tions for palaeoclimate studies, Clim. Past, 12, 2127–2143, https://doi.org/10.5194/cp-12-2127-2016, 2016.

Fleitmann, D., Burns, S.J., Mangini, A., Mudelsee, M., Kramers, J., Villa, I., Neff, U., Al-Subbary, A.A., Buettner, A., Hippler, D., Matter, A., 2007. Holocene ITCZ and Indian monsoon dynamics recorded in stalagmites from Oman and Yemen (Socotra). Quat. Sci. Rev. 26, 170e188.

Kurita, N. (2013), Water isotopic variability in response to mesoscale convective system over the tropical ocean, J. Geophys. Res. Atmos., 118, doi:10.1002/jgrd.50754.

Lekshmy, P., Midhun, M., Ramesh, R. et al. 18O depletion in monsoon rain relates to large scale organized convection rather than the amount of rainfall. Sci Rep 4, 5661 (2014). https://doi.org/10.1038/srep05661

Liu et al., 2014 Z.Y. Liu, X.Y. Wen, E.C. Brady, B. Otto-Bliesner, G. Yu, H.Y. Lu, H. Cheng, Y.J. Wang, W.P. Zheng, Y.H. Ding, R.L. Edwards, J. Cheng, W. Liu, H. Yang, Chinese cave records and the East Asia Summer Monsoon Quat. Sci. Rev., 83 (2014), pp. 115-128, 10.1016/j.quascirev.2013.10.021

Pausata, F.S.R., Battisti, D.S., Nisancioglu, K.H., Bitz, C.M., JUL 2011. Chinese stalagmite delta O-18 controlled by changes in the Indian monsoon during a simulated Heinrich event. Nat. Geosci. 4 (7), 474e480.

Scholze, M., P. Ciais, and M. Heimann (2008), Modeling terrestrial Biogeochem. Cycles, 22, GB1009, doi:10.1029/2006GB002899.

Sjolte, J., G. Hoffmann, S. J. Johnsen, B. M. Vinther, V. Masson‐Delmotte, and C. Sturm (2011), Modeling the water isotopes in Greenland precipitation 1959–2001 with the meso‐scale model REMO‐iso, J. Geophys. Res., 116, D18105, doi:10.1029/2010JD015287.

Sjolte J, Hoffmann G (2014) Modelling stable water isotopes in monsoon precipitation during the previous interglacial. Quat Sci Rev 85:119–135

Sjolte, J., Sturm, C., Adolphi, F., Vinther, B. M., Werner, M., Lohmann, G., and Muscheler, R.: Solar and volcanic forcing of North Atlantic climate inferred from a process-based reconstruc- tion, Clim. Past, 14, 1179–1194, https://doi.org/10.5194/cp-14-1179-2018, 2018.

Sjolte, J., Adolphi, F., Vinther, B. M., Muscheler, R., Sturm, C., Werner, M., and Lohmann, G.: Seasonal reconstructions coupling ice core data and an isotope-enabled climate model – methodological implications of seasonality, climate modes and selec- tion of proxy data, Clim. Past, 16, 1737–1758, https://doi.org/10.5194/cp-16-1737-2020, 2020.

Stevenson, S., Otto-Bliesner, B. L., Brady, E. C., Nusbaumer, J., Tabor, C., Tomas, R., et al. (2019). Volcanic eruption signatures in the isotope-enabled Last Millennium Ensemble. Paleoceanography and Paleoclimatology, 34, 1534–1552. https://doi.org/10.1029/2019PA003625

Swingedouw, D., Mignot, J., Ortega, P., Khodri, M., Menegoz, M., Cassou, C., and Han-quiez, V.: Impact of explosive volcanic eruptions on the main climate variability modes, Global Planet. Change, 150, 24–45, https://doi.org/10.1016/j.gloplacha.2017.01.006, 2017.

Vinther, B., Jones, P., Briffa, K., Clausen, H., Andersen, K., Dahl-Jensen, D., and Johnsen, S.: Climatic signals in multiple highly resolved sta- ble isotope records from Greenland, Quaternary Sci. Rev., 29, 522–538, https://doi.org/10.1016/j.quascirev.2009.11.002, 2010.

Vuille, M., M. Werner, R. S. Bradley, and F. Keimig (2005), Stable isotopes in precipitation in the Asian monsoon region, J. Geophys. Res., 110, D23108, doi:10.1029/2005JD006022.

Zhang X, Jin L. Association of the Northern Hemisphere circumglobal teleconnection with the Asian summer monsoon during the Holocene in a transient simulation. The

Holocene. 2016;26(2):290-301. doi:10.1177/0959683615608689

---

## Author Comment (AC1) · 6 Nov 2020

**Reply to the reviewers' comments: Comparison of the oxygen isotope signatures in speleothem records and iHadCM3 model simulations for the last millennium (cp-2019-121)**

Janica C. Bühler, Carla Roesch, Moritz Kirschner, Louise Sime, Max D. Holloway, Kira Rehfeld

November 6, 2020

**Summary of changes**

We thank the reviewer for her/his constructive comments and detailed reading. In response to the suggestions by the reviewer we plan to

- include the new plots on the influence of external forcing that the reviewer suggested in the discussion and add the figure to the supplement,
- additionally explore the impact of large scale climate modes on  $\delta^{18}O$  variability and include the results in the discussion,
- update Fig. 8 and SFig. 7 to include more information on age-model uncertainty as suggested by the reviewer,
- revise the text throughout the manuscript to clarify statements,
- enhance the discussion especially regarding all analyses and interpretations that include age-model model ensembles,
- and fix formatting where necessary.

A detailed response to the helpful remarks of the referee is given below.

**1 Reply to the first reviewer**

(Original report cited in italics)

Bühler et al. explore the temporal and spatial variability of speleothem d180 for the past1000 time frame (850-1850) using a global compilation of speleothem data and a 1000-year run with an isotope enabled climate model. The authors briefly investigate the relation

of d180 to temperature and precipitation in the model, and compare the modelled temperature and precipitation to speleothem d180. Next the authors explore the spatial relation between a number of variables such as latitude, annual mean temperature, precipitation, and the mean speleothem d180. They go on to compare the spectrum of temporal variability in the model to the speleothem data. Finally they investigate the teleconnection patterns in speletothem data and in the model. The main conclusions are that i) high frequency variability is dampened in the speleothem data due to the hydrological residence time before reaching the cave ii) modelled centennial variability is underestimated iii) teleconnections are hard to find in the speleothem data, while more easily identified in the model iv) low signal-to-noise ratio for the speleothem data due to local processes makes it difficult to interpret.

This study contains a lot of interesting and useful work to better understand speleothem and modelled d180. However, I find some aspects missing that would motivate some of the things studied in the paper, and more analysis is needed to round off the study. Overall the paper is well-written with minor typographical issues. I hope that the authors will see my comments as a positive contribution, as I think the study has a lot of potential, but requires some more work. In summary, I recommend that the manuscript requires major revisions, but will then most certainly be a valued contribution to the topic. We thank the reviewer for this positive assessment.

**Major comments:**

1) Why even look at teleconnections in the d18O data (Figure 8)? There is no mention of ITCZ variability, monsoon, NAO, or other mechanisms driving large-scale d18O variability. I can understand if the authors want to keep the analysis general, but through the whole paper there is no mention of any of the main climate patterns that could explain the teleconnections in Figure 8. See references listed in comment for L78-L80. This should at the very least be mentioned in the introduction, and included in the discussion. There is a lack in information of how HadCM3 performs when it comes to large-scale patterns, and what the imprint is on d18O. For example, add extra correlation maps in Figure 7 for the most important patterns, which quickly could be done. Correlating the monsoon index (e.g. Vuille et al., 2005) to d18O in precipitation should show a very clear pattern across the region around the Indian Ocean. This is not obvious when looking at grid point correlation of climate fields, because the main driving factor is not local precipitation amount, but down wind recycling of vapour in large-scale organized convection.

We thank the reviewer for this suggestion for additional analyses. We agree that the investigation of modes of variability and the modeled  $\delta^{18}O$  response is very interesting. However, attributing the variability of speleothem  $\delta^{18}O_{speleo}$  to specific modes is a challenge that we would not be able to address in a single manuscript. Also, a full-blown analysis on local drivers is not feasible and appropriate given the resolution of the model. The key idea is to show the general correlation patterns obtained from the data, and that of the model while taking into account the resolution difference. This does indeed not allow us to look into teleconnections or directly attribute to drivers of variability. Instead, it gives an overview over the general global spatial correlation structure, and whether there are similarities between the relationships obtained from proxy data, and from model simulations. We identify quite some gaps that require taking into account more proxy-related uncertainties and processes. Investigating to what extent large scale modes of variability impact  $\delta^{18}O$  variability would of course be very interesting, and while an exhaustive investigation is beyond the scope here, we plan to explore the potential for it. We started to initially test the impact of some modes (NAO-index, ENSO-index, monsoon-index) and plan to extend the discussion to include some of these new insights.

2) The authors mention external forcing several times as a driver of variability, but never explains or does any analysis to show how this this is related to climate or d180. This is of course a big topic (e.g. Swingedouw et al., 2017) and might be beyond the scope of the paper. Please either perform analysis of the impact of forcings or be more careful when making statements about what variability is forced and what is not forced.

A very interesting point, indeed. We will re-check our statements to ensure that forced and internal variability are appropriately distinguished. We have previously correlated the solar irradiance time series used for the forcing with  $\delta^{18}O$  variability at the annual scale without seeing strong impacts. As visible in Fig. 1 below, this yields hardly any regions with correlation coefficients clearly distinct from zero. Volcanic forcing, on the other hand shows a clearer imprint on  $\delta^{18}O$  variability. We will add a new figure to the supplement to underline these statements. These aspects are not explicitly discussed in the paper and we will amend the discussion to include them. Fig. 1 below illustrates the correlation map between a-c) volcanic forcing to ensemble mean temperature, precipitation and  $\delta^{18}O$  changes, and in d-f) solar forcing to the climate variables. Especially for temperature we see a clear climatic influence by volcanic forcing, which is also visible in the timeseries of GMST. Precipitation and its isotopic composition however shows only a weak and non-uniform influence of volcanic forcing. Generally, the influence of solar forcing on all climate variables is very weak. The area-weighted mean correlation to solar forcing to the isotopic composition of precipitation is -0.01 (-0.04, 0.06 90% confidence interval) and -0.08 (-0.18, 0.00) for volcanic forcing. We conclude that the external forcing has little influence on the  $\delta^{18}O$  signature in the simulation. We will include this figure in the supplement of the manuscript and incorporate these points in the discussion.

3) The authors have three simulations but appear to make very little use of the additional information to be gained from this. While three simulations is not a huge ensemble it still yields much more information on forced versus internal variability than a single

Figure 1: Correlation estimate fields of a) ensemble mean simulated temperature, b) precipitation, and c)  $\delta^{18}O$  to volcanic forcing, and the same ensemble mean climate variables to solar forcing d-f). Empty tiles mask gridboxes with p > 0.1. The area-weighted average correlation estimates with volcanic forcing are  $\rho(T, \text{volc}) = -0.34$  (-0.48, -0.11),  $\rho(P, \text{volc}) = -0.04$  (-0.15, 0.05),  $\rho(\delta^{18}O, \text{volc}) = -0.08$  (-0.18, 0.00). The area-weighted average correlation estimates with solar forcing are  $\rho(T, \text{sol}) = 0.01$  (-0.004, 0.03),  $\rho(P, \text{sol}) = 0.003$  (-0.009, 0.024),  $\rho(\delta^{18}O, \text{sol}) = -0.012$  (-0.035, 0.056). The correlation estimates are calculated with 989 degrees of freedom.

simulation. When you perform correlation analysis between speleothem data and simulated d180, this should be done using the ensemble mean. How similar are the ensemble runs in variability? How is the ensemble setup? There is very little information on this.

Thank you for this suggestion. Following them we will extend the description of the ensemble. The data, and its description, was uploaded to Pangaea prior to submission but apparently there is a backlog and they are not available yet. The three ensemble members were initialized from different years of the same spinup simulation. We will add this in the method section. We do not, however think that using the ensemble mean is necessarily appropriate in the assessment of variability changes, as this would amplify the forced response and dampen the dynamic part in the signal. The relative role of natural forcing is not the main focus of this paper. We will assess the degree of correlation between the  $\delta^{18}O$  fields that is due to the common forcing. The insight from this analysis will also be added to the discussion.

4) The study uses a shot gun kind of approach to age-model uncertainties. As I understand the different age-models of individual speleothems are sampled independently when testing the range of possible age-models. But are all age-models really equally likely, for example for neighbouring speleothems that we expect to be correlated? Related to this. The new age-models provided with the SISAL chronology (Comas-Bru, Rehfeld, Roesch et al., ESSD 2020) are not ranked by likelihood. All of them are consistent with the radiometric chronological constraints. Therefore we indeed consider all of them in the correlation analyses. In all other analyses we use the corresponding original age models. We make this clearer in the method section.

When comparing the down-sampled modelled d18O to speleothem data in Figure 8, shouldn't the age-model uncertainties also be included for the model data to make the results truly comparable? For completeness there should be two more tests plotted in Figure 8: i) model data which is not down sampled (include SF7 a) and b) in Figure 8, I suppose?) ii) model data including age-model uncertainties. I think this issue with the comparison of model and speleothem data and differences in teleconnections depending on data treatment should be more emphasized.

We absolutely agree with the reviewer that adding this information makes Fig. 8 more informative. We will update Fig. 8 accordingly. It will include the record data, the simulated data and the downsampled-data, including the age-model uncertainty testing. We will also adjust the color scheme, as suggested in the detailed comments.

**Detailed Comments**

L5-L12 Briefly say that d180 is a climate proxy before discussing all the implications of sampling etc.

We agree that adding this general statement prior to the specific impacts improves readability. We adjusted the text as follows:

'The oxygen isotopic ratio  $\delta^{18}O$ , a proxy for many different climate variables, is routinely measured in speleothem samples at decadal or higher resolution and single specimens can cover full Glacial-Interglacial cycles.'

L15 "We evaluate systematically. . . " change to "We systematically evaluate . . . "? Done.

L16 ". . . and test for the main climate drivers for individual records or regions." change to ". . . and test for the main climate drivers recorded in d180 for individual records or regions."?

**Done.**

L17-L19 Maybe it is be better (worse) to explain in full sentences (fancy truncated syntax)?

We liked the short syntax. Nevertheless, we adjusted the text as follows: 'However, using robust filters and spectral analysis, we show that the observed proxy-based variability of  $\delta^{18}O$  is lower than simulated by iHadCM3 on decadal, and higher on centennial timescales.' L28 "... natural and human systems ... "maybe change to "... human societies and the environment ... "? We adjusted the text as follows:

The impacts of a changing climate have been observed over the last century (IPCC, 2013) and indicate a strong sensitivity of **human societies and natural systems** to changes in climate.

L36 The delta-notation comes in here before defining it or telling what the proxy is good for. Either move the definition up in the manuscript and description of the d180 proxy or call it "the relative abundance of 18-O" until you get to the definition, and then explain briefly that this is a climate proxy. You can't discuss the challenges of the interpretation before telling the basics. Thank you for pointing this out. We adjusted the section as follows:

'... Therefore, for model evaluation on longer than centennial time scales, we have to rely on evidence from paleoclimate archives, such as trees, ice cores, foraminifera from marine sediment cores, or speleothems. The abundance of the heavy oxygen isotope 18O, further denoted as  $\delta^{18}O$ , is a proxy for many climate variables and can be measured on these, and quite a few other paleoclimate archives with high precision (Schmidt et al., 2014)...'

The formal definition then follows, starting in L63.

L41 You need to include the simulations with GISS ModelE2-R (Colose et al., 2016) and iCESM1 (Stevenson et al., 2019).

Thank you for pointing this out. We now include these studies in the literature review.

L42 Sjolte et al. (2018) compared the variability of the modelled ECHAM5/MPI-OM d180 to Greenland ice core d180 and used the model to assimilate the ice core data to produce gridded reconstructions. Never compare the proxy data to the model – it's the other way around!

L44 Again: Never compare the proxy data to the model! It's not the observations that are being evaluated. Thank you for the clarification. We will revise the manuscript for the expression and change the sentence stucture accordingly. The lines mentioned here were changed to:

'Few other transient model-data comparison studies focused on  $\delta^{18}O$  (e.g., Wackerbarth, 2012; Dee et al., 2015; Parker et al., 2020). For example, Sjolte et al. (2018) compared the variability of the simulated ECHAM5/MPI-OM  $\delta^{18}O$  to Greenland ice cores over the last millennium to assimilate the ice core data to produce gridded reconstructions. They were able to differentiate between solar and volcanic forcing effects from their reconstructions. On orbital timescales (150,000 yr), Caley et al. (2014) compared a transient isotope-enabled simulation with the model of intermediate complexity iLOVECLIM to speleothem records from South East Asia. They found model-data similarity for the broad temporal trends, but differences at shorter timescales, highlighting the role of seasonality.'

L56-L61 These are a very important points and is written in almost bullet point-style. Please add more details to make it more comprehensible to non-experts. For example, Laepple and Huybers (2014a) are talking about decadal and longer time scales. Laepple and Huybers (2014b) say that the models are too diffusive which is not the same as saying "too high diffusivity", depending on context. Here, they mean that the energy dissipates too quickly across the spectra of temporal variability, which is not clear in your text. My advice is to spend a bit more space on this part of the introduction and don't mix topics, such as too diffusive models and missing processes and feedbacks in the same sentence, unless linking these things directly together.

We agree that these are very important points. Therefore, thank you for pointing out that the discussion should be extended. We will grant the section more space in the revised version.

L64 Add white space after "climate system". Done

L78-L80 There is quite some evidence that d18O is not primarily a proxy of neither local temperature nor precipitation, but strongly related to circulation modes, large-scale climate patterns and downwind fractionation. For example, in the North Atlantic region the North Atlantic Oscillation (NAO) is important for d18O variability (Vinther et al., 2010; Sjolte et al., 2011; Deininger et al., 2016), while downwind fractionation connected with the Indian summer monsoon impacts the cave d18O in the region around the Northern Indian Ocean, China and South-East Asia (Vuille et al., 2005; Fleitmann et al., 2007; Pausata er al., 2011; Kurita et al., 2013; Lekshmy et al., 2014; Liu et al., 2014; Sjolte et al., 2014; Zhang and Jin, 2015). I think these factors should be highlighted in the introduction. Thank you for pointing this out. We adjusted the section as follows:

'...  $\delta^{18}O$  can be regarded as a proxy for example for surface temperature variations in higher latitudes, and precipitation amount in the tropics (Dansgaard, 1964), overlayed with distinct observable signatures of source water evaporation, transportation over longer distances (Bradley, 1999; Dansgaard, 1964), and large scale-climate patterns of circulation such as e.g. the North Atlantic Oscillation (NAO) (e.g. Vinther et al., 2010) or the El-Niño Southern Oscillation (ENSO) (Tindall et al., 2009). These signatures  $\delta^{18}O$  in precipitation may also visible in speleothem records, including additionally a fractionation process involved in the calcite formation, which is primarily temperature-dependent (Urey, 1948; McCrea, 1950)...' L91-L99 Here you mainly list the contents of the paper. Can you make the science questions that you are pursuing more clear? Maybe you are testing the climate controls on the variability in simulated d180 using an isotope enabled climate model and compare this to speleothem d180 in a global dataset? Formulate more like hypothesis testing rather than say what kind of analysis you are doing.

We plan to restructure the paragraph such that it reads:

Here, we present three new last millennium isotope-enabled simulations from the iGCM version 3 of the Hadley Model (iHadCM3) and test how similar the  $\delta^{18}O$  variations in iHadCM3 and speleothem records are (Sec. 4.1). A characterization of the datasets and relevant forcing can be found in Fig. 1. The robustness of the findings and methods are evaluated over the last millennium, for which a large number of high-resolution proxy datasets from the SISAL v.2. database (Comas-Bru et al., 2020) are available.

Our key question are: i) how similar are the modeled  $\delta^{18}O$  signatures to the speleothem records especially regarding variability, ii) can we distinguish main drivers for these signatures, and iii) how representative are the speleothem records for their region. To address these questions, we explore these similarities on both spatial and temporal scales, to distinguish patterns of the mean state (Sec. 4.1), the variability (Sec. 4.2 and Sec. 4.3), and the spatial representativity of speleothem climate records (Sec. 4.4 and Sec. 4.5). We examine the simulation's capability to simulate and the records' capability to capture variability on different time scales to improve our understanding of processes and uncertainties of both.

L108 Add white space ". . . 30min . . . " Done.

L115 What is meant by "ice sheet" here? I suppose the model doesn't have an ice sheet model?

We apologize for this mistake. This statement refers to the sea ice model as documented in Valdes et al. (2017).

'Compared to instrumental observations, the model represents sea surface temperature (SST), sea ice, and ocean heat content well (Gordon et al., 2000).'

L120 ". . . features like latitude effect, amount effect, or the continental effect . . . " this is partly repetition from L118. Why not lump these things together? Thank you for pointing this out, this is truly a repetition. We rearrange the sentence as follows:

'... The model simulates the major isotopic fractionation effects defined by Dansgaard

(1964) (e.g. the latitude effect, the amount effect, and the continental effect) appropriately compared to GNIP data (Zhang et al., 2012). Additionally, a broad agreement in isotopic output with GNIP data in the general spatial distribution can be observed and the **above mentioned** general oxygen isotopic ratio features are **represented** well (Tindall et al., 2009). As such, iHadCM3 captures large scale features of climate and oxygen isotope ratios while remaining computationally efficient for the simulation of timescales such as the last millennium...'

L123 So, what are the differences between the three model runs? Initial state of the ocean?

See response to major comment #3 above.

Figure 2, caption. Add white space "600yr".

L153 "600y" add white space, and I believe Clim Past uses "yr" shorthand for year. Thank you for your detailed reading. We revised the document for white spaces and modified the abbreviation.

L161-169 As I understand you allow any type of age model to be used out of the many options, and you pick the best fit independently for each site/speleothem? What are the criteria for accepting an age model, be sides that it is the best fit? Are there cases where the "best" age model is outside of the uncertainty range of the U/Th dating?

In general we use the authors original chronology for the analysis. In the cross-correlation analysis (esp. Fig 8 and 9) we test, how much correlation estimates change depending on the choice of age models. Here we use all age-model realizations provided by the SISALv2 database. The SISALv2 database defines one median best-fit estimate for each age modeling method, following their selection criteria (see Comas-Bru et al., 2020). As a median age model, it cannot lie outside the range of the ensemble members. Nevertheless, age controls only exist at the radiometric dates, and depending on the dating density and whether reversals were found some age models may of course lie outside individual dates if the dating evidence is contradictory. These age models are method-dependent, consistent with the evidence and free of reversals, as described in Comas-Bru et al. (2020). In the correlation assessment we use all available age model realizations. We will adjust the section. See also our response to major comment # 4.

L176 How do you decide on the nine clusters? Is this what you describe L236-L239. Please clarify.

We decided on eight distance-based clusters and manually added a ninth cluster, to separate a cluster containing all East Asian speleothems above 20°N from those below. We made the link to the latter section, where it is explained in more detail, clearer as follows:

L176: 'For the investigation of spatial correlation patterns by network analysis the set

of speleothems is divided into nine regional clusters (Fig. 2), as explained in detail in Sec.3.3.'

L236: 'We split the network into **eight** sub-networks by hierarchical distance-based clustering of the node locations. The cluster that includes all East Asian caves is manually split into two clusters, one for East Asia (all caves above 20°N) and a cluster of South East Asia (all caves below 20°N). With this, we end up with nine clusters as depicted in Fig. 2....'

L180 '. . . 10 or more d180 sampled.' should it be '. . . 10 or more d180 samples.'? Otherwise please rephrase. Thank you for noticing this typo. We corrected it.

L181 "We exclude six speleothems of mixed mineralogy." Why? This is an excellent question. We require information on the mineralogy of the samples for the conversion to drip water  $\delta^{18}O_{dw.eq}$ . For samples of mixed mineralogy it is unclear to what extent the correction is appropriate. Therefore, and following Comas-Bru et al. (2019), we excluded those speleothems with mixed mineralogy. We will add this clarification to L181 in the manuscript.

L228 If you chose the highest correlation out of a large ensemble of possible solutions, how do you account for this when determining the significance of the correlation? We only choose the highest correlation estimate from significant cross-correlation estimates. If the cross-correlation between two speleothems using a specific pair of age-models is nonsignificant, it is not chosen as a 'best fit'. We will clarify this in the manuscript at the respective section.

L256-L257 If you calculate the regional lapse rate of 180 in the model you can estimate the contribution of the model orography to the d180 biases. L265 Did you try doing multivarite regression? To know the influence on one parameter you need to isolate it from the other parameters.

Thank you for these two interesting suggestions which we will add to the discussion section. We will consider them for a further planned study, where we will look more closely at the biases and the influence of different parameters. In this manuscript, however, we only want to give a first glance at the potential of the analysis.

L273 Add white space "both in the annual mean and for ...". Thank you for noticing. We added the space.

L277-L278 "To analyze . . ." please rewrite this sentence more concisely and remember, again, that you are comparing the model to the data. We adjusted the sentence in accordance with your comment to L42 and clarified the statement as follows:

'..To analyze how similar the variability of the isotopic signal is in the iHadCM3 climate model and in the speleothems, we compare the total variance of the simulation to the speleothem records over the last millennium. The global distribution of variance ratios between  $\delta^{18}O_{dw.eq}$  and  $\delta^{18}O$  (Fig. 2a) shows overall higher variability in the speleothem records than in the simulation, with local exceptions...'

L311 Add white space "3yr". I see space missing many places before "yr". Please check in general.

Sorry for the trouble. We now checked the manuscript thoroughly for this particular mistake.

Figure 7, caption. Here, "insignificant" should be "non-significant". I assume you use the term "significant" in a statistical sense? Adjusted.

Section 4.4 Did you look at the relation of d18O to climate modes? See comment above to L78-L80. For example, the monsoon index (Vuille et al., 2005) might have a stronger imprint on d18O in the tropical Indian Ocean than local precipitation amount. See response to major comment #1.

L320 "and the climate variable is shown." Change to "and the climate variable is also shown."? Done.

L325 What about the correlation of LM2 and LM3 to the proxy data? Using the model ensemble could give a clue if the variability is related to forcing. We agree with the reviewer. See response to major comment # 3. We will adjust this paragraph according to our findings there.

L332 p

---

## Referee Comment (RC2) · Jens Fohlmeister (Referee) · 30 Nov 2020

The authors analysed a climate model simulation covering the last $\sim$ 1000 years mainly with respect to T, prcp, and d18O of prcp. They compared those values with speleothem data, obtained from a recently published data base. They compared the mean, variance, spectral characteristics, correlation pattern of speleothem data and model results on a global scale.

This project is a very ambitious one. To my opinion, the authors predominantly did a really good job. I appreciate the approach of analyzing the data on different aspects and with many, partly sophisticated, methods. The results that the speleothem and

model data seem to do not fit too nicely is very interesting, as it points to the need to further improve both - the performance of climate models and the understanding of proxy data from speleothems. A very first step was already applied in this study: I like a lot that the authors tried to translate the temporal high resolution data of the model results into a speleothem-typical time series (accounting for water residence time in soil and karst, applying speleothem sampling resolution and T-dependent HCO3- → CaCO3 fractionation of stable oxygen isotopes). Nevertheless, I think it is possible to do it even better than that.

For example, and this is my largest point of critics: I am really not sure, if it is the mean annual d18O of precipitation, what speleothem are recording, but what seems to be used here from the model output. Isn't it rather the case that speleothems record the amount weighted d18O values of precipitation? Or in some locations, especially in more arid regions, with low amount of precipitation but elevated T, speleothem d18O reflect more likely the amount weighted d18O of infiltrated water. Thus, evaporation processes are important but not considered here. However, to my understanding climate models do provide those variables. They should have evaporation processes on land included. Wouldn't it be an option to try this variable for analysis? If I remember correctly, Wackerbarth et al., (2012, CP) did such an approach.

Especially, as the models should even account for evaporation-dependent fractionation processes of oxygen isotopes during evaporation, comparing the speleothem results with those of the models should potentially result in better agreement. The d18O values of the remaining, the non-evaporated water, which is finally entering the deeper soil layers and the karst system, would be known from the model. I guess this would be the easiest way (without any need to use cave-site specific insights from cave monitoring studies) to better compare model results and speleothem d18O values in a more comprehensive way. At this point, I don't ask to redo all the analysis with an infiltration weighted mean d18O instead of an annual mean d18O of precipitation, but it would be appropriate to at least include this possibility in the discussion section (e.g. Sec. 5.4)

– and try this in a potential follow-up study.

To my understanding, those evaporation processes could very well explain, why the d18O model data of precipitation tend to underestimate the speleothem-recorded d18O values in warm and more arid regions (see your Figs. 3d and 4d). In those regions, evaporation is very important and influences both – the amount of infiltrating water per month and the evaporation-dependent d18O enrichment of the non-evaporated water. I think that if one would account for that, it would potentially improve your model-data comparison. At least when comparing the mean values of the last 1000 a. But it has maybe even the potential to increase the average variability of the modeled d18O values, compared to your approach using rain water. And it maybe even brings some additional variance on the longer scales into the data. The reason for that could be that d18O of infiltrating water would - in addition to changes in the d18O of precipitation – potentially show a large change due to temperature effects on the amount of evaporation and the d18O of the remaining infiltrating water.

Otherwise, I have only comments of more minor attitude. Please find them listed below (with some repeatedly occurring instances, where I address the advantages of accounting for evaporation and amounted weighted means).

To sum up, I would really like to see this work published pending on an improved manuscript version, where my major concern is accounted for (in which way the authors feel more comfortable with) and the smaller issues from below are considered/discussed.

Best wishes, Jens Fohlmeister

——————————————————————————

L 18: "proxy-based variability of d18O": d18O is a proxy. Thus your phrasing sounds a bit weird. In the context of the text you might mean 'archive-based'?

L19-20: You might should add, that most of the difference on the side of the 'shortterm variability' (<∼20a) comes from smoothing due to soil water residence time and resolution. You showed quite nicely that on the long frequencies, both types of data sets do not agree – whatever you tried.

L 64: a <space> is need after 'system'

L65: I would be more specific here and state that 'The ratio of H218O to H216O in precipitation is an indicator of evaporation temperature, . . .' as it is possible to determine the d18O values in other reservoirs as well. And there other effects are also important.

L83: ' . . . hampered by non-linear growth processes (Dreybrodt 1980).' Is Dreybrodt 1980 the correct reference, as he focussed only on precipitation of CaCO3? Not on d18O variations. Maybe use one of his later studies, e.g., Dreybrodt and Scholz, 2011 or Dreybrodt and Romanov, 2016.

L85: '. . .as well as dating uncertainties'. Please explain, how dating uncertainties shall have an influence on the interpretation of d18O as you state here.

L88: Please correct brackets around the reference in this line.

L116: '. . .freshwater hydrological cycle in the model shows only a slight overestimation in the local evaporation (Pardaens et al., 2003).' According to this statement, there is some evaporation included in the model. So it should be feasible to work with those data, instead to precipitation only (both amount and isotopic signature).

L130: 'Vegetation above the cave has an impact on the source water . . .'. This reads a bit strange. Really on source water? Or rather on the amount of soil water and its d18O, which is coming from some source with a certain isotopic composition? Alternatively, you could write something like: Vegetation above the cave can alter the amount of infiltrating water and its isotopic signature.

L130-131: Here you already state, what potentially can have some strong effect. Thus, I wonder a bit, why you do not have accounted for that stuff in your analysis.

L134-135: This sentence should be changed, as it is not completely correct, if you mean with 'surface' the atmosphere. In addition, the CO2 and Ca2+ charging processes should be mentioned to make this better understandable for the reader. Please consider to rephrase to something like that: 'Infiltrating surface water is charged with soil gas CO2, which concentration is about 1-2 magnitudes larger than that of the atmosphere and enables the carbonic acid driven CaCO3 dissolution of the host rock. The generally lower partial pCO2 pressure conditions in the cave environment compared to that of the soil and epikarst makes the drip water degas ...'

L138-139: This sounds very dramatic. Not from the wording, but from the implications. As it is written here, I hope this to be not true. Otherwise, nobody should trust such speleothem d18O reconstructions.

L180: samples instead of sampled

L202-203: Have you performed this averaging also for d18O in precipitation? From my understanding, of this sentence you do. But I think it would be better to use an amount weighted mean of d18O in precipitation? This is closer to the value really infiltrating into the soil/karst.

Based on that, what about evapotranspiration and changes of this during the modeled 1000 years? Have you accounted for that?

Maybe, I am wrong, but my understanding of those isotope enabled GCMs was, that they have at least an 'evaporation on land' module. This should also account for fractionation effects on the evaporated water, but also for the remaining water, what you are interested in. Would it be an option to use those d18O values instead of that of precipitation (again weighted by the amount of infiltration)? Maybe not for this manuscript, but in any future application.

L219-221: Here you use the first time d18O_(pw). It is not explained here nor somewhere else. What is this?

In addition, I am sorry, but I do not understand exactly, why you are doing this Greens function approach? I get pretty ugly results in terms of mass balance with this, if tau is small (e.g., 1, 2 ,3 years). For example using a tau of 1 year: even after 100 years waiting time, only 58 percent reached the cave. For tau=3 that are 84 %, what reached the cave after 100 years. For larger tau it works better, I admit. But as you use this filter with a residence time of 3 years, I would be happy if you please could explain why you used this way of residence time description. Have you normalized this somehow?

L228-229: Is it possible to rephrase this sentence? I regret to not being able to understand, what you mean by this.

Sec 3.2 and 3.3: Please explicitly state the number of used/available records for those approaches as those are most likely less, than that number mentioned in line 180.

L249: 'Generally, modeled values appear to be more depleted overall than the mean values of speleothem _18Odw.eq . . .'. Wouldn't this be a hint, that evaporation is important and should be accounted for (at least in any potential follow up study)?

L269: 'offsets also show a strong influence of temperature (Fig. 4d)' Only to repeat my statements above: At warmer climates there is more evaporation, which lead to enhanced d18O values of the remaining water compared to that of precipitation. The remaining water soaks finally into the soil and cave. Thus, it would be worth to check if soil water d18O works better than precipitation.

But this will most likely not solve the offset at colder T in the northern Hemisphere. But there, maybe it will work when using the amount weighted d18O of precipitation (or even better with infiltrating water) instead of annual mean d18O. As you brought up the example from Bunker cave. For this cave system and this region it was shown, that summer precipitation barely contributes to infiltrating water (most is gone by evaporation and transpiration), which is able to reach the cave (Riechelmann et al., 2011; Wackerbarth et al., 2010). As summer rain is more enriched in d18O compared to winter precipitation, this would shift the infiltrating water isotopic composition towards

lighter values compared with the heavier annual mean d18O of precipitation. And would thus potentially bring the model results closer to the observed speleothem values.

L298-279: I am sorry but I am confused again. You are writing: 'The global distribution of variance ratios between _18Odw.eq and _18O (Fig. 5a) shows overall higher variability in the speleothem records than in the simulation'. I agree with this observation, but this is somehow in contrast to Fig 5 b and c, where the variance ratio between d18O_dw.eq and d18O is smaller than one. Is it possible that in Fig 5a you are showing the variance ratio of d18O_dw.eq and the already down sampled d18O of the model simulation?

L288: I think the reference to Fig 5 is not correct, as this figure does not give a hint on a 'smoothed model pattern'.

Figure caption 6: 'the simulated _18Opw but down-sampled to the same temporal resolution as in (a) with 3 year filter. 'Just to be sure, as it is not written somewhere: You first applied the three year filter and then did the down sampling, correct?

L332: 'We find 18, 15 and 22 significant correlations from 87 entities. . . ': Out of curiosity, are the records/sites within those observed significant correlations in the three model runs always the same or are they varying? I mean if for example a record from one cave is significant for one run is the same cave record as well significant for the other simulations?

Figure caption 8: indicates instead of indicat.

Figure caption 9: gridboxes instead of gridboxe

L385: 'spatial pattern for the offsets were not distinguishable.' I would slightly disagree here. Isn't it the case that in warmer climates the offset is more negative than in colder climates? See fig. 4d. You even described the low to high latitude trend in the northern hemisphere by yourself in lines 249-250.

L388: 'They find a stronger influence of seasonality of precipitation in warmer climates, highlighting the importance of a karst recharge model' Wouldn't it highlight the fact, that the model d18O values should be calculated as a precipitation weighted mean? Or even as infiltration weighted mean?

L389-390: 'observed a strong temperature dependency reflected in the offset and _18Odw.eq over the last Millennium, showing the influence of fractionation . . .': You claimed, that you accounted for the temperature-dependent isotope fractionation during CaCO3 precipitation to calculate the d18O if the drip-water equivalent. So I think, this reason can be safely excluded.

I would evaluate it more likely that, as evaporation scales with temperature, the d18O values of precipitation have been changed by this process before the water even enters the epikarst zone. Thus, I would like to highlight it again (sorry for repeating myself), that using some infiltration weighted d18O mean is probably a better choice.

L410-411: 'However, we find little regional consistency and high heterogeneity in the variance estimates from the speleothem records'. While there is indeed some high heterogenity in the variance, I wonder, if this could be an argument for a strong influence of cave internal processes? That is a tricky one.

Later you discuss the correlation pattern, and this seems quite convincing that changes happen in the same direction and at about the same time at least at regional scale. Only the magnitude seems to change - as derived from the variance analysis. Wouldn't this rather mean, that each cave/stalagmite strengthen or weaken the initial climate signal, but that it is still contained? As you said, this alteration could happen by soil/karst/cave processes. This way, one could argue about the magnitude of the changes, but not about the variation itself. How do you think about this?

L416-417: 'Studies observing transit time in karst systems find increases in drip rate after an increase in precipitation e.g. after days (Riechelmann et al., 2011).' I suggest to term this Riechelmann et al., (2011) work rather as a study, which investigates the cave

reaction time to precipitation events - not a transit time study. And the cave reaction on heavy rain events is often fast (as observed in other caves as well), but it will not change the transfer time (too much). The residence time can only be found by tritium or other appropriate tracer isotopes (as you correctly describe below.)

L442: I guess, cave monitoring would not only help to compare two caves but also to improve the comparability between a cave and climate model results. You could add this as well. Furthermore, I think that not only monitoring would help, but that (model-based) weighted infiltration values would also help (as written already earlier). Even with your sentence in line 438 you imply so by yourself.

L455: 'but could also be influenced by non-climatic overprints on the _18O signal up to the centennial scale.' I would argue, that nearly all changes in the processes in the cave are climate driven. Unfortunately, sometimes they counter the pure climate imprint and on other locations they amplify it.

L476: 'as a bias of 1C in the simulated temperature would account for a change in _18Odw.eq of approximately 2‰.' If you really take the d18O-T relationship of Tremaine (or any similar) this statement appears to be wrong. Fractionation during CaCO3 precipitation is T-dependent by $\sim$0.25 permil per °C. So this would mean an offset of 1 permil, if the modeled T is wrong by 4 °C. I guess this would only be the case in mountainous regions, were the orography of the model is not close enough to the true altitude of the cave. You mentioned some examples earlier in your manuscript.

L477:'A bias of _1‰ in the _18Odw.eq however, accounts for a temperature change of 0.1_C for the lowest simulated annual mean cave temperature (3.1_C in Norway), and a change of 13.1_C for the highest simulated annual mean cave temperature (32.5_C in the tropics).' This puzzles me now quite a lot. You said in your earlier sentence, that 1°C is 2 permil. This does not fit with your statement in this sentence. Maybe, you mean something different?

---

## Author Comment (AC2) · 16 Dec 2020

**Reply to the reviewers' comments: Comparison of the oxygen isotope signatures in speleothem records and iHadCM3 model simulations for the last millennium (cp-2019-121)**

Janica C. Bühler, Carla Roesch, Moritz Kirschner, Louise Sime, Max D. Holloway, Kira Rehfeld

December 16, 2020

**Summary of changes**

We thank Jens Fohlmeister for his constructive comments and detailed reading. In response to his suggestions we plan to

- enhance the discussion especially including evaporation and resulting changes in the isotopic signatures of infiltrating water,

- carefully revise the method section and improve those sections, where precipitation weighted $\delta^{18}O$ is used and improve the explanation of the karst filter

- explore different choices for the karst-filtering analysis (reversed order of filtering and temporal degradation),

- revise the text throughout the manuscript to clarify statements,

- and fix formatting where necessary.

A detailed response to the helpful remarks of the referee is given below.

**1   Reply to the second reviewer**

(Original report cited in italics)

*The authors analysed a climate model simulation covering the last ∼ 1000 years mainly with respect to T, prcp, and d18O of prcp. They compared those values with speleothem data, obtained from a recently published data base. They compared the mean, variance, spectral characteristics, correlation pattern of speleothem data and model results on a global scale.*

*This project is a very ambitious one. To my opinion, the authors predominantly did a really good job. I appreciate the approach of analyzing the data on different aspects and with many, partly sophisticated, methods. The results that the speleothem and model data seem to do not fit too nicely is very interesting, as it points to the need to further improve both - the performance of climate models and the understanding of proxy data from speleothems. A very first step was already applied in this study: I like a lot that the authors tried to translate the temporal high resolution data of the model results into a speleothem-typical time series (accounting for water residence time in soil and karst, applying speleothem sampling resolution and T-dependent HCO3- → CaCO3 fractionation of stable oxygen isotopes). Nevertheless, I think it is possible to do it even better than that.*

We thank the reviewer for this positive assessment.

*For example, and this is my largest point of critics: I am really not sure, if it is the mean annual d18O of precipitation, what speleothem are recording, but what seems to be used here from the model output. Isn't it rather the case that speleothems record the amount weighted d18O values of precipitation? Or in some locations, especially in more arid regions, with low amount of precipitation but elevated T, speleothem d18O reflect more likely the amount weighted d18O of infiltrated water. Thus, evaporation processes are important but not considered here. However, to my understanding climate models do provide those variables. They should have evaporation processes on land included. Wouldn't it be an option to try this variable for analysis? If I remember correctly, Wackerbarth et al., (2012, CP) did such an approach.*

*Especially, as the models should even account for evaporation-dependent fractionation processes of oxygen isotopes during evaporation, comparing the speleothem results with those of the models should potentially result in better agreement. The d18O values of the remaining, the non-evaporated water, which is finally entering the deeper soil layers and the karst system, would be known from the model. I guess this would be the easiest way (without any need to use cave-site specific insights from cave monitoring studies) to better compare model results and speleothem d18O values in a more comprehensive way. At this point, I don't ask to redo all the analysis with an infiltration weighted mean d18O instead of an annual mean d18O of precipitation, but it would be appropriate to at least include this possibility in the discussion section (e.g. Sec. 5.4) – and try this in a potential follow-up study.*

*To my understanding, those evaporation processes could very well explain, why the d18O model data of precipitation tend to underestimate the speleothem-recorded d18O values in warm and more arid regions (see your Figs. 3d and 4d). In those regions, evaporation is very important and influences both – the amount of infiltrating water per month and the evaporation-dependent d18O enrichment of the non-evaporated water. I think that if one would account for that, it would potentially improve your model-data comparison. At least when comparing the mean values of the last 1000 a. But it has maybe even the potential to increase the average variability of the modeled d18O values, compared to your approach*

*using rain water. And it maybe even brings some additional variance on the longer scales into the data. The reason for that could be that d18O of infiltrating water would - in addition to changes in the d18O of precipitation – potentially show a large change due to temperature effects on the amount of evaporation and the d18O of the remaining infiltrating water.*

We agree with the reviewer, that the precipitation $\delta^{18}O$ signature that effectively gets transmitted to the karst system and the drip sites is dependent on the precipitation amounts. For that reason, we do account for precipitation amount impacts on the signal, by using precipitation-weighted $\delta^{18}O$ values. In most regions there is a good degree of correlation between precipitation-weighted, and unweighted $\delta^{18}O$. We did, however, not make that very clear in the methods. We will rectify this in the revised version.

We also agree on the benefits of using simulated infiltrating water amounts instead of the annual mean of precipitation. To calculate the infiltration rates the proportion of evaporation is needed for calculation. Evaporation processes are of course considered in HadCM3. Due to storage limitations, however, only selected variables were stored during the simulation. This did not include evaporation and all land model variables except for vegetation fractions. Obtaining evaporation could be possible by rerunning the model, or by estimating it through heuristic approaches as in Thornthwaite and Mather (1957) by considering the latent heat flux from the surface. However this would also introduce additional uncertainty into the analysis. In essence, we agree that we could better discuss the potential effects of evaporation changes on the speleothem records. In the minor comments the reviewer points out several occasions, where evaporation might help in the interpretation of the results. Therefore, we will check for the potential effects of evaporation in the results and add further discussion in the revision. The suggested analysis will also be taken into account for a follow-up project where we include a multi-model data comparison and have access to evaporation output from several simulations and we can cross-compare approaches to estimate evapotranspirative contributions.

*Otherwise, I have only comments of more minor attitude. Please find them listed below (with some repeatedly occurring instances, where I address the advantages of accounting for evaporation and amounted weighted means).*

*To sum up, I would really like to see this work published pending on an improved manuscript version, where my major concern is accounted for (in which way the authors feel more comfortable with) and the smaller issues from below are considered/discussed.*

*Best wishes, Jens Fohlmeister*

**Detailed Comments**

*L 18: "proxy-based variability of d18O": d18O is a proxy. Thus your phrasing sounds a bit weird. In the context of the text you might mean 'archive-based'?*

Thank you for spotting this. We will adjust it with your suggestion.

*L19-20: You might should add, that most of the difference on the side of the 'short-term variability' ($<\sim20a$) comes from smoothing due to soil water residence time and resolution. You showed quite nicely that on the long frequencies, both types of data sets do not agree – whatever you tried.*

Thank you for this specification. We adjusted the section as follows: '...Most of this difference can likely be attributed to the records' lower temporal resolution and averaging **or smoothing** processes affecting the $\delta^{18}O$ signal **e.g. through soil water residence times**. ...'

*L 64: a space is need after 'system'*
Done.

*L65: I would be more specific here and state that 'The ratio of H218O to H216O in precipitation is an indicator of evaporation temperature, . . .' as it is possible to determine the d18O values in other reservoirs as well. And there other effects are also important.*

Thanks for this clarification. We will adjust the sentence using your suggestion.

*L83: ' . . . hampered by non-linear growth processes (Dreybrodt 1980).' Is Dreybrodt 1980 the correct reference, as he focussed only on precipitation of CaCO3? Not on d18O variations. Maybe use one of his later studies, e.g., Dreybrodt and Scholz, 2011 or Dreybrodt and Romanov, 2016.*

Thank you for this correction. We will clarify and refer to Dreybrodt and Scholz (2011) instead.

*L85: '. . .as well as dating uncertainties'. Please explain, how dating uncertainties shall have an influence on the interpretation of d18O as you state here.*

"... The climatic interpretation of speleothem $\delta^{18}O$ variations in calcite or aragonite (hereafter $\delta^{18}O_{speleo}$ ) can be hampered by non-linear growth processes **(Dreybrodt and Scholz, 2011)**, and multiple cave-specific parameters such as vegetation cover (Haude, 1954; Wackerbarth et al., 2010), karst (Jean-Baptiste et al., 2019), and inner cave processes (Fairchild et al., 2006), which influence $\delta^{18}O_{speleo}$ . **Especially in the comparison between $\delta^{18}O_{speleo}$ of different speleothems, dating uncertainties complicate the assessment of climatic drivers, as they increase the uncertainty in pairwise comparisons and similarity estimates (Breitenbach et al., 2012; Rehfeld and Kurths, 2014)**. ..."

*L88: Please correct brackets around the reference in this line.*
Done.

*L116: '. . .freshwater hydrological cycle in the model shows only a slight overestimation in the local evaporation (Pardaens et al., 2003).' According to this statement, there is some evaporation included in the model. So it should be feasible to work with those data, instead to precipitation only (both amount and isotopic signature).*
See major comment section.

*L130: 'Vegetation above the cave has an impact on the source water . . .'. This reads a bit strange. Really on source water? Or rather on the amount of soil water and its d18O, which is coming from some source with a certain isotopic composition? Alternatively, you could write something like: Vegetation above the cave can alter the amount of infiltrating water and its isotopic signature.*
Thanks for the clarification here. We will use the last sentence that you suggested in the revised version.

*L130-131: Here you already state, what potentially can have some strong effect. Thus, I wonder a bit, why you do not have accounted for that stuff in your analysis.*
See major comment section.

*L134-135: This sentence should be changed, as it is not completely correct, if you mean with 'surface' the atmosphere. In addition, the CO2 and Ca2+ charging processes should be mentioned to make this better understandable for the reader. Please consider to rephrase to something like that: 'Infiltrating surface water is charged with soil gas CO2, which concentration is about 1-2 magnitudes larger than that of the atmosphere and enables the carbonic acid driven CaCO3 dissolution of the host rock. The generally lower partial pCO2 pressure conditions in the cave environment compared to that of the soil and epikarst makes the drip water degas ...'*
Thank you for this suggestion, which greatly improves readability. We will change the section, following your suggestion to:

**Infiltrating surface water is charged with soil gas $CO_2$, where the partial $CO_2$ pressure is larger than in the atmosphere, facilitating the carbonic acid driven $CaCO_3$ dissolution of the host rock. The generally lower partial $pCO_2$ pressure conditions in the cave environment compared to that of the soil and**

**epikarst makes the drip water degas** and precipitate calcite in a fractionation process, which consequently forms a speleothem (Tremaine et al., 2011).

*L138-139: This sounds very dramatic. Not from the wording, but from the implications. As it is written here, I hope this to be not true. Otherwise, nobody should trust such speleothem d18O reconstructions.*

We changed the section as follows:

"... During the calcification process, interactions with the cave environment or water inclusions within the mineral are still possible and, therefore, may further change the $\delta^{18}O_{speleo}$ archived in the speleothem. ..."

*L180: samples instead of sampled*

Done. Thanks.

*L202-203: Have you performed this averaging also for d18O in precipitation? From my understanding, of this sentence you do. But I think it would be better to use an amount weighted mean of d18O in precipitation? This is closer to the value really infiltrating into the soil/karst.*

We use precipitation weighted $\delta^{18}O$ throughout this analysis. We will make that more clear and changed the section as follows:

"...Temperature, precipitation, and isotopic data are extracted from the simulation at cave locations by bi-linear interpolation. Annual mean values for temperature, precipitation **and isotopic composition of precipitation** are formed by averaging over all months from April onwards to March of the following year. **This is also the time span for which precipitation weighted $\delta^{18}O$ ($\delta^{18}O_{pw}$) values are calculated.** ..."

*Based on that, what about evapotranspiration and changes of this during the modeled 1000 years? Have you accounted for that?*

*Maybe, I am wrong, but my understanding of those isotope enabled GCMs was, that they have at least an 'evaporation on land' module. This should also account for fractionation effects on the evaporated water, but also for the remaining water, what you are interested in. Would it be an option to use those d18O values instead of that of precipitation (again weighted by the amount of infiltration)? Maybe not for this manuscript, but in any future application.*

See major comment section.

*L219-221: Here you use the first time d18O$_{pw}$. It is not explained here nor somewhere else. What is this?*

Thanks for noticing. It stands for precipitation weighted $\delta^{18}O$ . We will explain it in the section starting L200, also following your remark to L202-203.

*In addition, I am sorry, but I do not understand exactly, why you are doing this Greens function approach? I get pretty ugly results in terms of mass balance with this, if tau is small (e.g., 1, 2 ,3 years). For example using a tau of 1 year: even after 100 years waiting time, only 58 percent reached the cave. For tau=3 that are 84 %, what reached the cave after 100 years. For larger tau it works better, I admit. But as you use this filter with a residence time of 3 years, I would be happy if you please could explain why you used this way of residence time description. Have you normalized this somehow?*

Following Dee et al. (2015), we use a normalization such that $\int_0^{TSend} g(t)dt = 1$, where $g(t) = 1/\tau \cdot e^{-t/\tau}$ as in the manuscript. Thank you for spotting that we did not mention this in the method section. We will add this in the revised manuscript.

*L228-229: Is it possible to rephrase this sentence? I regret to not being able to understand, what you mean by this.*

Thanks for pointing this out. This was also a comment by the first referee. We only choose the highest correlation estimate from significant cross-correlation estimates. If the cross-correlation between two speleothems using a specific pair of age-models is not significant at the 10% level, it is not chosen as a 'best fit'. We will clarify this in the manuscript at the respective section.

*Sec 3.2 and 3.3: Please explicitly state the number of used/available records for those approaches as those are most likely less, than that number mentioned in line 180.*

This is indeed a valuable information that is missing. There are three groups of speleothems that we analyze, each with more strict selection criteria depending on the analysis. We will add this to each analysis step in the results section.

*L249: 'Generally, modeled values appear to be more depleted overall than the mean values of speleothem _18Odw.eq . . .'. Wouldn't this be a hint, that evaporation is important and should be accounted for (at least in any potential follow up study)?*

See major comment section.

*L269: 'offsets also show a strong influence of temperature (Fig. 4d)' Only to repeat my*

*statements above: At warmer climates there is more evaporation, which lead to enhanced d18O values of the remaining water compared to that of precipitation. The remaining water soaks finally into the soil and cave. Thus, it would be worth to check if soil water d18O works better than precipitation.*

See major comment section.

*But this will most likely not solve the offset at colder T in the northern Hemisphere. But there, maybe it will work when using the amount weighted d18O of precipitation (or even better with infiltrating water) instead of annual mean d18O. As you brought up the example from Bunker cave. For this cave system and this region it was shown, that summer precipitation barely contributes to infiltrating water (most is gone by evaporation and transpiration), which is able to reach the cave (Riechelmann et al., 2011; Wackerbarth et al., 2010). As summer rain is more enriched in d18O compared to winter precipitation, this would shift the infiltrating water isotopic composition towards lighter values compared with the heavier annual mean d18O of precipitation. And would thus potentially bring the model results closer to the observed speleothem values.*

We do use precipitation weighted d18O throughout the analysis. However, this was not clearly stated in the methods section, which we will rectify in a revised version. As for the evaporation and infiltrated water, I refer to the major comments above.

*L298-279: I am sorry but I am confused again. You are writing: 'The global distribution of variance ratios between d18Odw.eq and d18O (Fig. 5a) shows overall higher variability in the speleothem records than in the simulation'. I agree with this observation, but this is somehow in contrast to Fig 5 b and c, where the variance ratio between d18Odw.eq and d18O is smaller than one. Is it possible that in Fig 5a you are showing the variance ratio of d18Odw.eq and the already down sampled d18O of the model simulation?*

You are correct. Fig 5a shows the global distribution where the simulation is already down-sampled to model resolution. We will clarify this in the stated line as follows:

"... The global distribution of variance ratios between $\delta^{18}O_{dw.eq}$ and **down-sampled** $\delta^{18}O$ (Fig. 5a) shows overall higher variability in the speleothem records than in the simulation, with local exceptions. ..."

We also updated the caption of figure 5 to prevent further confusion:

"(a) Spatial visualization of the site-based dimensionless variance ratio $V_{Rec}/V_{Sim}$, **where the simulated $\delta^{18}O$ is down-sampled to record resolution,** based on LM1. ..."

*L288: I think the reference to Fig 5 is not correct, as this figure does not give a hint on a 'smoothed model pattern'.*

Thanks for spotting this. The expression is very unfortunate. We wanted to express, that the modeled variances at the cave locations are very similar globally. However, as you state, this is not visible from Fig.5. We will leave out this reference in a revised version and add more information to the "smoothed model pattern."

*Figure caption 6: 'the simulated _18Opw but down-sampled to the same temporal resolution as in (a) with 3 year filter. 'Just to be sure, as it is not written somewhere: You first applied the three year filter and then did the down sampling, correct?*

This is a good point. So far the 3-year filter was applied after the degradation of the sampling resolution. We will explore to what extent applying the filter first changes the results in the revision process and ensure that this is better described in the methods section.

*L332: 'We find 18, 15 and 22 significant correlations from 87 entities. . . ': Out of curiosity, are the records/sites within those observed significant correlations in the three model runs always the same or are they varying? I mean if for example a record from one cave is significant for one run is the same cave record as well significant for the other simulations?*

Thank you for this attentive question. Indeed, from the total 55 significant correlations that we find for the three ensemble temperatures here, only half come from the same 12 entities, which again indicates at a very low signal to noise ratio in speleothems. We will add this, also for simulated precipitation and $\delta^{18}O$ , and include it in the discussion.

*Figure caption 8: indicates instead of indicat.*
Done.

*Figure caption 9: gridboxes instead of gridboxe*
Done.

*L385: 'spatial pattern for the offsets were not distinguishable.' I would slightly disagree here. Isn't it the case that in warmer climates the offset is more negative than in colder climates? See fig. 4d. You even described the low to high latitude trend in the northern hemisphere by yourself in lines 249-250.*

Thank you, we changed the section as follows:

"...Measured $\delta^{18}O_{dw.eq}$ followed general isotopic signature patterns as described by Dansgaard (1964). **The offsets are more positive in the extratropics of the Northern Hemisphere, which is also shown by their temperature dependency (Fig.4)."**

*L388: 'They find a stronger influence of seasonality of precipitation in warmer climates, highlighting the importance of a karst recharge model' Wouldn't it highlight the fact, that the model d18O values should be calculated as a precipitation weighted mean? Or even as infiltration weighted mean?*

See major comment section.

*L389-390: 'observed a strong temperature dependency reflected in the offset and ₋18Odw.eq over the last Millennium, showing the influence of fractionation . . .': You claimed, that you accounted for the temperature-dependent isotope fractionation during CaCO3 precipitation to calculate the d18O if the drip-water equivalent. So I think, this reason can be safely excluded.*

*I would evaluate it more likely that, as evaporation scales with temperature, the d18O values of precipitation have been changed by this process before the water even enters the epikarst zone. Thus, I would like to highlight it again (sorry for repeating myself), that using some infiltration weighted d18O mean is probably a better choice.*

Thank you for this clarification. We changed the section as follows:

"...Here, we also observed a strong temperature dependency reflected in the offset and $\delta^{18}O_{dw.eq}$ over the last millennium, showing the influence of cave internal processes on the $\delta^{18}O$ in drip water (Fig. 4) **but also additional fractionation processes or weighting through evaporation before the precipitated water enters the epikarst**. The higher offsets on the Northern Hemisphere possibly indicate a stronger influence of the continental effect. Still, from the records alone and with no karst-recharge **or evaporation** information, we were not able to distinguish specific climatic control regions. This requires a more thorough analysis including monitoring data **as well as more simulated variables**. ..."

*L410-411: 'However, we find little regional consistency and high heterogeneity in the variance estimates from the speleothem records'. While there is indeed some high heterogenity in the variance, I wonder, if this could be an argument for a strong influence of cave internal processes? That is a tricky one.*

*Later you discuss the correlation pattern, and this seems quite convincing that changes happen in the same direction and at about the same time at least at regional scale. Only the magnitude seems to change - as derived from the variance analysis. Wouldn't this rather mean, that each cave/stalagmite strengthen or weaken the initial climate signal, but that it*

*is still contained? As you said, this alteration could happen by soil/karst/cave processes. This way, one could argue about the magnitude of the changes, but not about the variation itself. How do you think about this?*

Thank you for this interesting thought. As discussed for Fig 9, we do find generally positive correlation estimates between entities within one cave, showing local consistency as you describe. However, this is not true for all caves. A more thorough analysis might be needed to systematically discuss this relationship, and also a larger global sample size of caves with multiple speleothems. Even though the SISALv2 database provides a very large dataset, for the last millennium only 12 caves exhibit multiple entities, that fit our analysis criteria. We will include a section on this in the discussion following L410-411, including your thoughts from above.

Also we change the section from L410 as follows:

These findings point to the influence of cave internal **and karst** processes **on mete-oric** $\delta^{18}O$ or the impact of seasonally filtered data captured by speleothems, which is in agreement with McDermott et al. (2001).

*L416-417: 'Studies observing transit time in karst systems find increases in drip rate after an increase in precipitation e.g. after days (Riechelmann et al., 2011).' I suggest to term this Riechelmann et al., (2011) work rather as a study, which investigates the cave reaction time to precipitation events - not a transit time study. And the cave reaction on heavy rain events is often fast (as observed in other caves as well), but it will not change the transfer time (too much). The residence time can only be found by tritium or other appropriate tracer isotopes (as you correctly describe below.)*

Thank you for suggesting this more precise term. We will replace "transit time" with "cave reaction time" in this sentence in the revised version.

*L442: I guess, cave monitoring would not only help to compare two caves but also to improve the comparability between a cave and climate model results. You could add this as well. Furthermore, I think that not only monitoring would help, but that (model-based) weighted infiltration values would also help (as written already earlier). Even with your sentence in line 438 you imply so by yourself.*

We changed the section as follows: "...Further dripwater monitoring studies **combined with a comparison to model data output would help to**characterize the seasonality of individual caves **and would, therefore, lead to deeper** understanding of which climatic signal is captured by speleothems and enhance comparability between different caves...."

*L455: 'but could also be influenced by non-climatic overprints on the _18O signal up to*

*the centennial scale.' I would argue, that nearly all changes in the processes in the cave are climate driven. Unfortunately, sometimes they counter the pure climate imprint and on other locations they amplify it.*

Thanks for pointing this out. With non-climatic, we meant to say environmental overprints such as changes in the epikarst which may result in different growth rates, that have not directly climate driven. We change the section as follows:

"...In part, this can be due to heterogeneous temporal resolution, but could also be influenced by non-climatic **environmental** overprints on the $\delta^{18}O$ signal up to the centennial scale. ..."

*L476: 'as a bias of 1C in the simulated temperature would account for a change in $_{18}Odw.eq$ of approximately 2 ‰' If you really take the d18O-T relationship of Tremaine (or any similar) this statement appears to be wrong. Fractionation during CaCO3 precipitation is T-dependent by $\sim 0.25$ permil per $^\circ C$. So this would mean an offset of 1 permil, if the modeled T is wrong by 4 $^\circ C$. I guess this would only be the case in mountainous regions, were the orography of the model is not close enough to the true altitude of the cave. You mentioned some examples earlier in your manuscript.*

*L477:'A bias of 1 ‰in the $_{18}Odw.eq$ however, accounts for a temperature change of $0.1\_C$ for the lowest simulated annual mean cave temperature ($3.1\_C$ in Norway), and a change of $13.1\_C$ for the highest simulated annual mean cave temperature ($32.5\_C$ in the tropics).' This puzzles me now quite a lot. You said in your earlier sentence, that 1 $^\circ C$ is 2 permil. This does not fit with your statement in this sentence. Maybe, you mean something different?*

Apologies for the inconsistency. We have checked the calculation. The corrected numbers do not change our interpretation. We will clarify the calculation in the manuscript and change the numbers in this section as follows:

"...Knowing the actual temperature history of the caves better could strongly reduce the uncertainty, as a bias of $\Delta 1^\circ C$ in the simulated temperature would account for a change in $\delta^{18}O_{dw.eq}$ of approximately $\Delta 0.2$‰. A bias of $\Delta 1$‰ in the $\delta^{18}O_{dw.eq}$ however, accounts for a temperature change of $4.5^\circ C$ for the lowest simulated annual mean cave temperature ($3.1^\circ C$ in Norway), and a change of $5.5^\circ C$ for the highest simulated annual mean cave temperature ($32.5^\circ C$ in the tropics). ..."

**References**

Breitenbach, S. F., Rehfeld, K., Goswami, B., Baldini, J. U., Ridley, H. E., Kennett, D. J., Prufer, K. M., Aquino, V. V., Asmerom, Y., Polyak, V. J., Cheng, H., Kurths, J., and Marwan, N.: Constructing proxy records from age models (COPRA), Climate of the Past, 8, 1765–1779, https://doi.org/10.5194/cp-8-1765-2012, 2012.

Dansgaard, W.: Stable isotopes in precipitation, Tellus, 16, 436–468, https://doi.org/10.3402/tellusa.v16i4.8993, 1964.

Dee, S., Emile-Geay, J., Evans, M. N., Allam, A., Steig, E. J., and Thompson, D. M.: PRYSM: An open-source framework for PRoxY System Modeling, with applications to oxygen-isotope systems, Journal of Advances in Modeling Earth Systems, 7, 1220–1247, https://doi.org/10.1002/2015MS000447, 2015.

Dreybrodt, W. and Scholz, D.: Climatic dependence of stable carbon and oxygen isotope signals recorded in speleothems: From soil water to speleothem calcite, Geochimica et Cosmochimica Acta, 75, 734 – 752, https://doi.org/https://doi.org/10.1016/j.gca.2010.11.002, URL `http://www.sciencedirect.com/science/article/pii/S0016703710006277`, 2011.

Fairchild, I. J., Smith, C. L., Baker, A., Fuller, L., Sp??tl, C., Mattey, D., and McDermott, F.: Modification and preservation of environmental signals in speleothems, Earth-Science Reviews, 75, 105–153, https://doi.org/10.1016/j.earscirev.2005.08.003, 2006.

Haude, W.: Zur praktischen Bestimmung aktuellen und potentiellen Evaporation und Evapotranspiration, Mitteilungen des Deutschen Wetterdienstes, Band 1, 1954.

Jean-Baptiste, P., Genty, D., Fourré, E., and Régnier, E.: Tritium dating of dripwater from Villars Cave (SW-France), Applied Geochemistry, 107, 152–158, https://doi.org/10.1016/j.apgeochem.2019.06.005, 2019.

McDermott, F., Mattey, D. P., and Hawkesworth, C.: Centennial-scale holocene climate variability revealed by a high-resolution speleothem $\delta18O$ record from SW Ireland, Science, 294, 1328–1331, https://doi.org/10.1126/science.1063678, 2001.

Rehfeld, K. and Kurths, J.: Similarity estimators for irregular and age-uncertain time series, Climate of the Past, pp. 107–122, https://doi.org/10.5194/cp-10-107-2014, 2014.

Thornthwaite, C. W. and Mather, J. R.: Instructions and tables for computing potential evapotranspiration and the water balance, Tech. rep., Centerton, 1957.

Tremaine, D. M., Froelich, P. N., and Wang, Y.: Speleothem calcite farmed in situ: Modern calibration of $\delta18O$ and $\delta13C$ paleoclimate proxies in a continuously-monitored natural

cave system, Geochimica et Cosmochimica Acta, https://doi.org/10.1016/j.gca.2011.06.005, 2011.

Wackerbarth, A., Scholz, D., Fohlmeister, J., and Mangini, A.: Modelling the delta 18 O value of cave drip water and speleothem calcite, Earth and Planetary Science Letters, 299, 387–397, https://doi.org/10.1016/j.epsl.2010.09.019, 2010.

---

## Author Response (AR1)

**Authors' response to the reviewers' comments: Comparison of the oxygen isotope signatures in speleothem records and iHadCM3 model simulations for the last millennium (cp-2019-121)**

February 20, 2021

**Summary of changes**

In response to the suggestions by the two reviewers, we implemented the following changes to the manuscript:

- We included a discussion of the influence of external forcing (one new figure in the supplement) as well as potential local evaporative and infiltration effects on speleothem oxygen isotope ratios

- Age-model uncertainty information was included in Fig. 8 and SFig. 7, and more thoroughly included throughout the discussion

- We carefully revised the method section and improve those sections, where precipitation weighted $\delta^{18}O$ is used and improve the explanation of the karst filter

- We modified the spectral analysis: We now first apply the karst filter to annual precipitation-weighted data, and then apply the temporal degradation. This did not affect the overall results, but as one reviewer pointed out, is closer to physical reality. Fig.6 was updated accordingly.

- We revised the text throughout the manuscript to clarify statements and fixed formatting where necessary.

Note that we also found and fixed a bug in the extraction scripts for the model data at cave locations, which led to small changes in the precipitation-weighted isotope ratio. This did not affect the overall results, but changed those for individual sites (Sect. 4.1, Fig. 3). The dataset (on Zenodo) and the software repository on Github have been amended accordingly.

We thank both reviewers for their constructive comments and detailed reading, which greatly helped to improve the manuscript. A detailed response to their helpful remarks is given below in blue, and a summary of the actions we took are given in green.

**1 Reply to the first reviewer**

(Original report cited in italics)

*Major comment #1 Why even look at teleconnections in the d18O data (Figure 8)? There is no mention of ITCZ variability, monsoon, NAO, or other mechanisms driving large-scale d18O variability. I can understand if the authors want to keep the analysis general, but through the whole paper there is no mention of any of the main climate patterns that could explain the teleconnections in Figure 8. See references listed in comment for L78-L80. This should at the very least be mentioned in the introduction, and included in the discussion. There is a lack in information of how HadCM3 performs when it comes to large-scale patterns, and what the imprint is on d18O. For example, add extra correlation maps in Figure 7 for the most important patterns, which quickly could be done. Correlating the monsoon index (e.g. Vuille et al., 2005) to d18O in precipitation should show a very clear pattern across the region around the Indian Ocean. This is not obvious when looking at grid point correlation of climate fields, because the main driving factor is not local precipitation amount, but down wind recycling of vapour in large-scale organized convection.*

We agree that the investigation of modes of variability and the modeled $\delta^{18}O$ response is very interesting. However, attributing the variability of speleothem $\delta^{18}O_{speleo}$ to specific modes is a challenge that we would not be able to address in a single manuscript. Also, a full-blown analysis on local drivers is not feasible and appropriate given the resolution of the model. The key idea is to show the general correlation patterns obtained from the data, and that of the model while taking into account the resolution difference. This does indeed not allow us to look into teleconnections or directly attribute to drivers of variability. Instead, it gives an overview of the general global spatial correlation structure, and whether there are similarities between the relationships obtained from proxy data, and from model simulations. We identify quite some gaps that require taking into account more proxy-related uncertainties and processes. Investigating to what extent large-scale modes of variability impact $\delta^{18}O$ variability (following e.g. Midhun et al., 2020) would of course be very interesting, and while an exhaustive investigation is beyond the scope here, we plan to explore the potential for it. We started to initially test the impact of some modes (NAO-index, ENSO-index) and plan to extend the discussion to include some of these new insights.

Action: Paragraph added in the discussion and SF19 added to supplements.

***Major comment #2***  *The authors mention external forcing several times as a driver of variability, but never explains or does any analysis to show how this this is related to climate or d18O. This is of course a big topic (e.g. Swingedouw et al., 2017) and might be beyond the scope of the paper. Please either perform analysis of the impact of forcings or be more careful when making statements about what variability is forced and what is not forced.*

A very interesting point, indeed. We will re-check our statements to ensure that forced and internal variability are appropriately distinguished. We have previously correlated the solar irradiance time series used for the forcing with $\delta^{18}O$ variability at the annual scale without seeing strong impacts. As visible in Fig. 1 below, this yields hardly any regions with correlation coefficients clearly distinct from zero. Volcanic forcing on the other hand, shows a clearer imprint on $\delta^{18}O$ variability. We will add a new figure to the supplement to underline these statements. These aspects are not explicitly discussed in the paper and we will amend the discussion to include them. Fig. 1 below illustrates the correlation map between a-c) volcanic forcing to ensemble mean temperature, precipitation, and $\delta^{18}O$ changes, and in d-f) solar forcing to the climate variables. Especially for temperature, we see a clear climatic influence by volcanic forcing, which is also visible in the time series of GMST. Precipitation and its isotopic composition however show only a weak and non-uniform influence of volcanic forcing. Generally, the influence of solar forcing on all climate variables is very weak. The area-weighted mean correlation to solar forcing to the isotopic composition of precipitation is $-0.01$ ($-0.04$, 0.06 90% confidence interval) and $-0.08$ ($-0.18$, 0.00) for volcanic forcing. We conclude that the external forcing has little influence on the $\delta^{18}O$ signature in the simulation. We will include this figure in the supplement of the manuscript and incorporate these points in the discussion.

Action: Figure included as SF17. Comment added in section 5.4 Limitations of the Discussion.

***Major comment #3***  *The authors have three simulations but appear to make very little use of the additional information to be gained from this. While three simulations is not a huge ensemble it still yields much more information on forced versus internal variability than a single simulation. When you perform correlation analysis between speleothem data and simulated d18O, this should be done using the ensemble mean. How similar are the ensemble runs in variability? How is the ensemble setup? There is very little information on this.*

Following the suggestion, we will extend the description of the ensemble. The data, and its description, was uploaded to Pangaea prior to submission but apparently, they are not available yet. The three ensemble members were initialized from different years of the same spinup simulation. We will add this in the method section. We do not however, think that using the ensemble mean is necessarily appropriate in the assessment of variability changes, as this would amplify the forced response and dampen the dynamic part in the

[Figure]

Figure 1: Correlation estimate fields of a) ensemble mean simulated temperature, b) precipitation, and c) $\delta^{18}O$ to volcanic forcing, and the same ensemble mean climate variables to solar forcing d-f). Empty tiles mask gridboxes with $p > 0.1$. The area-weighted average correlation estimates with volcanic forcing are $\rho(T,volc) = -0.34$ ($-0.48$, $-0.11$), $\rho(P,volc) = -0.04$ ($-0.15$, 0.05), $\rho(\delta^{18}O ,volc) = -0.08$ ($-0.18$, 0.00). The area-weighted average correlation estimates with solar forcing are $\rho(T,sol) = 0.01$ ($-0.004$, 0.03), $\rho(P,sol)$ = 0.003 ($-0.009$, 0.024), $\rho(\delta^{18}O ,sol) = -0.012$ ($-0.035$, 0.056). The correlation estimates are calculated with 989 degrees of freedom.

signal. The relative role of natural forcing is not the main focus of this paper. We will assess the degree of correlation between the $\delta^{18}O$ fields that is due to the common forcing. The insight from this analysis will also be added to the discussion.

Action: Done. Adjustment in the model description and in the methods. The data is now uploaded to Zenodo and cited in the manuscript.

**Major comment #4** *The study uses a shot gun kind of approach to age-model uncertainties. As I understand the different age-models of individual speleothems are sampled independently when testing the range of possible age-models. But are all age-models really equally likely, for example for neighbouring speleothems that we expect to be correlated? Related to this.*

The new age-models provided with the SISAL chronology (Comas-Bru, Rehfeld, Roesch et al., ESSD 2020) are not ranked by likelihood. All of them are consistent with the radiometric chronological constraints. Therefore we indeed consider all of them in the correlation analyses. In all other analyses we use the corresponding original age models. We make this clearer in the method section.

Action: Adjusted in the Data and Methods section

*When comparing the down-sampled modelled d18O to speleothem data in Figure 8,*

*shouldn't the age-model uncertainties also be included for the model data to make the results truly comparable? For completeness there should be two more tests plotted in Figure 8: i) model data which is not down sampled (include SF7 a) and b) in Figure 8, I suppose?) ii) model data including age-model uncertainties. I think this issue with the comparison of model and speleothem data and differences in teleconnections depending on data treatment should be more emphasized.*

We absolutely agree with the reviewer that adding this information makes Fig. 8 more informative. We will update Fig. 8 accordingly. It will include the record data, the simulated data and the downsampled-data, including the age-model uncertainty testing. We will also adjust the color scheme, as suggested in the detailed comments.

Action: Performed down-sampled age-model sensitivity test; added different colors; figure expanded; added explanation and interpretation in results and discussion

**2    Reply to the second reviewer**

**Major comment #5**

*For example, and this is my largest point of critics: I am really not sure, if it is the mean annual d18O of precipitation, what speleothem are recording, but what seems to be used here from the model output. Isn't it rather the case that speleothems record the amount weighted d18O values of precipitation? Or in some locations, especially in more arid regions, with low amount of precipitation but elevated T, speleothem d18O reflect more likely the amount weighted d18O of infiltrated water. Thus, evaporation processes are important but not considered here. However to my understanding climate models do provide those variables. They should have evaporation processes on land included. Wouldn't it be an option to try this variable for analysis? If I remember correctly, Wackerbarth et al., (2012, CP) did such an approach.*

*Especially, as the models should even account for evaporation-dependent fractionation processes of oxygen isotopes during evaporation, comparing the speleothem results with those of the models should potentially result in better agreement. The d18O values of the remaining, the non-evaporated water, which is finally entering the deeper soil layers and the karst system, would be known from the model. I guess this would be the easiest way (without any need to use cave-site specific insights from cave monitoring studies) to better compare model results and speleothem d18O values in a more comprehensive way. At this point, I don't ask to redo all the analysis with an infiltration weighted mean d18O instead of an annual mean d18O of precipitation, but it would be appropriate to at least include this possibility in the discussion section (e.g. Sec. 5.4) – and try this in a potential follow-up study.*

*To my understanding, those evaporation processes could very well explain, why the d18O*

*model data of precipitation tend to underestimate the speleothem-recorded d18O values in warm and more arid regions (see your Figs. 3d and 4d). In those regions, evaporation is very important and influences both – the amount of infiltrating water per month and the evaporation-dependent d18O enrichment of the non-evaporated water. I think that if one would account for that, it would potentially improve your model-data comparison. At least when comparing the mean values of the last 1000 a. But it has maybe even the potential to increase the average variability of the modeled d18O values, compared to your approach using rain water. And it maybe even brings some additional variance on the longer scales into the data. The reason for that could be that d18O of infiltrating water would - in addition to changes in the d18O of precipitation – potentially show a large change due to temperature effects on the amount of evaporation and the d18O of the remaining infiltrating water.*

We agree with the reviewer, that the precipitation $\delta^{18}O$ signature that effectively gets transmitted to the karst system and the drip sites is dependent on the precipitation amounts. For that reason, we do account for precipitation amount impacts on the signal, by using precipitation-weighted $\delta^{18}O$ values. In most regions there is a good degree of correlation between precipitation-weighted, and unweighted $\delta^{18}O$. We did, however, not make that very clear in the methods. We will rectify this in the revised version.

We also agree on the benefits of using simulated infiltrating water amounts instead of the annual mean of precipitation. To calculate the infiltration rates the proportion of evaporation is needed for calculation. Evaporation processes are of course considered in HadCM3. Due to storage limitations, however, only selected variables were stored during the simulation. This did not include evaporation and all land model variables except for vegetation fractions. Obtaining evaporation could be possible by rerunning the model, or by estimating it through heuristic approaches as in Thornthwaite and Mather (1957) by considering the latent heat flux from the surface. However, this would also introduce additional uncertainty into the analysis. In essence, we agree that we could better discuss the potential effects of evaporation changes on the speleothem records. In the minor comments the reviewer points out several occasions, where evaporation might help in the interpretation of the results. Therefore, we will check for the potential effects of evaporation in the results and add further discussion in the revision. The suggested analysis will also be taken into account for a follow-up project where we include a multi-model data comparison and have access to evaporation output from several simulations and we can cross-compare approaches to estimate evapotranspirative contributions.

Action: larger paragraph in discussion-limitations and methods added; small clarifications on evaporation throughout the text

*Otherwise, I have only comments of more minor attitude. Please find them listed below (with some repeatedly occurring instances, where I address the advantages of accounting for evaporation and amounted weighted means).*

*To sum up, I would really like to see this work published pending on an improved manuscript version, where my major concern is accounted for (in which way the authors feel more comfortable with) and the smaller issues from below are considered/discussed.*

*Best wishes, Jens Fohlmeister*

**3  Detailed Comments**

We sorted the detailed comments by their reference to lines in the manuscript. The reviewers are indicated as **R1** and **R2** respectively. Where both reviewers commented on the same section, we combined the answers such that both comments are addressed in one segment.

**R1:** *L5-L12 Briefly say that d18O is a climate proxy before discussing all the implications of sampling etc.*

We agree that adding this general statement prior to the specific impacts improves readability. We adjusted the text as follows:

'... The oxygen isotopic ratio $\delta^{18}O$ **, a proxy for many different climate variables,** is routinely measured in speleothem samples at decadal or higher resolution and single specimens can cover full Glacial-Interglacial cycles. ...'

Action: Done.

**R1:** *L15 "We evaluate systematically. . . " change to "We systematically evaluate . . . "?*

Done.

Action: Done.

**R1:** *L16 ". . . and test for the main climate drivers for individual records or regions." change to ". . . and test for the main climate drivers recorded in d18O for individual records or regions."?*

Done.

Action: Done.

**R1:** *L17-L19 Maybe it is be better (worse) to explain in full sentences (fancy truncated syntax)?*

**R2:** *L 18: "proxy-based variability of d18O": d18O is a proxy. Thus your phrasing sounds a bit weird. In the context of the text you might mean 'archive-based'?*

We liked the short syntax. Nevertheless, we adjusted the text with the suggestions of both reviewers as follows:

'... However, using robust filters and spectral analysis, we show that the observed **archive-**

based variability of $\delta^{18}O$ is lower than simulated by iHadCM3 on decadal**, and higher on centennial** timescales. ...'

Action: Done.

**R2:** *L19-20: You might should add, that most of the difference on the side of the 'short-term variability' ($<\sim20a$) comes from smoothing due to soil water residence time and resolution. You showed quite nicely that on the long frequencies, both types of data sets do not agree – whatever you tried.*

We adjusted the section as follows:

'...Most of this difference can likely be attributed to the records' lower temporal resolution and averaging **or smoothing** processes affecting the $\delta^{18}O$ signal **e.g. through soil water residence times**. ...'

Action: Done.

**R1:** *L28 ". . . natural and human systems . . . " maybe change to ". . . human societies and the environment . . . "?*

We adjusted the text as follows:

'... The impacts of a changing climate have been observed over the last century (IPCC, 2013) and indicate a strong sensitivity of **human societies and natural systems** to changes in climate. ...'

Action: Done.

**R1:** *L36 The delta-notation comes in here before defining it or telling what the proxy is good for. Either move the definition up in the manuscript and description of the d18O proxy or call it "the relative abundance of 18-O" until you get to the definition, and then explain briefly that this is a climate proxy. You can't discuss the challenges of the interpretation before telling the basics.*

We adjusted the section as follows:

'... Therefore, for model evaluation on longer than centennial time scales, we have to rely on evidence from paleoclimate archives, such as trees, ice cores, foraminifera from marine sediment cores, or speleothems. **The abundance of the heavy oxygen isotope $^{18}$O, further denoted as $\delta^{18}O$ , is a proxy for many climate variables and** can be measured on these, and quite a few other paleoclimate archives with high precision (Schmidt et al., 2014). ...'

The formal definition then follows, starting in L63.

Action: Done.

**R1:** *L41 You need to include the simulations with GISS ModelE2-R (Colose et al., 2016) and iCESM1 (Stevenson et al., 2019).*

We now include these studies in the literature review.
 Action: Done.

**R1:** *L42 Sjolte et al. (2018) compared the variability of the modelled ECHAM5/MPI-OM d18O to Greenland ice core d18O and used the model to assimilate the ice core data to produce gridded reconstructions. Never compare the proxy data to the model – it's the other way around!*

**R1:** *L44 Again: Never compare the proxy data to the model! It's not the observations that are being evaluated.*

We will revise the manuscript for the expression and change the sentence structure accordingly. The lines mentioned here were changed to:

'... Few other transient model-data comparison studies focused on $\delta^{18}O$ (e.g., Wackerbarth et al., 2012; Dee et al., 2015; Parker et al., 2020). For example, Sjolte et al. (2018) **compared the variability of the simulated ECHAM5/MPI-OM $\delta^{18}O$ to Greenland ice cores** over the last millennium **to assimilate the ice core data to produce gridded reconstructions. They** were able to differentiate between solar and volcanic forcing effects from their reconstructions. On orbital timescales (150,000 yr), Caley et al. (2014) compared **a transient isotope-enabled simulation with the model of intermediate complexity iLOVECLIM to speleothem records from South East Asia**. They found model-data similarity for the broad temporal trends, but differences at shorter timescales, highlighting the role of seasonality. ...'
 Action: Done.

**R1:** *L56-L61 These are a very important points and is written in almost bullet point-style. Please add more details to make it more comprehensible to non-experts. For example, Laepple and Huybers (2014a) are talking about decadal and longer time scales. Laepple and Huybers (2014b) say that the models are too diffusive which is not the same as saying "too high diffusivity", depending on context. Here, they mean that the energy dissipates too quickly across the spectra of temporal variability, which is not clear in your text. My advice is to spend a bit more space on this part of the introduction and don't mix topics, such as too diffusive models and missing processes and feedbacks in the same sentence, unless linking these things directly together.*

We agree with the reviewer on the importance of these points and will therefore grant the section more space in the revised version.
 Action: Done.

**R1:** *L64 Add white space after "climate system".*
**R2:** *L 64: a space is need after 'system'*
Done.
Action: Done.

***R2:*** *L65: I would be more specific here and state that 'The ratio of H218O to H216O in precipitation is an indicator of evaporation temperature, . . .' as it is possible to determine the d18O values in other reservoirs as well. And there other effects are also important.*

We will adjust the sentence using the reviewer's suggestion.

Action: Done.

***R1:*** *L78-L80 There is quite some evidence that d18O is not primarily a proxy of neither local temperature nor precipitation, but strongly related to circulation modes, large-scale climate patterns and downwind fractionation. For example, in the North Atlantic region the North Atlantic Oscillation (NAO) is important for d18O variability (Vinther et al., 2010; Sjolte et al., 2011; Deininger et al., 2016), while downwind fractionation connected with the Indian summer monsoon impacts the cave d18O in the region around the Northern Indian Ocean, China and South-East Asia (Vuille et al., 2005; Fleitmann et al., 2007; Pausata er al., 2011; Kurita et al., 2013; Lekshmy et al., 2014; Liu et al., 2014; Sjolte et al., 2014; Zhang and Jin, 2015). I think these factors should be highlighted in the introduction.*

We adjusted the section as follows:

'... $\delta^{18}O$ can be regarded as a proxy **for example** for surface temperature variations in higher latitudes, and precipitation amount in the tropics (Dansgaard, 1964), overlayed with distinct observable signatures of source water evaporation**,** transportation over longer distances (Bradley, 1999; Dansgaard, 1964)**, and large scale-climate patterns of circulation such as e.g. the North Atlantic Oscillation (NAO) (e.g. Vinther et al., 2010) or the El-Niño Southern Oscillation (ENSO) (Tindall et al., 2009)**. These $\delta^{18}O$ **signatures in precipitation may** also visible in speleothem records, including additionally a fractionation process involved in the calcite formation, which is primarily temperature-dependent (Urey, 1948; McCrea, 1950)...'

Action: Done.

***R2:*** *L83: ' . . . hampered by non-linear growth processes (Dreybrodt 1980).' Is Dreybrodt 1980 the correct reference, as he focussed only on precipitation of CaCO3? Not on d18O variations. Maybe use one of his later studies, e.g., Dreybrodt and Scholz, 2011 or Dreybrodt and Romanov, 2016.*

***R2:*** *L85: '. . .as well as dating uncertainties'. Please explain, how dating uncertainties shall have an influence on the interpretation of d18O as you state here.*

We will exchange the reference and add an explanation to the influence of dating uncertainties as follows:

"... The climatic interpretation of speleothem $\delta^{18}O$ variations in calcite or aragonite

(hereafter $\delta^{18}O_{speleo}$ ) can be hampered by non-linear growth processes **(Dreybrodt and Scholz, 2011)**, and multiple cave-specific parameters such as vegetation cover (Haude, 1954; Wackerbarth et al., 2010), karst (Jean-Baptiste et al., 2019), and inner cave processes (Fairchild et al., 2006), which influence $\delta^{18}O_{speleo}$ . **Especially in the comparison between $\delta^{18}O_{speleo}$ of different speleothems, dating uncertainties complicate the assessment of climatic drivers, as they increase the uncertainty in pairwise comparisons and similarity estimates (Breitenbach et al., 2012; Rehfeld and Kurths, 2014)**. ..."

Action: Done.

**R2:** *L88: Please correct brackets around the reference in this line.*
Done.
Action: Done.

**R1:** *L91-L99 Here you mainly list the contents of the paper. Can you make the science questions that you are pursuing more clear? Maybe you are testing the climate controls on the variability in simulated d18O using an isotope enabled climate model and compare this to speleothem d18O in a global dataset? Formulate more like hypothesis testing rather than say what kind of analysis you are doing.*

We plan to restructure the paragraph such that it reads:
"... Here, we present three new last millennium isotope-enabled simulations from the iGCM version 3 of the Hadley Model (iHadCM3) and test how similar the $\delta^{18}O$ variations in iHadCM3 and speleothem records are (Sec. 4.1). A characterization of the datasets and relevant forcing can be found in Fig. 1. The robustness of the findings and methods are evaluated over the last millennium, for which a large number of high-resolution proxy datasets from the SISAL v.2. database (Comas-Bru et al., 2020) are available. **Our key question are: i) how similar are the modeled $\delta^{18}O$ signatures to the speleothem records especially regarding variability, ii) can we distinguish main drivers for these signatures, and iii) how representative are the speleothem records for their region. To address these questions,** we explore these similarities on both spatial and temporal scales, to distinguish patterns of the mean state (Sec. 4.1), the variability (Sec. 4.2 and Sec. 4.3), and the spatial representativity of speleothem climate records (Sec. 4.4 and Sec. 4.5). We examine the simulation's capability to simulate and the records' capability to capture variability on different time scales to improve our understanding of processes and uncertainties of both. ..."

**R1:** *L108 Add white space ". . . 30min . . . "*
Done.
Action: Done.

**R1:** *L115 What is meant by "ice sheet" here? I suppose the model doesn't have an ice sheet model?*

This statement refers to the sea ice model as documented in Valdes et al. (2017).

'... Compared to instrumental observations, the model represents sea surface temperature (SST), **sea ice**, and ocean heat content well (Gordon et al., 2000). ...'
Action: Done.

**R2:** *L116: '. . .freshwater hydrological cycle in the model shows only a slight overestimation in the local evaporation (Pardaens et al., 2003).' According to this statement, there is some evaporation included in the model. So it should be feasible to work with those data, instead to precipitation only (both amount and isotopic signature).*

See major comment #5 section.

Action: We added the explanation, why no evaporation was used in the limitations chapter.

**R1:** *L120 ". . . features like latitude effect, amount effect, or the continental effect . . . " this is partly repetition from L118. Why not lump these things together?*

We rearrange the sentence to avoid repetition as follows:

'...The model simulates the major isotopic fractionation effects defined by Dansgaard (1964) (e.g. the latitude effect, the amount effect, and the continental effect) appropriately compared to GNIP data (Zhang et al., 2012). Additionally, a broad agreement in isotopic output with GNIP data in the general spatial distribution can be observed and the **above mentioned** general oxygen isotopic ratio features are **represented** well (Tindall et al., 2009). As such, iHadCM3 captures large scale features of climate and oxygen isotope ratios while remaining computationally efficient for the simulation of timescales such as the last millennium...'
Action: Done.

**R1:** *L123 So, what are the differences between the three model runs? Initial state of the ocean?*

See response to major comment #3 above.
Action: Done. See action response to major comment #3

**R2:** *L130: 'Vegetation above the cave has an impact on the source water . . .'. This reads a bit strange. Really on source water? Or rather on the amount of soil water and its d18O, which is coming from some source with a certain isotopic composition? Alternatively, you could write something like: Vegetation above the cave can alter the amount of infiltrating water and its isotopic signature.*

As suggested, we will use the clarification in the last sentence in the revised version.
Action: Done.

**R2:** *L130-131: Here you already state, what potentially can have some strong effect. Thus, I wonder a bit, why you do not have accounted for that stuff in your analysis.*

See major comment #5 section.
Action: See major comment #5 section.

**R2:** *L134-135: This sentence should be changed, as it is not completely correct, if you mean with 'surface' the atmosphere. In addition, the CO2 and Ca2+ charging processes should be mentioned to make this better understandable for the reader. Please consider to rephrase to something like that: 'Infiltrating surface water is charged with soil gas CO2, which concentration is about 1-2 magnitudes larger than that of the atmosphere and enables the carbonic acid-driven CaCO3 dissolution of the host rock. The generally lower partial pCO2 pressure conditions in the cave environment compared to that of the soil and epikarst makes the drip water degas ...'*

We will change the section to improve readability, following the reviewer's suggestion to:

"... **Infiltrating surface water is charged with soil gas $CO_2$, where the partial $CO_2$ pressure is larger than in the atmosphere, facilitating the carbonic acid driven $CaCO_3$ dissolution of the host rock. The generally lower partial $pCO_2$ pressure conditions in the cave environment compared to that of the soil and epikarst makes the drip water degas** and precipitate calcite in a fractionation process, which consequently forms a speleothem (Tremaine et al., 2011). ..."
Action: Done.

**R2:** *L138-139: This sounds very dramatic. Not from the wording, but from the implications. As it is written here, I hope this to be not true. Otherwise, nobody should trust such speleothem d18O reconstructions.*

We changed the section as follows:

"... **During the calcification process**, interactions with the cave environment or water inclusions within the mineral are still possible and, therefore, may further change the $\delta^{18}O_{speleo}$ archived in the speleothem. ..."
Action: Done.

**R1:** *Figure 2, caption. Add white space "600yr".*
**R1:** *L153 "600y" add white space, and I believe Clim Past uses "yr" shorthand for year.*

We revised the document for white spaces and modified the abbreviation.
Action: Done.

**_R1:_** _L161-169 As I understand you allow any type of age model to be used out of the many options, and you pick the best fit independently for each site/speleothem? What are the criteria for accepting an age model, be sides that it is the best fit? Are there cases where the "best" age model is outside of the uncertainty range of the U/Th dating?_

In general we use the authors' original chronology for the analysis. In the cross-correlation analysis (esp. Fig 8 and 9) we test, how much correlation estimates change depending on the choice of age models. Here we use all age-model realizations provided by the SISALv2 database. The SISALv2 database defines one median best-fit estimate for each age modeling method, following their selection criteria (see Comas-Bru et al., 2020). As a median age model, it cannot lie outside the range of the ensemble members. Nevertheless, age controls only exist at the radiometric dates and depending on the dating density and whether reversals were found some age models may of course lie outside individual dates if the dating evidence is contradictory. These age models are method-dependent, consistent with the evidence, and free of reversals, as described in Comas-Bru et al. (2020). In the correlation assessment, we use all available age model realizations. We will adjust the section. See also our response to major comment # 4.

Action: additional paragraph in methods (not here in data)

**_R1:_** _L176 How do you decide on the nine clusters? Is this what you describe L236-L239. Please clarify._

We decided on eight distance-based clusters and manually added a ninth cluster, to separate a cluster containing all East Asian speleothems above 20°N from those below. We made the link to the latter section, where it is explained in more detail, clearer as follows:

L176: 'For the investigation of spatial correlation patterns by network analysis the set of speleothems is divided into nine regional clusters (Fig. 2)**, as explained in detail in Sec. 3.3**.'

L236: 'We split the network into **eight** sub-networks by hierarchical distance-based clustering of the node locations. The cluster that includes all East Asian caves is manually split into two clusters, one for East Asia (all caves above 20°N) and a cluster of South East Asia (all caves below 20°N). **With this, we end up with nine clusters as depicted in Fig. 2.**...'

Action: Done.

**_R1:_** _L180 '. . . 10 or more d18O sampled.' should it be '. . . 10 or more d18O samples.'? Otherwise please rephrase._

**_R2:_** _L180: samples instead of sampled_

Done.

Action: Done.

**R1:** *L181 "We exclude six speleothems of mixed mineralogy." Why?*

We require information on the mineralogy of the samples for the conversion to drip water $\delta^{18}O_{dw.eq}$ . For samples of mixed mineralogy it is unclear to what extent the correction is appropriate. Therefore, and following Comas-Bru et al. (2019), we excluded those speleothems with mixed mineralogy. We will add this clarification to L181 in the manuscript.

Action: Done.

**R2:** *L202-203: Have you performed this averaging also for d18O in precipitation? From my understanding, of this sentence you do. But I think it would be better to use an amount weighted mean of d18O in precipitation? This is closer to the value really infiltrating into the soil/karst.*

We use precipitation weighted $\delta^{18}O$ throughout this analysis. We will make that more clear and changed the section as follows:

"...Temperature, precipitation, and isotopic data are extracted from the simulation at cave locations by bi-linear interpolation. Annual mean values for temperature, precipitation **and isotopic composition of precipitation** are formed by averaging over all months from April onwards to March of the following year. **This is also the time span for which precipitation weighted $\delta^{18}O$ ($\delta^{18}O_{pw}$) values are calculated.** ..."

Action: Done.

**R2:** *Based on that, what about evapotranspiration and changes of this during the modeled 1000 years? Have you accounted for that?*

**R2:** *Maybe, I am wrong, but my understanding of those isotope enabled GCMs was, that they have at least an 'evaporation on land' module. This should also account for fractionation effects on the evaporated water, but also for the remaining water, what you are interested in. Would it be an option to use those d18O values instead of that of precipitation (again weighted by the amount of infiltration)? Maybe not for this manuscript, but in any future application.*

See major comment #5 section.

Action: See major comment #5 section.

**R2:** *L219-221: Here you use the first time d18O$_{pw}$. It is not explained here nor somewhere else. What is this?*

It stands for precipitation weighted $\delta^{18}O$ . We will explain it in the section starting L200, also following the remark to L202-203.

Action: Done

**R2:** *In addition, I am sorry, but I do not understand exactly, why you are doing this Greens function approach? I get pretty ugly results in terms of mass balance with this, if*

*tau is small (e.g., 1, 2 ,3 years). For example using a tau of 1 year: even after 100 years waiting time, only 58 percent reached the cave. For tau=3 that are 84 %, what reached the cave after 100 years. For larger tau it works better, I admit. But as you use this filter with a residence time of 3 years, I would be happy if you please could explain why you used this way of residence time description. Have you normalized this somehow?*

Following Dee et al. (2015), we use a normalization such that $\int_0^{TSend} g(t)dt = 1$, where $g(t) = 1/\tau \cdot e^{-t/\tau}$ as in the manuscript. We will add this to the method section in the revised manuscript.

Action: Done.

**R1:** *L228 If you chose the highest correlation out of a large ensemble of possible solutions, how do you account for this when determining the significance of the correlation?*

**R2:** *L228-229: Is it possible to rephrase this sentence? I regret to not being able to understand, what you mean by this.*

We only choose the highest correlation estimate from significant cross-correlation estimates. If the cross-correlation between two speleothems using a specific pair of age-models is not significant at the 10% level, it is not chosen as a 'best fit'. We will clarify this in the manuscript in the respective section.

Action: Done.

**R2:** *Sec 3.2 and 3.3: Please explicitly state the number of used/available records for those approaches as those are most likely less, than that number mentioned in line 180.*

This is indeed valuable information that is missing. There are three groups of speleothems that we analyze, each with more strict selection criteria depending on the analysis. We will add this to each analysis step in the results section.

Action: Done.

**R2:** *L249: 'Generally, modeled values appear to be more depleted overall than the mean values of speleothem _18Odw.eq . . .'. Wouldn't this be a hint, that evaporation is important and should be accounted for (at least in any potential follow up study)?*

See major comment #5 section.

Action: See major comment #5 section.

**R1:** *L256-L257 If you calculate the regional lapse rate of 18O in the model you can estimate the contribution of the model orography to the d18O biases.*

**R1:** *L265 Did you try doing multivarite regression? To know the influence on one parameter you need to isolate it from the other parameters.*

We thank the reviewer for the interesting suggestion and will add it to the discussion section. We will consider them for a further planned study, where we will look more closely at the biases and the influence of different parameters. In this manuscript, however, we only want to give a first glance at the potential of the analysis.

Action: Add to the discussion section

**R1:** *L273 Add white space "both in the annual mean andfor ...".*
We added the space.
Action: Done.

**R1:** *L277-L278 "To analyze . . ." please rewrite this sentence more concisely and remember, again, that you are comparing the model to the data.*
**R2:** *L278-279: I am sorry but I am confused again. You are writing: 'The global distribution of variance ratios between d18Odw.eq and d18O (Fig. 5a) shows overall higher variability in the speleothem records than in the simulation'. I agree with this observation, but this is somehow in contrast to Fig 5 b and c, where the variance ratio between d18Odw.eq and d18O is smaller than one. Is it possible that in Fig 5a you are showing the variance ratio of d18Odw.eq and the already down sampled d18O of the model simulation?*
This is correct. Fig 5a shows the global distribution where the simulation is already down-sampled to model resolution. We will clarify this in the stated line and adjust in accordance with RC L42 as follows:

'..To analyze how similar the variability of the isotopic signal is **in the iHadCM3 climate model and in the speleothems**, we compare the total variance of **the simulation to the speleothem records** over the last millennium. The global distribution of variance ratios between $\delta^{18}O_{dw.eq}$ and **down-sampled** $\delta^{18}O$ (Fig. 5a) shows overall higher variability in the speleothem records than in the simulation, with local exceptions...'
Action: Done.

We also updated the caption of figure 5 to prevent further confusion:

"(a) Spatial visualization of the site-based dimensionless variance ratio $V_{Rec}/V_{Sim}$, **where the simulated $\delta^{18}O$ is down-sampled to record resolution,** based on LM1. ..."
Action: Done.

**R2:** *L288: I think the reference to Fig 5 is not correct, as this figure does not give a hint on a 'smoothed model pattern'.*
The expression is very unfortunate. We wanted to express, that the modeled variances at the cave locations are very similar globally. However, as stated, this is not visible from Fig.5. We will leave out this reference in a revised version and add more information to the "smoothed model pattern."
Action: Done.

**R2:** *Figure caption 6: 'the simulated _18Opw but down-sampled to the same temporal resolution as in (a) with 3 year filter. 'Just to be sure, as it is not written somewhere: You first applied the three year filter and then did the down sampling, correct?*

This is a good point. So far the 3-year filter was applied after the degradation of the sampling resolution. We will explore to what extent applying the filter first changes the results in the revision process and ensure that this is better described in the methods section.

Action: We changed the order such that we first applied the filter on the annual data and then did the down-sampling. This order is more intuitive. The change did however not alter the results obtained from the filtering. We adjusted the method section to explain the procedure.

**R1:** *L311 Add white space "3yr". I see space missing many places before "yr". Please check in general.*

Done. See comment to L153
Action: Done.

**R1:** *Figure 7, caption. Here, "insignificant" should be "non-significant". I assume you use the term "significant" in a statistical sense?*

Adjusted.
Action: Done

**R1:** *Section 4.4 Did you look at the relation of d18O to climate modes? See comment above to L78-L80. For example, the monsoon index (Vuille et al., 2005) might have a stronger imprint on d18O in the tropical Indian Ocean than local precipitation amount.*

See response to major comment #1.
Action: See response to major comment #1.

**R1:** *L320 "and the climate variable is shown." Change to "and the climate variable is also shown."?*

Done.
Action: Done.

**R1:** *L325 What about the correlation of LM2 and LM3 to the proxy data? Using the model ensemble could give a clue if the variability is related to forcing.*

See response to major comment # 3.
Action: See response to major comment # 3.

**R1:** *L332 $p < 0.1$ is not a strong significance criterion. How many samples are there? And in the first place can we expect much correlation between a single model run and observed climate? Changing the initial conditions of the model run would likely affect these*

[Figure]

Figure 2: Left: Visualization of the degrees of freedom corresponding to the analysis of the correlation estimates between simulated temperature and speleothem $\delta^{18}O$ as in Fig.7 of the manuscript. Right: The degrees of freedom in relation to the length of the time series.

*correlations, since this is just one realisation, no?*

Indeed, $p < 0.1$ is not generally strong criterion for significance of correlation estimates. However, we aim to choose criteria that are appropriate for both palaeoclimate archive and model data time series. Therefore, we need to balance this strictness and the expected level of false positives against that of data demands and the available number of samples $N$. In Fig. 7, in particular, we show both model-model ($N = 1000$) and model-proxy ($N$ varying) correlations. For irregular time series the effective degrees of freedom differ from the nominal value of $N$. The p-values for irregular series are estimated based on a t-distribution, with the degrees of freedom estimated from the temporal coverages $R_{x,y}$ and the persistence time $\tau_{x,y}$ as $N_{\text{eff}} = \min(\max(R_x/\tau_x, R_y/\tau_y, \text{na.rm=TRUE}), \max(N_x, N_y))$. This is implemented in the R package nest (https://github.com/krehfeld/nest, Rehfeld et al., 2011; Rehfeld and Kurths, 2014). For the regular time series p-values are calculated via Pearson's product moment correlation (via the function cor.test). We will add the degrees of freedom in the manuscript where different correlation estimators are used. In the case of the records, the estimated effective degrees of freedom range from $N_{eff} = 20$ to $N_{eff} = 470$, and they are generally similar to the length of the records (see histogram in Fig. 2). This indicates that the estimated persistence time is often of the order of magnitude of the sampling resolution.

Action: Larger paragraph included in the method section.

**R2:** *L332: 'We find 18, 15 and 22 significant correlations from 87 entities. . . ':*

*Out of curiosity, are the records/sites within those observed significant correlations in the three model runs always the same or are they varying? I mean if for example a record from one cave is significant for one run is the same cave record as well significant for the other simulations?*

From the total 55 significant correlations that we find for the three ensemble temperatures here, only half come from the same 12 entities, which again indicates at a very low signal to noise ratio in speleothems. We will add this, also for simulated precipitation and $\delta^{18}O$ , and include it in the discussion.

Action: Done. Also added as output to the code.

***R1:*** *Figure 8: I found the choice of colours confusing in Figure 8d. The smoothed lines are red and blue in the same shade as the markers for the correlation, which made me think at first that the smoothed lines were for the data marked of similar colours, which doesn't make sense. It's quite a busy plot. Consider making it easier to read by choosing different colours or making an extra subplot.*

***R2:*** *Figure caption 8: indicates instead of indicat.* Action: Done.

We will adjust Figure 8 according to the major comments #4 and also account for the color-confusion.

Action: Done.

***R2:*** *Figure caption 9: gridboxes instead of gridboxe*

Done.

Action: Done.

***R2:*** *L385: 'spatial pattern for the offsets were not distinguishable.' I would slightly disagree here. Isn't it the case that in warmer climates the offset is more negative than in colder climates? See fig. 4d. You even described the low to high latitude trend in the northern hemisphere by yourself in lines 249-250.*

We changed the section as follows:

"...Measured $\delta^{18}O_{dw.eq}$ followed general isotopic signature patterns as described by Dansgaard (1964). **The offsets are more positive in the extratropics of the Northern Hemisphere, which is also shown by their temperature dependency (Fig.4)."**

Action: Done

***R2:*** *L388: 'They find a stronger influence of seasonality of precipitation in warmer climates, highlighting the importance of a karst recharge model' Wouldn't it highlight the fact, that the model d18O values should be calculated as a precipitation weighted mean? Or even as infiltration weighted mean?*

See major comment #5 section.

Action: Done.

**R2:** *L389-390: 'observed a strong temperature dependency reflected in the offset and _18Odw.eq over the last Millennium, showing the influence of fractionation . . .': You claimed, that you accounted for the temperature-dependent isotope fractionation during CaCO3 precipitation to calculate the d18O if the drip-water equivalent. So I think, this reason can be safely excluded.*
**R2:** *I would evaluate it more likely that, as evaporation scales with temperature, the d18O values of precipitation have been changed by this process before the water even enters the epikarst zone. Thus, I would like to highlight it again (sorry for repeating myself), that using some infiltration weighted d18O mean is probably a better choice.*

We changed the section as follows:

"...Here, we also observed a strong temperature dependency reflected in the offset and $\delta^{18}O_{dw.eq}$ over the last millennium, showing the influence of cave internal processes on the $\delta^{18}O$ in drip water (Fig. 4) **but also additional fractionation processes or weighting through evaporation before the precipitated water enters the epikarst**. The higher offsets on the Northern Hemisphere possibly indicate a stronger influence of the continental effect. Still, from the records alone and with no karst-recharge **or evaporation** information, we were not able to distinguish specific climatic control regions. This requires a more thorough analysis including monitoring data **as well as more simulated variables**. ..."

Action: Done.

**R1:** *L396-L398 "In general, . . . " I don't follow this sentence. Seems like a leap in topic. How can you say anything about forced variability without analysing it? Also, I believe Jungclaus et al. (2010) are discussing the hemispheric mean temperature, while the speleothem d18O data is temperature, precipitation, evaporation and circulation dependent.*

We agree with the reviewer, the sentence structure is unfortunate. We removed the statement to forcings, and will include it in the analysis to major comment #2. For the analysis here we only want to address the total variability over the last millennium and adjusted the sentence as follows:

'... In general, the total variance **of the simulated** $\delta^{18}O$ **and of the speleothem isotopic signatures** over the last millennium are consistent. Differences in variance can, to some extent, be attributed to the sample resolution of the records, whereas down-sampling of simulated $\delta^{18}O$ decreases the variability on decadal time scales... '

Action: Done.

**R1:** *L410 "However, we find little regional consistency . . . " couldn't this be due to time scale uncertainties? You find no structure in correlation for the speleothem data in*

*Figure 8, but there could be a correlation/regional climate signal, just as well as there could be no correlation.*

*   ***R2:*** *L410-411: 'However, we find little regional consistency and high heterogeneity in the variance estimates from the speleothem records'. While there is indeed some high heterogenity in the variance, I wonder, if this could be an argument for a strong influence of cave internal processes? That is a tricky one.*

*   ***R2:*** *Later you discuss the correlation pattern, and this seems quite convincing that changes happen in the same direction and at about the same time at least at regional scale. Only the magnitude seems to change - as derived from the variance analysis. Wouldn't this rather mean, that each cave/stalagmite strengthen or weaken the initial climate signal, but that it is still contained? As you said, this alteration could happen by soil/karst/cave processes. This way, one could argue about the magnitude of the changes, but not about the variation itself. How do you think about this?*

The reviewers state different possible reasons for the little regional consistency: time-scale uncertainties and cave internal processes.

We agree with the first reviewer that the lack of correlation could be due to time scale uncertainties. Therefore we included these uncertainties in our cross-correlation analysis where we account for age-model sensitivity. There, as the reviewer points out, we find no structure in the correlation for the speleothem data. Under the assumption, that the true age time series is covered with the age-model ensembles, we account for all age uncertainties. However, this assumption may not be true. We will clarify these potential reasons for underestimated correlations in the paragraph.

We also agree with the second reviewer. As discussed for Fig 9, we do find generally positive correlation estimates between entities within one cave, showing local consistency as the reviewer describes. However, this is not true for all caves. A more thorough analysis might be needed to systematically discuss this relationship, and also a larger global sample size of caves with multiple speleothems. Even though the SISALv2 database provides a very large dataset, for the last millennium only 12 caves exhibit multiple entities, that fit our analysis criteria. We will include a section on this in the discussion following L410-411, including the stated thoughts from above.

We change the section from L410 with the suggestion of both reviewers as follows:

'...However, we find little regional consistency and high heterogeneity in the variance estimates from the speleothem records. These findings point to the influence of cave internal **and karst** processes **on meteoric** $\delta^{18}O$ or the impact of seasonally filtered data captured by speleothems, which is in agreement with McDermott et al. (2001). **Age uncertainties, that are not covered by the age-model ensembles, could also be responsible for the low similarity between isotopic signals of neighboring speleothem entities.**

**R2:** *L416-417: 'Studies observing transit time in karst systems find increases in drip rate after an increase in precipitation e.g. after days (Riechelmann et al., 2011).' I suggest to term this Riechelmann et al., (2011) work rather as a study, which investigates the cave reaction time to precipitation events - not a transit time study. And the cave reaction on heavy rain events is often fast (as observed in other caves as well), but it will not change the transfer time (too much). The residence time can only be found by tritium or other appropriate tracer isotopes (as you correctly describe below.)*

We will replace "transit time" with the suggested more precise term "cave reaction time" in this sentence in the revised version.

Action: Done.

**R1:** *L428 "longer than50yr" Spaces!*

Done.

Action: Done.

**R1:** *L428 "by 4% (3, 4)" Upper confidence bounds same as median? Or is this due to the number of significant digits?*

We adjusted for one extra digit, so we arrive at $4.0\%$ $(3.3, 4.4)$.

Action: Done.

**R1:** *L434 "However, no systematic pattern and few significant correlations were found for the speleothem records (Fig. 7)." Again, I'm really not surprised that there is no correlation between a free running simulation and the proxy data. There might be forced common variability between model run and proxies (volcanic, solar), but then you need to check the model and proxy response to forcings.*

We will relate to this according to our analysis to major comment #2 and include the results and findings in this paragraph in the discussion.

Action: Done

**R2:** *L442: I guess, cave monitoring would not only help to compare two caves but also to improve the comparability between a cave and climate model results. You could add this as well. Furthermore, I think that not only monitoring would help, but that (model-based) weighted infiltration values would also help (as written already earlier). Even with your sentence in line 438 you imply so by yourself.*

We changed the section as follows:

"... Further drip water monitoring studies **combined with a comparison to model data output would help to**characterize the seasonality of individual caves **and would, therefore, lead to deeper** understanding of which climatic signal is captured by speleothems

and enhance comparability between different caves. ..."
Action: Done.

**R2:** *L455: 'but could also be influenced by non-climatic overprints on the _18O signal up to the centennial scale.' I would argue, that nearly all changes in the processes in the cave are climate-driven. Unfortunately, sometimes they counter the pure climate imprint and on other locations they amplify it.*

With non-climatic, we meant to say environmental overprints such as changes in the epikarst which may result in different growth rates, that are not directly climate driven. We change the section as follows:

"...In part, this can be due to heterogeneous temporal resolution, but could also be influenced by non-climatic **environmental** overprints on the $\delta^{18}O$ signal up to the centennial scale. ..."
Action: Done.

**R1:** *L464 "We use a three member initial-condition ensemble from a single iGCM in this study." Please describe the model ensemble initiation in the methods section.*

See comment L123
Action: Done.

**R1:** *L470 ". . . as suggested by Dalaiden et al. (2020)." There are lots of examples of offline data assimilation. Maybe provide a few more? E.g., see references in introduction of Sjolte et al. (2020). Ice core data is synchronized using volcanic markers. Any particular age-model related uncertainties to take into account that might complicate the assimilation of speleothem data?*

Data assimilation would indeed be a very interesting application for our dataset. As speleothems are dated radiometrically, the uncertainty of the age depends primarily on the concentration of the relevant isotopes used and the age limits of the method, e.g. the secular equilibrium for U/Th-dating (Scholz and Hoffmann, 2008). An additional source of uncertainty stems from growth irregularities and outliers. However, in contrast to ice cores, the uncertainty does not per se increase with depth, and no synchronization with volcanic markers are needed. Similar to this study, age uncertainties can be accounted for by the provided age-model ensembles (Comas-Bru et al., 2020). We will add this explanation and also more examples as suggested to the section corresponding to L470.
Action: Done.

**R2:** *L476: 'as a bias of 1C in the simulated temperature would account for a change in _18Odw.eq of approximately 2 ‰' If you really take the d18O-T relationship of Tremaine (or any similar) this statement appears to be wrong. Fractionation during CaCO3 precipitation is T-dependent by ∼0.25 permil per °C. So this would mean an offset of 1 permil, if the*

*modeled T is wrong by 4 °C. I guess this would only be the case in mountainous regions, were the orography of the model is not close enough to the true altitude of the cave. You mentioned some examples earlier in your manuscript.*

**R2:** *L477:'A bias of 1 ‰in the _18Odw.eq however, accounts for a temperature change of 0.1_C for the lowest simulated annual mean cave temperature (3.1_C in Norway), and a change of 13.1_C for the highest simulated annual mean cave temperature (32.5_C in the tropics).' This puzzles me now quite a lot. You said in your earlier sentence, that 1 °C is 2 permil. This does not fit with your statement in this sentence. Maybe, you mean something different?*

Apologies for the inconsistency. We have checked the calculation. The corrected numbers do not change our interpretation. We will clarify the calculation in the manuscript and change the numbers in this section as follows:

"...Knowing the actual temperature history of the caves better could strongly reduce the uncertainty, as a bias of $\Delta 1°C$ in the simulated temperature would account for a change in $\delta^{18}O_{dw.eq}$ of approximately $\Delta 0.2$‰. A bias of $\Delta 1$‰ in the $\delta^{18}O_{dw.eq}$ however, accounts for a temperature change of $4.5°C$ for the lowest simulated annual mean cave temperature $(3.1°C$ in Norway), and a change of $5.5°C$ for the highest simulated annual mean cave temperature $(32.5°C$ in the tropics). ..."

Action: Done.

**R1:** *L483 ". . . such as d13C cannot (yet) be implemented in GCMs . . . " It's not that far away (Scholze et al., 2008; Camino-Serrano et al., 2019).*

We will include the references in the conclusion.

Action: Done.

**R1:** *L503 ". . . low signal-to-noise ratios . . . " For the speleothem data?*

Clarified:

[revised manuscript text omitted]